# The Future of MLLM Prompting is Adaptive:
# A Comprehensive Experimental Evaluation of Prompt Engineering Methods for Robust Multimodal Performance

**Anwesha Mohanty**                                     *anwesha.mohanty@ucd.ie*
*CeADAR: Ireland's Centre for AI*
*University College Dublin, Belfield, Dublin 4, Ireland*

**Venkatesh Balavadhani Parthasarathy**                *venkatesh.parthasarathy@ucd.ie*
*CeADAR: Ireland's Centre for AI*
*University College Dublin, Belfield, Dublin 4, Ireland*

**Arsalan Shahid**                                      *arsalan.shahid@ucd.ie*
*CeADAR: Ireland's Centre for AI*
*University College Dublin, Belfield, Dublin 4, Ireland*

**Reviewed on OpenReview:** *https://openreview.net/forum?id=B1L8HrjoA1*

## Abstract

Multimodal Large Language Models (MLLMs) are set to transform how machines process and generate human-like responses by integrating diverse modalities such as text, images, and code. In this study, we specifically focus on text–image multimodal reasoning and understanding, evaluating their performance across diverse task categories. Yet, effectively harnessing their capabilities hinges on optimal prompt engineering. We present a comprehensive experimental evaluation of seven prompt engineering methods applied to 13 open-source MLLMs over 24 tasks spanning Reasoning and Compositionality, Multimodal Understanding and Alignment, Complex Code Generation and Execution, and Knowledge Retrieval and Integration. Our approach stratifies models by parameter count into Small ($< 4B$), Medium (4B–10B), and Large ($> 10B$) categories and compares prompting techniques including Zero-Shot, One-Shot, Few-Shot, Chain-of-Thought, Analogical, Generated Knowledge, and Tree-of-Thought. Our experiments reveal that while Large MLLMs excel in structured tasks such as code generation and execution, achieving accuracies as high as 96.88% under Few-Shot prompting. In multimodal understanding and alignment (with relevance scores reaching 100% using Zero-Shot prompting), all models struggle with complex reasoning and abstract model understanding, often yielding accuracies below 60% and high hallucination rates. Notably, structured reasoning prompts (Chain-of-Thought, Analogical, Generated Knowledge and Tree-of-Thought) frequently increased hallucination up to 75% in small models and led to longer response times (exceeding 20 seconds in Large MLLMs), while simpler prompting methods (One-Shot and Few-Shot) provided more concise and efficient outputs. Our findings underscore that no single prompting method uniformly optimizes all task types. Instead, adaptive prompting strategies that combine the strengths of example-based guidance with selective structured reasoning are essential to enhance robustness, efficiency, and factual accuracy in MLLMs. Our work provides critical insights and actionable recommendations for optimizing prompt engineering in text–image multimodal contexts, paving the way for more reliable deployment of MLLMs in real-world applications ranging from AI-assisted coding and knowledge retrieval to visual–textual content understanding.

# 1   Introduction

The rapid evolution of Multimodal Large Language Models (MLLMs) has catalyzed a paradigm shift in bridging visual representation learning with natural language understanding, thereby enabling sophisticated multimodal reasoning and broader real-world applicability. Although conventional Large Language Models (LLMs) have demonstrated impressive scaling behaviors (Zeng et al., 2025), the leap toward multimodality is primarily driven by the advent of increasingly capable LLM backbones (Liu et al., 2024b). In this work, we focus specifically on the text–image modality, which remains the most mature and widely benchmarked setting for MLLMs, while recognising that the broader class of multimodal systems can also incorporate audio, video, and other sensory data. Nonetheless, a critical gap persists in the seamless integration of visual processing components with language models. Many current MLLMs employ vision transformer-based architectures, most notably CLIP (Radford et al., 2021; Zhai et al., 2023) as feature extractors, often relegating visual understanding to a secondary role rather than embedding it integrally within the reasoning pipeline. Although alternative approaches, such as self-supervised learning methods exemplified by DINO (Oquab et al., 2023), have shown promise, systematic studies that jointly address architectural design and contextual prompting strategies remain scarce (Lu et al., 2023; Liu et al., 2023).

MLLMs are inherently designed to process heterogeneous data modalities, thereby expanding their task coverage significantly. However, their effective instruction-following and context comprehension are hindered by suboptimal integration of vision and language modules, inadequate prompt design, and inconsistencies in input representation (Liu et al., 2023; Arif et al., 2025). While many evaluations focus on benchmarking MLLM performance across diverse tasks, they frequently overlook critical factors such as:

- The compatibility between MLLM architectures and the evaluation dimensions pertinent to specific tasks.
- The alignment between training datasets and evaluation benchmarks, ensuring models are optimally calibrated for the tasks they are assessed on.
- The effectiveness of prompt engineering techniques in enhancing multimodal understanding and robust instruction-following.

To address these shortcomings, our study explores an evaluation-centric and prompt-based framework. We begin by surveying prevalent use cases and delineating the task requirements and evaluation aspects essential for effective multimodal reasoning. This analysis informs our selection of MLLMs, highlighting both architectural diversity and contextual considerations. Subsequently, we rigorously investigate a range of prompt engineering methodologies, examining their impact on enhancing multimodal context integration and overall instruction adherence. In doing so, we propose a comprehensive evaluation framework that elevates multimodal instruction-following as a pivotal performance metric for MLLMs.

## 1.1   MLLM Architecture and Applications

Although LLMs are optimized for text-based inputs and outputs, MLLMs extend these capabilities to incorporate images, videos, and audio, necessitating more complex architectural integrations. Typically, an MLLM comprises three primary components: a Modality Encoder, a Transformation Layer, and an LLM backbone, as depicted in Figure 1 (Vaswani et al., 2023; Zhang et al., 2024a).

MLLMs, especially those with the capability to process images and videos, have seen rapid development, as underscored by the HuggingFace VLM Leaderboard (Duan et al., 2024), which now lists more than 200 models introduced since 2022. Notable contributions in this space include the categorization of vision language models (VLMs) by (Ghosh et al., 2024) and (Yin et al., 2024a), which analyze architectures, training methodologies, and evaluation metrics.

Specialized applications further highlight the potential of MLLMs. (Niu et al., 2024) demonstrate the integration of multimodal data in healthcare for clinical decision-making and medical imaging analysis. (Wang et al., 2023) introduce FinVis-GPT for financial graph analysis through custom datasets and instruction-based annotations, while (Liang et al., 2024) present DrugChat for the prediction of drug molecule properties.

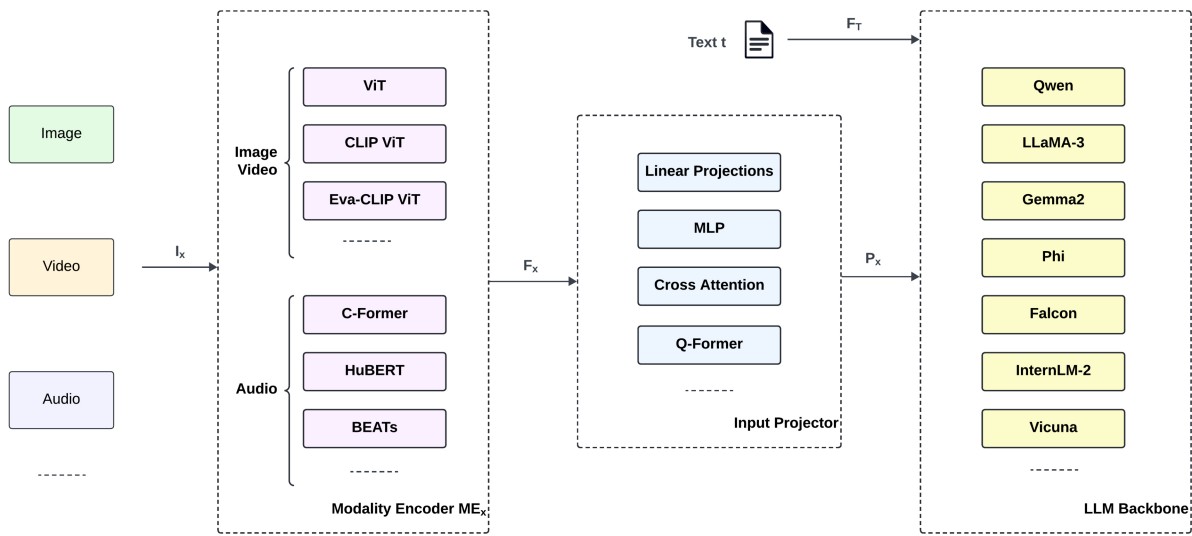

Figure 1: A high-level overview of a typical MLLM pipeline. Multiple input modalities (e.g., images, video, audio) are first processed by dedicated modality encoders (e.g., ViT (Dosovitskiy et al., 2020), CLIP-ViT (Radford et al., 2021), BEiT (Bao et al., 2021)). The encoded features are then projected or transformed via components such as linear projections, MLPs, or cross-attention to align with the text embedding space. Finally, the LLM backbone (e.g., Qwen, LLaMA, Falcon) integrates these multimodal features for unified reasoning and generation.

Furthermore, (Bewersdorff et al., 2025) propose a framework for integrating MLLMs into science education, and (Yang et al., 2024b) develop SEED-Story for generating multimodal narratives using novel attention mechanisms.

A common architectural trend among these models is the use of a vision encoder coupled with an LLM, often linked via a transformation layer (Zhang et al., 2024a). While CLIP-based models excel in zero-shot prompting scenarios (Kojima et al., 2022), they frequently fall short on fine-grained tasks. In contrast, ViT-based architectures (Dosovitskiy et al., 2020; Wu et al., 2020; Xiao et al., 2021) offer robust spatial attention but incur higher computational costs and reduced interpretability.

## 1.2 Evaluation Challenges and Recent Advances

The HuggingFace VLM Leaderboard (Duan et al., 2024) reveals heterogeneous performance across benchmarks; ranging from Vision Q&A and OCR to RealWorldQA, ChartQA, and MathQA. Notably, leading models such as Step-1o (Duan et al., 2024) underperform on benchmarks like HallusionBench (Guan et al., 2024), highlighting nuanced trade-offs in current evaluation schemes. Most benchmarks follow a zero-shot evaluation approach with their own set of metrics, typically presenting models with a question and multiple-choice options based on an image. The use of simple and uniform prompts for all samples can lead to an underestimation of model capabilities. The sensitivity of both LLMs and MLLMs to prompt variations, as noted by (Xie et al., 2024), further suggests that uniform prompt designs may fail to capture a model's true potential.

Recent studies have attempted to overcome these limitations through heuristic and ensemble prompting techniques in zero-shot and few-shot settings (Brown et al., 2020; Sivarajkumar et al., 2024). Enhanced evaluation frameworks, such as those proposed by (Hao et al., 2025), indicate that even state-of-the-art models like GPT-o1 (Jaech et al., 2024) struggle to exceed 50% accuracy in multimodal reasoning tasks despite employing Chain-of-Thought (CoT) prompting (Ge et al., 2023). Structured CoT (SCoT) approaches have

demonstrated improvements of up to 13.79% over traditional CoT methods (Brown et al., 2020; DeepSeek-AI et al., 2025).

Complementing these evaluations, (Jiang et al., 2024b) provide a holistic assessment of Large Vision-Language Models (LVLMs) across both specialized tasks (e.g., object detection and medical diagnosis) and general tasks (e.g., object counting and spatial reasoning). Their evaluation, which includes models such as MiniGPT-v2 (Chen et al., 2023a), LLaVA-1.5 (Liu et al., 2024a), and Shikra (Chen et al., 2023b), along with assessments via GPT-4V (OpenAI, 2025) highlights ongoing challenges including limited cognition, object hallucination, and robustness issues. Additionally, research by (Li et al., 2025) explores Structured CoT for improving code generation by incorporating programming principles (sequential, branch, and loop) to guide reasoning steps before coding; however, code generation remains underexplored within the MLLM context.

Different models exhibit varying sensitivities to the same prompt changes, and existing evaluation frameworks often fail to address prompt-induced bias, leading to unfair comparisons. These architectural limitations, inconsistencies in reported benchmark accuracies, and current evaluation methodologies raise significant questions about the true potential of these models.

### 1.3  Our Approach and Contributions

To address these challenges, we developed a comprehensive experimental framework to rigorously evaluate 13 open-source MLLMs across four key aspects: Multimodal Reasoning, Model Understanding, Knowledge Retrieval, and Code Generation. The selected models represent a stratified sample based on parameter sizes and are paired with diverse LLM backbones to facilitate a robust evaluation of both performance and real-world applicability. The main contributions of this paper include:

1. A comparative study of seven prompt engineering methods applied to 13 open-source MLLMs, evaluating their performance on 24 tasks across four evaluation aspects. In doing so, we design a diverse set of tasks, create standardized prompt templates, and share both the datasets and templates to ensure reproducibility.

2. An in-depth analysis of how prompt engineering strategies interact with task types and model scales, providing insights into model performance, reliability, and resource requirements, as well as offering best practices and actionable recommendations for optimizing performance in real-world applications.

## 2  Methods

To understand the impact of prompt engineering techniques across diverse tasks and evaluation metrics, we employed a four-staged experimental design and evaluation framework including:

- **Stage 1 – Defining Core Evaluation Aspects:** We define four Evaluation Aspects (EAs) to provide a comprehensive analysis of model performance across reasoning, multimodal interpretation, code generation, and knowledge integration. Detailed discussion of these aspects is provided in Section 2.1. For each EA, we curated a set of tasks designed to challenge the models' ability to integrate and process multimodal inputs (primarily images) across real-world scenarios. The corresponding tasks are listed in Tables 1, 2, 3, and 4.

- **Stage 2 – Review and Selection of MLLMs:** A diverse set of 13 open-source MLLMs were chosen based on their architecture, parameter size, and availability (as discussed in Section 2.2). These models showcase the breadth of current MLLM capabilities by combining different image and text models, each built upon a distinct text encoder.

- **Stage 3 – Selection of Prompt Engineering Methods:** We apply seven prompting methods including Zero-Shot, One-Shot, Few-Shot, Chain-of-Thought, Analogical, Generated Knowledge, and Tree-of-Thought prompting. Section 2.3 provides further discussions on selected methods.

- **Stage 4 – Evaluation Framework:** Our experimental setup was designed to ensure consistency and reliability across multiple tasks and prompting techniques. Model outputs were evaluated along two primary dimensions. The first dimension focuses on task performance, using four key metrics: accuracy, relevancy, conciseness, and hallucination (detailed in Section 2.4). The second dimension assesses resource consumption, including inference time and memory consumption. A comprehensive manual review process is devised to analyse MLLM outputs to assess aforementioned metrics.

## 2.1 Stage 1: Defining Core Evaluation Aspects

Evaluating MLLMs is crucial to understanding their capabilities, limitations, and applicability across diverse domains. Prior studies (Li et al., 2024a; Wu et al., 2025; Yu et al., 2023) have explored various evaluation aspects, from perceptual understanding and compositional reasoning to multimodal alignment and task-specific problem solving. Unlike traditional unimodal models, MLLMs require frameworks that assess their ability to process, integrate, and generate outputs from multiple modalities (e.g., text, images, and in some cases audio and video). Reviews such as (Yin et al., 2024b) typically categorize evaluations into general multimodal understanding, task-specific assessments, and trustworthiness metrics, ensuring that models transition from broad reasoning to reliable, specialized real-world applications. Motivated by these insights and based on our review of current applications and evaluations in MLLM landscape (see Appendix A), we select four core Evaluation Aspects (EAs) for their broad impact and practical relevance:

1. **Reasoning and Compositionality (EA1):** This aspect centers on the model's ability to process textual and visual cues, perform logical deductions, and synthesizing disparate information into coherent outputs. It tests capabilities such as Multi-Step Problem Solving, Pattern Recognition, Logical Deduction, and Compositional Synthesis (see Table 1).

2. **Multimodal Understanding and Alignment (EA2):** Evaluates how well the model aligns, integrates, and interprets information across modalities. It is critical for tasks that demand accurate cross-referencing between text and visual data. Key capabilities to evaluate include cross-modal referencing and alignment image interpretation and description, and consistency in visual–textual context (see Table 2).

3. **Complex Code Generation and Execution (EA3):** This aspect evaluates the model's ability to interpret instructions, extract data from visual inputs, and generate executable code. It is essential for programming tasks where code must be both syntactically correct and logically coherent. The key evaluation capabilities include data extraction from visual input, programmatic transformation and logic construction, and accurate code synthesis, debugging, and explanation (see Table 3).

4. **Knowledge Retrieval and Integration (EA4):** Focuses on the model's ability to recall, verify, and merge factual information from textual and visual sources into coherent responses. This is critical for domains such as research, journalism, medicine, and data science. The key capabilities to evaluate include factual recall and verification, domain-specific context and explanation, and cross-modal knowledge synthesis (see Table 4).

For each EA, we curated tasks to challenge models in realistic, application-oriented scenarios. Task objectives, and key challenges are summarized in the following tables and design rationales are elaborated in Appendices D.1, D.2, D.3, and D.4. Full task descriptions and expected outputs are available in the supplementary material S1 to S4.

## 2.2 Stage 2: Review and Selection of MLLMs

This stage consists of two parts. First, we review key models current MLLM landscape, encompassing both proprietary and open-source developments. Second, we detail the selection criteria and present the 13 open-source MLLMs chosen for our evaluation.

Proprietary models such as OpenAI's GPT-4o, GPT-4.5 Preview (OpenAI, 2025), Anthropic's Claude 3 (Anthropic, 2024), and Google's Gemini series (DeepMind, 2025) demonstrate strong multimodal reasoning, particularly in instruction-following and contextual understanding. For instance, GPT-4 introduced

Table 1: Overview of Evaluation Aspect 1 (EA1): Reasoning and Compositionality, comprising four tasks (T1 to T4) focused on visual pattern recognition, logical deduction, mathematical reasoning, and narrative synthesis.

| Task(s) | Objective(s) | Key Challenges |
|---|---|---|
| **EA1_T1:** Pattern Recognition in Visual Sequences | Test the model's ability to detect and generalise patterns in a sequence of related images or diagrams. | Identifying logical/visual patterns and extrapolating rules from limited data. |
| **EA1_T2:** Logical Deduction from Text and Simplified Diagram | Evaluate the model's capacity to interpret textual instructions alongside a simple diagram to reach a correct conclusion. | Integrating textual clues with diagrammatic cues and ensuring consistency in multi-step reasoning. |
| **EA1_T3:** Mathematical Puzzle with Visual Data | Assess how the model handles numeric computations and interprets simple visual representations (e.g., shapes or charts). | Bridging quantitative reasoning with visual elements and avoiding arithmetic errors. |
| **EA1_T4:** Story Synthesis from Text and Image | Check the model's ability to create a coherent narrative by merging textual descriptions and a relevant image. | Maintaining logical flow in narrative form and blending visual context with text. |

Table 2: Overview of Evaluation Aspect 2 (EA2): Multimodal Understanding and Alignment, including four tasks (T1 to T4) that test the integration and interpretation of information across text, images, and charts.

| Task(s) | Objective(s) | Key Challenge(s) |
|---|---|---|
| **EA2_T1:** Image-Text Matching and Explanation | Verify the model's capacity to match an image with a corresponding text description and explain the match. | Correctly identifying key features and ensuring textual alignment with visual elements. |
| **EA2_T2:** Inferring Context from Combined Modalities | Determine how the model integrates separate text and image inputs to deduce higher-level context. | Seamlessly fusing diverse information sources and handling ambiguous or incomplete data. |
| **EA2_T3:** Cross-Modal Translation | Evaluate the model's ability to translate visual information (e.g., symbols or icons) into meaningful text. | Accurately handling symbolic representations and capturing fine-grained details. |
| **EA2_T4:** Aligning Data from Charts and Text | Assess how well the model interprets and aligns quantitative data from a chart with textual analysis. | Avoiding misinterpretation of graphical data and integrating numerical details with text. |

multimodal inputs to process both text and images, while Claude 3 (including its variants Claude 3 Opus, Sonnet, and Haiku) has been optimized for text–image reasoning tasks. Despite their robust performance, the closed-source nature of these models limits customization, transparency, and independent research. In contrast, the open-source ecosystem has rapidly evolved, providing competitive alternatives with enhanced

Table 3: Overview of Evaluation Aspect 3 (EA3): Complex Code Generation and Execution, comprising eight tasks (T1 to T8) that evaluate a model's ability to generate executable code from visual and textual inputs across a range of structured reasoning and programming challenges.

| Task(s) | Objective(s) | Key Challenges |
|---|---|---|
| **EA3_T1:** Data Visualization from an Image of a Table | Generate a script that converts table data in an image into a visualization (e.g., bar chart). | Handling OCR-like interpretation, mapping image data to structured format, and producing valid code. |
| **EA3_T2:** Drawing a Shape Based on an Image | Produce code that programmatically draws a shape using visual hints from an image. | Translating visual references into geometric coordinates and ensuring syntax correctness. |
| **EA3_T3:** Calculating a Sum from Text in an Image | Write a function to parse textual or numeric data from an image and compute a sum. | Accurate text extraction, handling parsing errors, and verifying arithmetic accuracy. |
| **EA3_T4:** Creating a Dictionary from an Image of a Chart | Convert labels and values in a chart image into a dictionary or key-value structure. | Extracting structured data, ensuring correct type conversion, and code clarity. |
| **EA3_T5:** Summing Prices from a Shopping List Image | Generate code to read item prices from an image of a shopping list and calculate the total. | Handling varied text formats, summing accurately, and managing currency symbols or decimals. |
| **EA3_T6:** Parsing a Simple CSV Structure from an Image | Build a script that interprets an image containing CSV-like text and converts it into a data table. | Accurate extraction of rows/columns and handling formatting inconsistencies. |
| **EA3_T7:** Generating Fibonacci Sequence Based on Image Instruction | Produce code to generate a Fibonacci sequence following instructions specified in an image. | Interpreting visual instructions accurately and ensuring logical code correctness. |
| **EA3_T8:** Responding to a Flowchart Image | Interpret a flowchart diagram and output code or logic to implement the described process. | Translating flowchart nodes into algorithmic steps and ensuring overall coherence. |

accessibility and community-driven improvements. Models such as Meta's LLaMA series (Meta, 2024a), Mistral (MistralAI, 2024), Falcon (Malartic et al., 2024), and DeepSeek (DeepSeek-AI et al., 2025) offer greater flexibility in fine-tuning and deployment. Nonetheless, open-source MLLMs still face challenges in matching proprietary models' instruction-following, contextual learning, and multimodal alignment. A comprehensive discussion of the various distinct MLLMs including their architectural designs, multimodal processing capabilities, and evaluation suitability is provided in Appendix B.

After conducting an in-depth review of various models, several factors influenced the exclusion of specific models from our selection:

Table 4: Overview of Evaluation Aspect 4 (EA4): Knowledge Retrieval and Integration, comprising eight tasks (T1 to T8) that assess how effectively a model combines visual cues and textual context to retrieve, interpret, and explain domain-specific knowledge.

| Task(s) | Objective(s) | Key Challenges |
|---|---|---|
| **EA4_T1:** Historical Monument Identification and Explanation | Identify a famous monument from an image and provide its historical context. | Handling historical facts, distinguishing similar monuments, and accurately explaining cultural significance. |
| **EA4_T2:** Scientific Data Interpretation from Graph and Text | Integrate textual and graphical data to answer a scientific question and summarize key findings. | Extracting relevant data from graphs, merging with textual context, and ensuring scientific accuracy. |
| **EA4_T3:** Medical Image Analysis with Knowledge Integration | Provide a brief medical interpretation from an image (e.g., an X-ray) along with a textual description of symptoms. | Understanding medical terminology, ensuring factual accuracy, and integrating visual and textual cues. |
| **EA4_T4:** Cultural Artifact Interpretation | Identify an artifact from an image and explain its cultural/historical background using textual clues. | Combining historical and cultural knowledge accurately and presenting a coherent explanation. |
| **EA4_T5:** Integrating Knowledge from a Map and Text Description | Combine visual map data with textual instructions to address location-based or geographical queries. | Accurately interpreting map symbols and reconciling textual directions with visual references. |
| **EA4_T6:** Integrating Information from a Chart and Article | Merge insights from a chart (e.g., population growth) with an accompanying article to produce a synthesized summary. | Ensuring correct numerical interpretation, linking data points to textual arguments, and forming a comprehensive summary. |
| **EA4_T7:** Multimodal Fact Checking | Verify the factual accuracy of a statement by cross-referencing an image (e.g., a photograph) with textual sources. | Cross-validating visual evidence with textual claims and identifying potential inconsistencies. |
| **EA4_T8:** Integrating Visual Art and Historical Context | Explain an artwork shown in an image, including its historical context and cultural details. | Recognizing artistic styles, contextualizing the piece historically, and referencing relevant artistic movements. |

- Modality Focus – Models explicitly designed for video, audio, or long-sequence temporal reasoning (e.g., VITA (Fu et al., 2024a), Long-VITA (Shen et al., 2025), mPLUG-Owl3 (Ye et al., 2024)) were not included, as our evaluation focuses on text-image multimodal reasoning rather than temporal or multi-frame processing.

- Task-Specific Specialization – Some models are highly optimised for specific domains rather than general-purpose multimodal reasoning. For example, MoAI (Lee et al., 2024) is designed for OCR-centric tasks, ChatRex (Jiang et al., 2024a) focuses on object detection, and ViP-LLaVA (Cai et al.,

2024) supports region-aware multimodal interaction. As a result, these models are less suitable for a structured comparative analysis.

- Reliance on External Modules – Models like Molmo (Deitke et al., 2024) and Cambrian-1 (Tong et al., 2025), which incorporate external retrieval mechanisms or specialized visual processing modules, introduce dependencies that complicate direct performance comparisons across standardized benchmarks.

- Performance-Compute Tradeoff – While Falcon2-11B (Malartic et al., 2024) and MiniCPM (Yao et al., 2024b) are highly efficient, they are optimized for lightweight multimodal interactions rather than advanced vision-language compositionality, making them less aligned with our focus on deeper reasoning and complex multimodal integration.

- Maturity and Benchmarking Limitations – Some recently released models, such as Meteor (Lee et al., 2025) and Cambrian-1 (Tong et al., 2025) lack comprehensive benchmarking on widely used multimodal datasets, making it difficult to systematically compare their performance against well-established counterparts.

Taking these factors into account, we structured our model selection process to ensure a balanced, computationally feasible, and diverse evaluation of open-source MLLMs.

We selected 13 open-source MLLMs for detailed evaluation, focusing on models that offer a balanced combination of scalability, architectural diversity, and multimodal reasoning capabilities. Our selection criteria include:

- Parameter Scale and Architectural Diversity - Models are categorized as Small (<4B), Medium (4B–10B), and Large (>10B) to capture variations in computational demands and performance.

- Multimodal Integration - Preference was given to models integrating different vision encoders (e.g., ViT, SigLIP, CLIP, EVA) with various language models (e.g., Qwen, Gemma, Llama, Phi), enabling an exploration of how vision–language pairings influence task performance.

- Open-Source Availability - Only models with fully open-source code and weights were considered to ensure reproducibility and community accessibility.

- Practical Feasibility - Due to GPU constraints, models exceeding 15B parameters were generally excluded from direct evaluation. However, we incorporated one model above 15B parameters using quantization techniques to examine the trade-offs between model size and inference efficiency.

For each category, four models were selected (with five in the large model group due to the inclusion of a quantized model). This strategy ensures a fair comparison across small, medium, and large models while addressing key questions related to parameter scaling, vision–language integration, and computational efficiency.

Table 5 summarizes the final list of models along with key specifications such as parameter count, release date, underlying language and vision models, and image input support.

## 2.3 Stage 3: Selection of Prompt Engineering Methods

Understanding multimodal content requires deep multimodal knowledge and models must not only grasp information within each modality but also accurately infer how these modalities interact to support effective reasoning (Yang et al., 2023). Prompt engineering has emerged as a straightforward and efficient method for guiding LLMs and MLLMs, enabling enhanced performance in complex reasoning tasks. This approach generally involves two types of prompts: instruction-based and example-based (Bhattacharjya et al., 2024). Instruction-based prompts include system-level prompts that establish overarching guidelines and task-specific prompts tailored to particular objectives, while example-based prompts rely on a few illustrative examples to define desired input-output relationships.

Table 5: List of Models Finalized for Evaluation. This table includes the model name, parameter size, release date, and details of the language and vision model combination.

| Model | Params (B) | Release Date | Language Model | Vision Model |
|---|---|---|---|---|
| InternVL-2 | 1 | 8-Jul-24 | Qwen2.5-0.5B | InternViT-300M |
| Qwen2-VL | 2 | 29-Aug-24 | Qwen2-1.5B | ViT-600M |
| MiniMonkey | 2.2 | 9-Aug-24 | InternLM2-1.8B | InternViT-300M |
| Paligemma-3B-mix-448 | 3 | 14-May-24 | Gemma-2B | SigLIP-400M |
| Phi-3.5 VLM | 4 | 21-May-24 | Phi-3.5 | CLIP ViT-L/14 |
| LLaVA OneVision-7B | 8 | 14-Sep-24 | Qwen2-7B | SigLIP-400M |
| Ovis 1.5-Llama 3-8B | 8 | 17-Jun-24 | Llama-3-8B-Instruct | SigLIP-400M |
| GLM-4v-9B | 9 | 30-Jul-24 | GLM-4-9B | EVA-02-5B |
| Ovis-1.6 | 10.2 | 17-Jun-24 | Gemma2-9B-lt | SigLIP-400M |
| Llama3.2-Vision | 11 | 25-Sep-24 | Llama 3.1 | ViT |
| Pixtral | 12 | 17-Sep-24 | Nemo-12B | ViT-400M |
| OmChat V2 | 13 | 6-Jul-24 | Qwen2-7B | InternViT-6B |
| InternVL-2 | 26 | 8-Jul-24 | InternLM2-20B | InternViT-6B |

Numerous studies (Jiang et al., 2022; Zamfirescu-Pereira et al., 2023) have identified two key challenges in prompting: crafting effective prompts and evaluating their efficacy. In particular, example-based approaches have proven effective in guiding large models (Mann et al., 2020; Wei et al., 2022; Yao et al., 2024a). Although many interactive systems support prompt engineering, most focus predominantly on textual or limited visual inputs. This narrow focus overlooks the intricate interactions between modalities, thereby limiting the development of prompts that fully leverage the contextual richness of multimodal inputs to enhance reasoning (Zamfirescu-Pereira et al., 2023).

Models can quickly adapt to new downstream tasks in few-shot or even zero-shot settings without requiring retraining (Liu et al., 2023). As exemplified by the pioneering Chain-of-Thought (CoT) prompting technique, which prompts LLMs to generate intermediate reasoning steps, mirroring human cognitive processes (Wei et al., 2022; Kojima et al., 2022; Zhang et al., 2022); the concept has been extended to the multimodal domain (M-CoT) in several studies (Rose et al., 2023; Zhang et al., 2023; Ge et al., 2023). Analogical Reasoning Prompting leverages shared structural similarities between scenarios to unlock a model's analogical reasoning abilities (Yasunaga et al., 2023), while Generated Knowledge Prompting encourages models to generate additional background knowledge to enhance reasoning (Liu et al., 2021; 2023). Tree-of-Thought (ToT) prompting further extends CoT by structuring reasoning into a decision tree that explores multiple pathways before converging on a solution (Yao et al., 2024a).

To effectively guide MLLMs across diverse tasks, we adopt and implement seven distinct prompting techniques:

1. Zero-Shot Prompting (Radford et al., 2019)

2. One-Shot Prompting (Mann et al., 2020)

3. Few-Shot Prompting (Mann et al., 2020)

4. Chain-of-Thought (CoT) Prompting (Wei et al., 2022)

5. Analogical Prompting (Yasunaga et al., 2023)

6. Generated Knowledge Prompting (Liu et al., 2021)

    7. Tree-of-Thought (ToT) Prompting (Yao et al., 2024a)

For detailed reviews, prompt templates, and usage scenarios for each prompting technique, please refer to Appendix C. By systematically designing and refining these prompts, our approach aims to generate consistent, rationale-driven outputs across a wide range of multimodal tasks.

## 2.4 Stage 4: Evaluation Framework

Our evaluation framework is designed to rigorously assess model outputs across multiple tasks and prompting techniques. All experiments were performed on high-performance Nvidia GPUs (see Appendix H). To ensure consistency, inference parameters, including temperature, maximum token length, and decoding strategies were held constant across all models. We assessed model outputs along two primary dimensions: task performance and resource consumption. Task performance metrics include accuracy, relevancy, conciseness, and hallucination. Accuracy captures whether the response correctly addresses all components of the task. Relevancy assesses how well the response aligns with the task's context and objectives. Conciseness evaluates the clarity and brevity of the response. Hallucination measures the extent to which responses include irrelevant, redundant, or fabricated content. Resource consumption metrics include inference time and memory usage, recorded to evaluate model efficiency.

To conduct a structured evaluation, two expert annotators independently assessed the model outputs for each task. For each of the four dimensions such as Accuracy, Hallucination, Relevance, and Conciseness, we used well-defined scoring rubrics (see Appendix E, 16). Accuracy was assessed against expected ground truth answers. For the other three aspects, annotators applied the scoring definitions to assign a rating per response. To reduce bias and improve consistency, each annotator reviewed a disjoint set of tasks and subsequently cross-validated a subset of each other's evaluations. This peer-review process ensured agreement on borderline cases and enabled calibration between annotators.

Table 6 summarizes the empirical thresholds established for these metrics. Table 16 supplement these empirical thresholds to ensure consistent assessment of model performance.

Table 6: Empirical thresholds for evaluation metrics based on industry benchmarks and prior research (DeepMind, 2025; Jiang et al., 2025; Huang et al., 2023; Zhang et al., 2024c; Adler et al., 2024; Meta, 2024b). The Accuracy metric determines if all task elements are correctly addressed, while Relevancy ensures that responses remain contextually aligned. Conciseness evaluates clarity and brevity, and Hallucination flags irrelevant or repetitive content.

| Metric | Threshold |
|---|---|
| Accuracy | $\geq 80\%$ |
| Hallucination | $< 5\%$ |
| Relevancy | $\geq 90\%$ |
| Conciseness | $\geq 80\%$ direct, $< 10\%$ under-explained |

### 2.4.1 Inter-Annotator Agreement and Evaluation Consistency

Since manual evaluation has been adopted to assess model responses across multiple dimensions such as correctness, hallucination, relevancy, and conciseness, it is essential to ensure consistency and reliability in the evaluation process. To achieve this, we implemented a structured cross-review methodology, ensuring that the scoring process remained objective and reproducible.

The evaluation was conducted by two independent annotators, each responsible for assessing specific Evaluation Aspects (EAs). To maintain consistency and minimize subjectivity, the following structured approach was adopted.

**Annotator A** evaluated EA1 and EA4, while **Annotator B** evaluated EA2 and EA3. Each annotator assessed the model outputs independently, applying the predefined evaluation criteria outlined in Table 16.

To validate scoring consistency, the annotators cross-reviewed each other's evaluations: Annotator B reviewed EA1 and EA4, originally assessed by Annotator A. Annotator A reviewed EA2 and EA3, originally assessed by Annotator B. The cross-review focused on identifying discrepancies in accuracy, relevancy, conciseness, and hallucination assessments, ensuring that both annotators followed the evaluation framework consistently.

If any discrepancies were identified during the cross-review, they were flagged for discussion between the two annotators. Consensus resolution was prioritized, where both annotators examined the specific response, revisited the criteria, and agreed on a final label. If necessary, criteria definitions were refined to enhance clarity for future evaluations.

A structured guideline document was used by both annotators to ensure uniformity in interpretation. Each evaluation metric such as Accuracy, Relevancy, Conciseness, and Hallucination was defined with clear decision rules to ensure judgments were consistent across different EAs. By implementing this cross-review and consensus resolution process, we ensured a high level of reliability and reproducibility in our evaluation. The application of this methodology across different EAs, along with any refinements made based on annotator discussions.

### 2.4.2 Threshold Selection

We employ a range of metrics to evaluate the performance of AI models, each of which must be quantified to ensure objective assessment. The primary objective of this quantification process is to evaluate various prompt engineering techniques across different models for specific tasks. Our goal is to identify an optimal combination where a given prompt enables the model to generate accurate results with minimal hallucination, high relevance, and concise yet effective explanations. To achieve this, we have established thresholds for each metric, ensuring a structured and rigorous evaluation framework.

The accuracy threshold for the models has been set at 80% or higher, as maintaining a high accuracy level is crucial given their training on extensive datasets. While the standard benchmark for accuracy is typically 75%, we have opted for 80% to ensure improved reliability and effectiveness beyond this baseline. Many industry benchmarks and model evaluation studies support this decision, suggesting that maintaining an accuracy above 80% is desirable for real-world applications (Jiang et al., 2025). For instance, multimodal models evaluated in benchmark tests like MMLU and Visual Question Answering (VQA) often struggle to exceed 75%, highlighting the challenge of ensuring consistency across diverse tasks. However, in high-stakes domains such as autonomous systems, healthcare AI, and industrial automation, achieving at least 80% accuracy is often necessary to minimize errors and improve decision-making reliability (Huang et al., 2023; Zhang et al., 2024c). Additionally, frontier industries leading AI research indicate that state-of-the-art models consistently target higher accuracy rates to enhance generalization across multimodal inputs (Adler et al., 2024; DeepMind, 2025; Meta, 2024b). Therefore, while 75% serves as a common baseline in general AI benchmarks, raising the threshold to 80% reinforces a commitment to higher performance standards and better real-world applicability.

While LLMs and MLLMs have emerged as powerful tools capable of tackling complex problems through reasoning, explanation, summarization, interpretation, and retrieval, they are also prone to generating false or irrelevant information, a phenomenon commonly referred to as hallucination. This study explores the different types of hallucinations that arise from various prompting techniques, which are ideally used to mitigate hallucinations in these advanced models. However, as discussed in depth in this study (Chakraborty et al., 2025), even state-of-the-art prompting mechanisms cannot completely eliminate hallucinations. Given the critical role hallucination plays in model reliability, we have specifically evaluated hallucination as a key component of our study. Recent research confirms that leading LLMs exhibit hallucination rates within a measurable range. For instance, Google Gemini-2.0-Flash-001 has a hallucination rate of 0.7%, OpenAI GPT-4.5 Preview hallucinates 1.2% of the time, and Claude-3.5-Sonnet exhibits a 4.6% hallucination rate, as reported in the Vectara Hallucination Leaderboard (Vectara, 2024) and Galileo's Hallucination Index (AI, 2024). These results align with established research on hallucination detection, including (Laban et al., 2022) and (Honovich et al., 2022), both of which emphasize the importance of strict factual adherence in AI-generated summaries. Based on extensive evaluations, a hallucination threshold of < 5% is a reasonable and empirically supported benchmark for ensuring high factual accuracy in AI systems, particularly

in domains requiring trustworthy and verifiable outputs. Since our evaluation focuses on complex tasks requiring reasoning and accurate outputs, we expect minimal hallucination. This Hallucination threshold/rate exhibit significant degradation in factual grounding, where misinformation could have severe consequences, particularly in long-context reasoning.

Regarding relevancy, the model must provide highly relevant results. For fully relevant outputs, a minimum threshold of 90% has been set, ensuring that the model consistently delivers accurate and meaningful responses. However, the model must not generate irrelevant results under any circumstances, reinforcing the expectation of precise and meaningful responses.

In terms of conciseness, the model is expected to provide explanations that align with the complexity of the tasks. Under-explained responses should be kept below 10%, as these tasks require the model to justify its conclusions effectively. For to-the-point explanations, at least 80% of the model's responses should be direct and concise. However, some room is allowed for over-explained responses, particularly in cases where prompting techniques, such as analogical prompting, necessitate additional elaboration. Manual analysis has shown that in such cases, generating more analogies can be beneficial to comprehension.

This structured approach ensures that the model performs optimally across multiple dimensions, balancing accuracy, hallucination control, relevancy, and conciseness to deliver high-quality, reliable, and meaningful results.

## 3    Results

This section presents a comprehensive evaluation of model performance across multiple evaluation aspects (EAs) using a diverse set of tasks. The evaluation encompasses seven aforementioned prompting techniques and key performance indicators such as Accuracy, Hallucination control, Response Relevance, Irrelevance, and Conciseness. For Conciseness, results are divided into Under-Explained (UE) and a combined measure of Target Precision and Over-Explained (TP + OE).

Models are categorized based on their parameter count as follows:

- Small MLLMs ($< 4$B parameters),

- Medium MLLMs (4B–10B parameters),

- Large MLLMs ($> 10$B parameters).

### 3.1    Summary of Average Model Performance Across Evaluation Aspects

Fig 2–5 report the average performance (in %) for each model category (Small, Medium, and Large MLLMs) under the seven prompting techniques across the four evaluation aspects.

In EA1 (Reasoning and Compositionality) tasks, which assess a model's ability to perform multi-step problem solving and integrate information across modalities, Fig 2 highlights several key findings. Few-shot prompting yields the highest accuracy for large multimodal language models (MLLMs), achieving 45% and outperforming both Chain-of-Thought and Tree-of-Thought prompting strategies. In contrast, small MLLMs exhibit notably higher hallucination rates, reaching up to 75% when using Tree-of-Thought, while medium and large models demonstrate substantially lower hallucination levels. Relevance scores remain consistently high for large MLLMs, with performance exceeding 90%, whereas small models tend to lag in aligning their responses with task context. Finally, in terms of conciseness, small MLLMs are more likely to produce under-explained or verbose responses, in contrast to medium and large models, which maintain clearer and more concise outputs.

In EA2 (Multimodal Understanding and Alignment) tasks, which assess how well the model aligns, integrates, and interprets information across modalities, Fig 3 summarizes the performance on four tasks across three model sizes of MLLMs. Large and medium MLLMs achieve near-perfect relevance, close to 100% when using Zero-Shot, One-Shot, and Few-Shot prompting techniques. In contrast, small MLLMs demonstrate higher

hallucination rates, particularly when using Tree-of-Thought prompting. While accuracy across all model sizes remains moderate, small MLLMs tend to score lower on average. Conciseness metrics further reveal that small models often produce shorter, under-explained responses, whereas medium and large models provide more complete and detailed outputs.

For complex code generation and execution tasks under EA3, the results summarized in Fig 4 indicate that large MLLMs achieve the highest accuracy, reaching up to 96.88% with Few-Shot prompting. Hallucination levels are nearly zero in medium and large MLLMs across several prompting methods, whereas small MLLMs exhibit considerably higher hallucination. Relevance scores remain uniformly high, approaching 100% for medium and large MLLMs. In terms of response quality, small MLLMs tend to produce under-explained outputs, while medium and large models generate more balanced and detailed code responses. Notably, EA3 displays a distinct conciseness pattern compared to other evaluation aspects, with Large MLLMs showing extremely low UE scores (as low as 3.12%) and very high TP+OE scores (often nearly and above 90%). This reflects the nature of code generation tasks, where explicit reasoning and fully elaborated code listings are typically prioritised to ensure correctness and completeness.

In knowledge retrieval and integration tasks (EA4), Fig 5 shows that large MLLMs achieve the highest accuracy, up to 87.5%, along with near-perfect relevance (close to 100%), particularly when using Zero-Shot prompting. In contrast, small MLLMs display higher hallucination rates, exceeding 40% in some cases, while medium MLLMs fall between the two in terms of performance. Additionally, medium MLLMs are more likely to produce under-explained outputs compared to both small and large categories.

Notably, conciseness in EA3 requires a different interpretation than in other EAs. In EA1, EA2, and EA4, excessive explanation may reduce clarity, making concise, to-the-point answers preferable. In contrast, for EA3's code generation tasks, detailed, step-by-step reasoning and fully commented code (reflected in higher TP+OE scores and lower UE) are advantageous because they improve correctness, reproducibility, and debugging. The observed trend in EA3, large models producing more fully explained outputs and fewer under-explained responses; therefore reflects a task-appropriate conciseness preference, even though it diverges from the patterns seen in non-code EAs.

## 3.2 Performance Profiling of MLLMs

To better understand the computational efficiency and generative behavior of the evaluated models, we analysed response time, output length, and memory utilization across all four evaluation aspects. Descriptive statistics, including average (AVG), standard deviation (STD), median (MEDIAN), minimum (MIN), and maximum (MAX) were computed for each metric and are presented in Tables 7 – 14.

The results are discussed in three parts. First, we examine model-specific trends in response time across prompting methods and task types, highlighting differences across small, medium, and large MLLMs. Next, we present evaluation aspect-specific insights, comparing how models behave across reasoning, multimodal understanding, code generation, and knowledge retrieval tasks. Finally, we report the memory footprint of each model, noting significant variation based on architecture and scale.

### 3.2.1 Model-Specific Observations:

For small MLLMs, CoT prompting exhibited stable processing times across tasks, ranging approximately from 5.5 to 6.2 seconds. Among all techniques, Analogical and ToT prompting resulted in the longest processing times, with knowledge retrieval tasks requiring the most time and code generation tasks the least.

Medium MLLMs showed similar trends, where Analogical prompting remained the slowest across tasks. Few-Shot prompting was generally the fastest, although in model understanding and code generation tasks, One-Shot prompting was marginally quicker.

For large MLLMs, Analogical and ToT prompting consistently led to the highest processing times. Few-Shot prompting was typically faster across tasks, except in reasoning and code generation, where the time differences were minimal. CoT prompting was notably stable for large models, with processing durations ranging between 13.5 and 16 seconds.

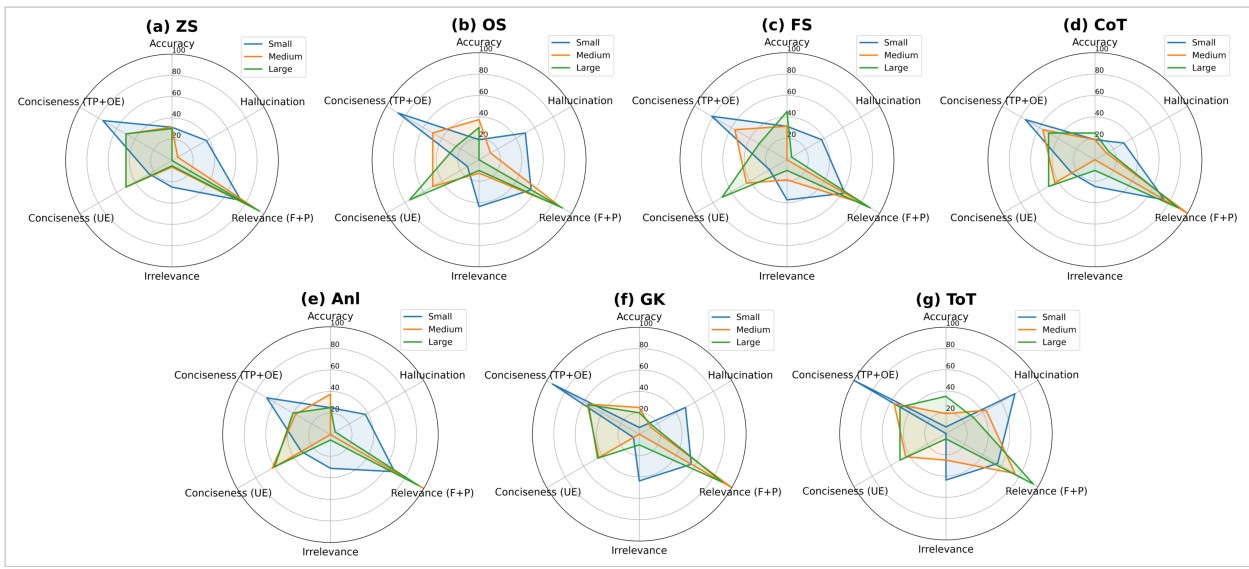

Figure 2: EA1: Reasoning and Compositionality Tasks Results Summary. Radar plots (a–g) present the average performance (in %) of Small (S-MLLMs, < 4B), Medium (M-MLLMs, 4B–10B), and Large (L-MLLMs, > 10B) models across prompting techniques: (a) ZS = Zero-Shot, (b) OS = One-Shot, (c) FS = Few-Shot, (d) CoT = Chain-of-Thought, (e) Anl = Analogical, (f) GK = Generated Knowledge, and (g) ToT = Tree-of-Thought. Performance metrics include Accuracy, Hallucination, Relevance (Fully and Partially Relevant), Irrelevance, and Conciseness (Under-Explained – UE, To the Point – TP, and Over-Explained – OE). These plots illustrate how prompting strategies and model scale shape reasoning behaviour. See Section 3.1 for detailed discussion and Table 17 in the appendix for comprehensive statistics.

Within the Large model group, response times naturally vary based on parameter size, architecture, and runtime configuration. For example, in EA1-T1 (Reasoning and Compositionality) (see Tab1), One-Shot prompting shows relatively modest variation; from 0.48 s for Llama3.2-Vision (11B) to 13.09 s for Ovis-1.6 (10.2B), 1.89 s for OmChat V2 (13B), 8.05 s for Pixtral-12B, and with InternVL2-26B at 10.66 s, only about 22× slower than the fastest model. In contrast, Analogical prompting exhibits the widest spread, ranging from 0.40 s for Llama3.2-Vision to 77.69 s for InternVL2-26B nearly 194× slower with intermediate values for Ovis-1.6 (14.17 s), Pixtral-12B (11.03 s), and OmChat V2 (4.08 s). These differences are expected given the diversity in architecture, parameter count, and quantized/runtime configurations. Crucially, despite absolute latency differences, InternVL2-26B aligns with the group's overall trends in accuracy, hallucination, relevance, and conciseness, supporting its inclusion in the Large category for comparative completeness. This behaviour is consistently observed across all 24 tasks, where similar prompting styles exhibit comparable latency patterns within the Large model group.

### 3.2.2 Evaluation Aspect Specific Obeservations

For EA1 (Reasoning and Compositionality) tasks, Table 7 presents the response time statistics for reasoning tasks, and Table 8 shows the corresponding output lengths. For Small MLLMs, response times are relatively short (AVG ≈ 5.3–6.2 s) and outputs are concise, while Large MLLMs exhibit higher variance in response time (AVG up to 22.97 s) and produce more verbose outputs.

For EA2 (Multimodal Understanding and Alignment) tasks, Tables 9 and 10 provide the response time and output length statistics for tasks requiring multimodal understanding. Small MLLMs respond faster (AVG ≈ 5.91 s with ZS) and generate shorter outputs (AVG ≈ 917 characters with ZS), whereas Large MLLMs require significantly more time (AVG ≈ 24.67 s with ZS) and produce longer responses (AVG ≈ 1417 characters with ZS).

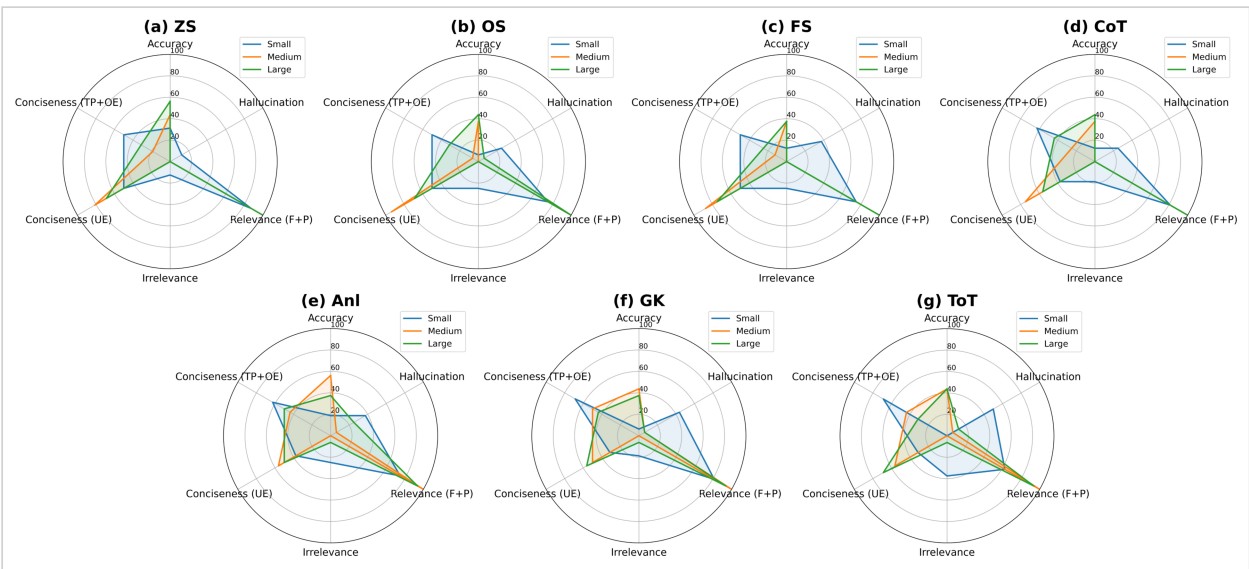

Figure 3: EA2: Multimodal Understanding and Alignment Tasks Results Summary. Radar plots (a–g) present the average performance (in %) of Small (S-MLLMs, < 4B), Medium (M-MLLMs, 4B–10B), and Large (L-MLLMs, > 10B) models across prompting techniques: (a) ZS = Zero-Shot, (b) OS = One-Shot, (c) FS = Few-Shot, (d) CoT = Chain-of-Thought, (e) Anl = Analogical, (f) GK = Generated Knowledge, and (g) ToT = Tree-of-Thought. Performance metrics include Accuracy, Hallucination, Relevance (Fully and Partially Relevant), Irrelevance, and Conciseness (Under-Explained – UE, To the Point – TP, and Over-Explained – OE). These plots illustrate how different prompting strategies and model scales influence multimodal understanding and alignment. See Section 3.1 for detailed discussion and Table 18 in the appendix for comprehensive statistics.

For EA3 (Complex Code Generation and Execution) tasks, Tables 11 and 12 report the response time and output length for code generation tasks. Large MLLMs show the highest accuracy with Few-Shot prompting and require longer response times and output lengths compared to smaller models.

For EA4 (Knowledge Retrieval and Integration) tasks, Tables 13 and 14 detail the response time and output length for knowledge retrieval tasks. In these tasks, Large MLLMs achieve high accuracy with minimal hallucination and produce longer, more detailed responses than Small and Medium models.

### 3.2.3 Model Memory Allocation

Table 15 lists the selected MLLMs along with their categorization (Small, Medium, or Large), model size (in billions), and allocated GPU memory (in GB). For example, the Pixtral 12B model uses 35 GB, while the InternVL2-1B model uses only 0.05 GB. These variations highlight the impact of architecture-specific optimizations in addition to model size.

In summary, our analysis reveals consistent patterns across prompting techniques and model sizes. Analogical prompting typically resulted in the longest response times and the most verbose outputs, followed by Tree-of-Thought (ToT). In contrast, Few-Shot and One-Shot prompting were generally faster and produced more concise outputs. Among task types, code generation was the fastest to process, while response time tended to increase with model size.

## 4 Discussion

Our evaluation reveals significant variations in MLLM performance across task types and prompting techniques. Overall, larger models consistently outperform medium and small models, especially in tasks such as knowledge retrieval and code generation, yet tasks requiring complex reasoning and nuanced understanding

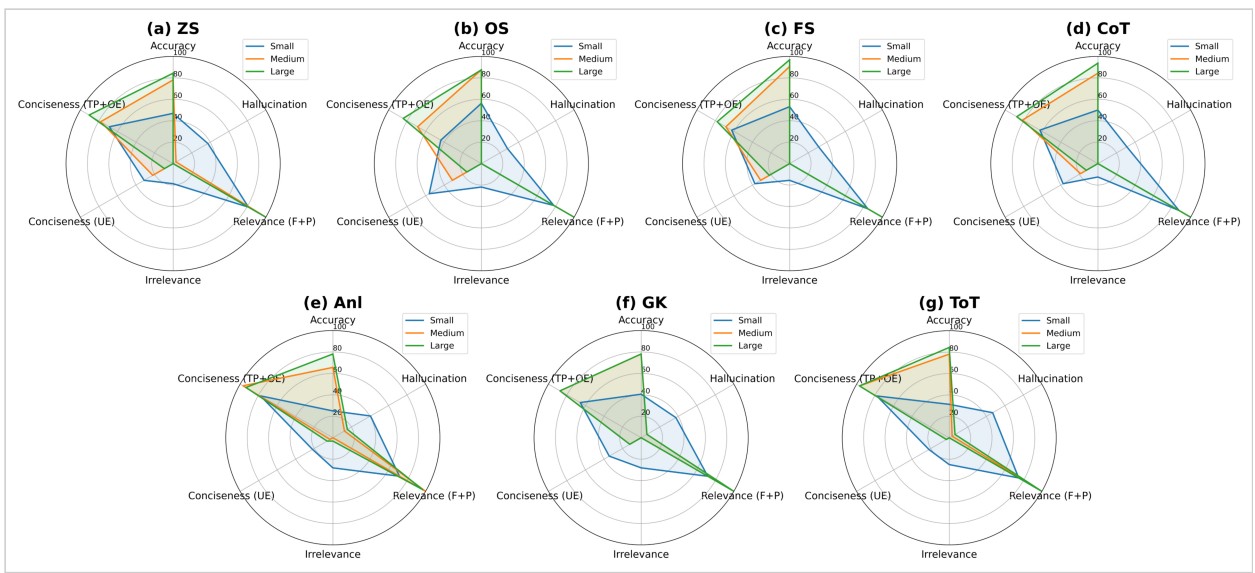

Figure 4: EA3: Complex Code Generation and Execution Tasks Results Summary. Radar plots (a–g) show the average performance (in %) of Small (S-MLLMs, < 4B), Medium (M-MLLMs, 4B–10B), and Large (L-MLLMs, > 10B) models across prompting techniques: (a) ZS = Zero-Shot, (b) OS = One-Shot, (c) FS = Few-Shot, (d) CoT = Chain-of-Thought, (e) Anl = Analogical, (f) GK = Generated Knowledge, and (g) ToT = Tree-of-Thought. Performance metrics include Accuracy, Hallucination, Relevance (Fully and Partially Relevant), Irrelevance, and Conciseness (Under-Explained – UE, To the Point – TP, and Over-Explained – OE). These plots highlight how model scale and prompting strategies affect performance in code generation and execution tasks. See Section 3.1 for detailed discussion and Table 19 in the appendix for comprehensive statistics.

still yield relatively low accuracies (often below 60%). This disparity indicates that while scaling improves certain capabilities, even the largest models struggle with multi-step reasoning and abstract deduction.

For tasks that require multi-step reasoning and compositional problem solving, our results show that providing multiple examples (Few-Shot prompting) enhances accuracy in large models. However, more structured prompting approaches (such as Chain-of-Thought, Analogical, and Tree-of-Thought) tend to increase hallucination rates, particularly in smaller models. This suggests that, although structured prompts are designed to guide logical inference by encouraging intermediate reasoning steps, they can sometimes introduce extraneous or confabulated details that ultimately undermine output quality.

In tasks demanding multimodal understanding and alignment (EA2), large models achieve near-perfect relevance scores when using simpler prompting strategies (e.g., Zero-Shot and One-Shot). This implies that pre-trained multimodal embeddings in these models are highly effective at integrating text and visual inputs.

Zero-Shot prompting emerged as the most effective technique in EA2, achieving the highest accuracy and lowest hallucination rates across model sizes. This suggests that MLLMs may rely heavily on pre-trained multimodal embeddings rather than explicit reasoning across different modalities. In contrast, complex reasoning-based prompts, such as Analogical and Tree-of-Thought, degraded performance, indicating that multimodal models struggle when required to interpret and synthesize abstract relationships between text and images. These results highlight limitations in current MLLMs' spatial and contextual awareness, which are critical for applications such as visual question answering (VQA), AI-generated content moderation, and automated medical image interpretation. While MLLMs can extract information from multimodal inputs, they lack deep semantic alignment; a challenge that must be addressed before deploying these models in high-risk environments. Given these limitations, current open-source MLLMs cannot be relied upon for reasoning-intensive tasks without human oversight or external validation mechanisms.

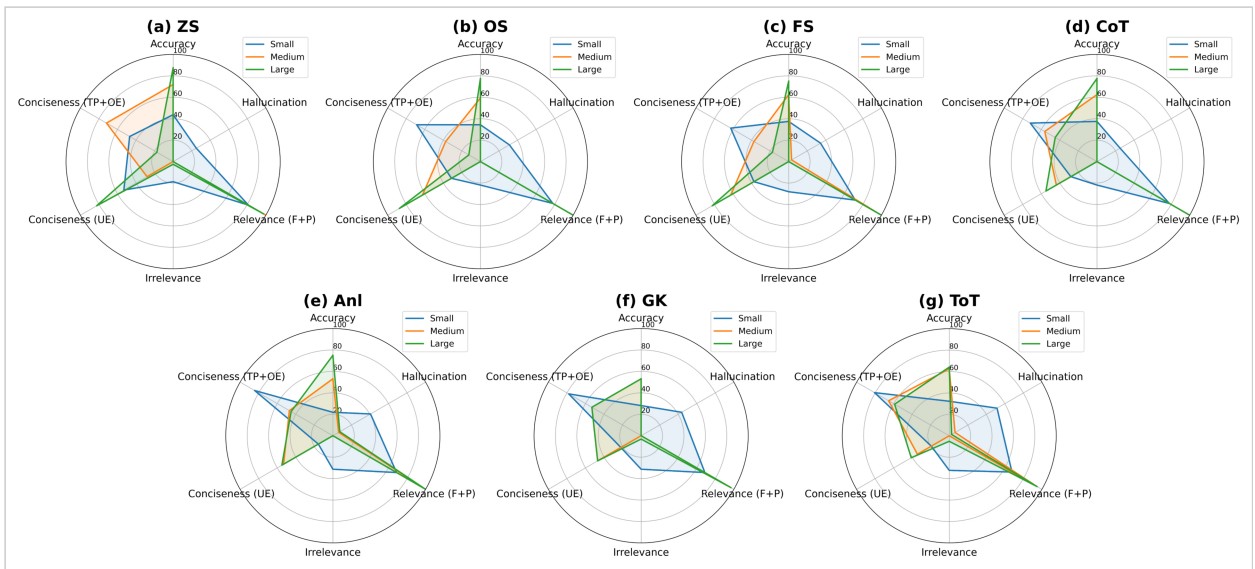

Figure 5: EA4: Knowledge Retrieval and Integration Tasks Results Summary. Radar plots (a–g) present the average performance (in %) of Small (S-MLLMs, < 4B), Medium (M-MLLMs, 4B–10B), and Large (L-MLLMs, > 10B) models across prompting techniques: (a) ZS = Zero-Shot, (b) OS = One-Shot, (c) FS = Few-Shot, (d) CoT = Chain-of-Thought, (e) Anl = Analogical, (f) GK = Generated Knowledge, and (g) ToT = Tree-of-Thought. Performance metrics include Accuracy, Hallucination, Relevance (Fully and Partially Relevant), Irrelevance, and Conciseness (Under-Explained – UE, To the Point – TP, and Over-Explained – OE). These plots demonstrate how different prompting strategies and model sizes influence factual retrieval and integration capabilities. See Section 3.1 for detailed discussion and Table 20 in the appendix for comprehensive statistics.

Code generation tasks exhibited the highest accuracies across all model sizes. The structured nature of programming tasks appears to benefit from Few-Shot prompting, which provides clear examples that guide both syntactic and semantic generation. Nonetheless, hallucination remains an issue, especially in smaller models; which is critical in contexts such as software development and cybersecurity where errors can have severe consequences.

Knowledge retrieval tasks also demonstrate the advantage of scaling: large MLLMs achieve the highest accuracy and relevance, particularly with Zero-Shot prompting. However, these models sometimes present outputs with unwarranted confidence, even when portions of the retrieved information are incorrect. This lack of reliable verification is problematic in domains that demand high factual accuracy, such as legal, medical, and scientific applications.

Hallucination remained a fundamental challenge across all models and prompting strategies, particularly in tasks requiring abstract reasoning. Analogical, General Knowledge, and Tree-of-Thought prompting exhibited the highest hallucination rates, suggesting that the current implementations of structured reasoning within MLLMs remain unreliable. This is especially concerning in safety-critical applications where factual correctness is imperative, such as AI-generated medical reports, legal document drafting, and financial risk assessments.

Our analysis of response times and output lengths further underscores the trade-offs inherent in different prompting techniques. More complex methods like Analogical and Tree-of-Thought prompting require longer processing times and produce more verbose outputs, while One-Shot and Few-Shot prompting yield faster and more concise responses. Although larger models generally incur higher computational costs and longer response times, the improvements in accuracy and relevance, particularly for multimodal understanding and knowledge retrieval often justify these trade-offs.

Table 7: Response Time Statistics for EA1 (Reasoning and Compositionality) Tasks. Descriptive statistics (in seconds: AVG, STD, MEDIAN, MIN, MAX) are provided for each prompting technique for Small, Medium, and Large MLLMs. Abbreviations: ZS = Zero-Shot, OS = One-Shot, FS = Few-Shot, CoT = Chain-of-Thought, Anl = Analogical, GK = Generated Knowledge, ToT = Tree-of-Thought. Refer Section 3.2.2.

| Model Category | Prompt Technique | AVG | STD | MEDIAN | MIN | MAX |
|---|---|---|---|---|---|---|
| Small MLLM | ZS | 6.24 | 5.24 | 5.17 | 0.49 | 14.51 |
| | OS | 5.55 | 4.73 | 4.80 | 0.49 | 13.95 |
| | FS | 5.30 | 4.69 | 4.45 | 0.48 | 14.52 |
| | CoT | 6.51 | 3.91 | 6.15 | 0.87 | 13.91 |
| | Anl | 7.47 | 4.68 | 7.51 | 0.81 | 14.48 |
| | GK | 4.83 | 3.93 | 3.36 | 0.55 | 13.95 |
| | ToT | 9.51 | 6.26 | 10.33 | 0.47 | 19.22 |
| Medium MLLM | ZS | 9.30 | 4.62 | 8.09 | 1.30 | 17.10 |
| | OS | 9.52 | 5.74 | 8.69 | 1.32 | 24.29 |
| | FS | 9.00 | 5.73 | 7.33 | 1.27 | 23.12 |
| | CoT | 10.86 | 5.89 | 8.63 | 4.74 | 25.75 |
| | Anl | 12.35 | 5.87 | 11.80 | 3.34 | 25.38 |
| | GK | 12.02 | 7.39 | 9.49 | 4.00 | 29.60 |
| | ToT | 13.52 | 5.74 | 11.88 | 7.49 | 30.02 |
| Large MLLM | ZS | 22.97 | 26.38 | 12.74 | 1.90 | 101.69 |
| | OS | 19.57 | 25.01 | 10.90 | 0.40 | 101.08 |
| | FS | 21.28 | 26.46 | 10.73 | 0.44 | 101.24 |
| | CoT | 26.07 | 30.67 | 13.70 | 0.48 | 101.71 |
| | Anl | 29.29 | 35.76 | 16.72 | 0.39 | 120.92 |
| | GK | 20.95 | 24.55 | 13.24 | 0.67 | 101.56 |
| | ToT | 26.67 | 26.80 | 16.12 | 0.55 | 101.85 |

In summary, no single prompting method optimally addresses every task. The effectiveness of a prompting strategy is highly dependent on the nature of the task and the model scale. For instance, Few-Shot prompting appears best suited for structured tasks like code generation, while simpler prompting techniques are more effective for multimodal alignment. These findings suggest that hybrid approaches, combining example-based prompts with selective structured reasoning, may offer a promising path toward more reliable and contextually aware multimodal reasoning.

## 4.1 Implications and Use Cases

These findings highlight critical implications for deploying MLLMs in real-world scenarios. While large MLLMs demonstrate strong retrieval and structured output generation, their failure in logical reasoning and multimodal alignment indicates that they are currently unsuitable for fully autonomous decision-making systems in healthcare, finance, or legal domains. Instead, their most effective applications lie in AI-assisted software development, where few-shot prompting can improve code generation workflows while integrating human validation to mitigate errors. Automated knowledge retrieval systems, where large MLLMs can assist in search and summarization tasks but require additional verification mechanisms to ensure factual accuracy. AI-powered tutoring systems, where structured output generation can support educational applications, but deeper logical reasoning capabilities must be further refined. Visual question answering and multimodal content moderation, where large MLLMs can be used to process images and text but require improvements in contextual alignment. Conversely, caution must be exercised when integrating MLLMs into fields where reasoning-based accuracy is paramount. Current models struggle to maintain logical consistency in long-form reasoning tasks, limiting their utility in legal contract analysis, autonomous robotic planning, and financial forecasting.

Table 8: Response Length (Character) Statistics for EA1 (Reasoning and Compositionality) Tasks. This table reports the average output lengths (in characters) along with STD, MEDIAN, MIN, and MAX values for Small, Medium, and Large MLLMs across the prompting techniques. Abbreviations: ZS = Zero-Shot, OS = One-Shot, FS = Few-Shot, CoT = Chain-of-Thought, Anl = Analogical, GK = Generated Knowledge, ToT = Tree-of-Thought. Refer Section 3.2.2.

| Model Category | Prompt Technique | AVG | STD | MEDIAN | MIN | MAX |
|---|---|---|---|---|---|---|
| Small MLLM | ZS | 878 | 819 | 751 | 11 | 2294 |
| | OS | 763 | 656 | 839 | 11 | 1712 |
| | FS | 785 | 753 | 658 | 11 | 2699 |
| | CoT | 960 | 589 | 1069 | 17 | 1869 |
| | Anl | 1147 | 823 | 1157 | 67 | 2800 |
| | GK | 670 | 535 | 607 | 1 | 1946 |
| | ToT | 1475 | 1008 | 1550 | 1 | 2744 |
| Medium MLLM | ZS | 1082 | 596 | 1105 | 82 | 2294 |
| | OS | 1094 | 645 | 873 | 81 | 2174 |
| | FS | 1022 | 677 | 889 | 73 | 2124 |
| | CoT | 1151 | 496 | 1074 | 452 | 2402 |
| | Anl | 1482 | 663 | 1406 | 397 | 3190 |
| | GK | 1339 | 707 | 1193 | 409 | 3343 |
| | ToT | 1645 | 705 | 1573 | 691 | 3845 |
| Large MLLM | ZS | 1417 | 780 | 1313 | 81 | 3111 |
| | OS | 1179 | 630 | 1164 | 86 | 2437 |
| | FS | 1206 | 708 | 1083 | 79 | 2580 |
| | CoT | 1459 | 642 | 1410 | 438 | 2810 |
| | Anl | 1630 | 660 | 1595 | 392 | 2725 |
| | GK | 1435 | 574 | 1439 | 453 | 2457 |
| | ToT | 1768 | 905 | 1602 | 770 | 4432 |

## 4.2 Future Research Directions

The findings of this study highlight several critical areas for future research to enhance the reliability and effectiveness of MLLMs. One promising direction is the development of hybrid prompting strategies, where a combination of few-shot examples and explicit logical structuring could improve reasoning-intensive tasks by guiding models through step-by-step inference. Additionally, memory-augmented models could be explored to enable MLLMs to reference factual information more effectively, reducing the likelihood of hallucinations and improving long-term contextual understanding. Another essential avenue is the advancement of explainability and verification frameworks, particularly for high-stakes applications in legal, medical, and financial domains, where the factual consistency of generated content is crucial.

Furthermore, integrating neurosymbolic AI approaches, which combine deep learning with symbolic reasoning, could enhance logical inference capabilities, especially in tasks requiring structured decision-making. In the context of multimodal alignment, research should focus on improving spatial awareness, cross-modal dependencies, and semantic consistency, ensuring that MLLMs can effectively interpret and synthesize diverse inputs. While this study deliberately scoped to text–image tasks for reproducibility, extending evaluations to temporal modalities such as video remains a natural next step, and we flag this as a priority for future multimodal benchmarking.

Equally important is deeper diagnostic analysis of prompting methods. Although this paper compared seven prompt families, future work should conduct fine-grained ablation studies such as toggling intermediate steps in CoT or analogy scaffolding to isolate which elements contribute most to accuracy or hallucination. This will provide a clearer attribution of risk and reliability across prompting strategies.

Finally, bias and robustness studies remain indispensable, as the failure of reasoning-based prompting techniques suggests that current architectures may not generalize well to unseen logical structures. Investigating dataset biases and refining training methodologies will be critical to ensuring that MLLMs become more reliable, fair, and interpretable across a wider range of real-world applications. Addressing these challenges

Table 9: Response Time Statistics for EA2 (Model Understanding and Alignment) Tasks. This table presents the response time (in seconds) descriptive statistics for each prompting technique for Small, Medium, and Large MLLMs. Abbreviations: ZS = Zero-Shot, OS = One-Shot, FS = Few-Shot, CoT = Chain-of-Thought, Anl = Analogical, GK = Generated Knowledge, ToT = Tree-of-Thought. Refer Section 3.2.2.

| Model Category | Prompt Technique | AVG | STD | MEDIAN | MIN | MAX |
|---|---|---|---|---|---|---|
| Small MLLM | ZS | 5.91 | 4.40 | 4.76 | 0.42 | 15.01 |
| | OS | 3.60 | 2.08 | 3.84 | 0.78 | 7.67 |
| | FS | 3.80 | 2.32 | 3.96 | 0.51 | 8.55 |
| | CoT | 5.90 | 3.80 | 5.64 | 0.61 | 14.24 |
| | Anl | 8.54 | 5.46 | 10.26 | 0.65 | 16.08 |
| | GK | 6.21 | 4.54 | 5.12 | 0.61 | 14.53 |
| | ToT | 6.95 | 4.97 | 6.68 | 0.62 | 18.72 |
| Medium MLLM | ZS | 8.69 | 3.25 | 8.32 | 2.64 | 14.68 |
| | OS | 9.05 | 3.69 | 7.71 | 4.37 | 15.40 |
| | FS | 9.25 | 3.90 | 7.91 | 4.09 | 17.42 |
| | CoT | 11.98 | 3.02 | 12.01 | 7.73 | 19.84 |
| | Anl | 14.05 | 4.89 | 14.94 | 7.96 | 25.33 |
| | GK | 11.29 | 4.98 | 11.24 | 3.72 | 21.71 |
| | ToT | 13.85 | 7.07 | 12.18 | 3.63 | 32.82 |
| Large MLLM | ZS | 24.67 | 27.87 | 15.15 | 0.95 | 102.57 |
| | OS | 19.82 | 21.79 | 13.15 | 0.60 | 77.78 |
| | FS | 19.70 | 24.63 | 14.13 | 0.76 | 113.42 |
| | CoT | 25.06 | 27.28 | 15.86 | 0.87 | 103.77 |
| | Anl | 31.67 | 34.42 | 20.38 | 0.68 | 103.31 |
| | GK | 25.67 | 29.44 | 13.93 | 0.90 | 99.00 |
| | ToT | 28.89 | 29.90 | 19.34 | 0.94 | 119.25 |

Table 10: Response Length (Character) Statistics for EA2 (Model Understanding and Alignment) Tasks. This table lists the output length (in characters) descriptive statistics for Small, Medium, and Large MLLMs. Abbreviations: ZS = Zero-Shot, OS = One-Shot, FS = Few-Shot, CoT = Chain-of-Thought, Anl = Analogical, GK = Generated Knowledge, ToT = Tree-of-Thought. Refer Section 3.2.2.

| Model Category | Prompt Technique | AVG | STD | MEDIAN | MIN | MAX |
|---|---|---|---|---|---|---|
| Small MLLM | ZS | 917 | 751 | 706 | 4 | 2100 |
| | OS | 565 | 419 | 522 | 62 | 1558 |
| | FS | 608 | 492 | 617 | 4 | 1790 |
| | CoT | 894 | 614 | 799 | 20 | 1979 |
| | Anl | 1319 | 841 | 1640 | 11 | 2380 |
| | GK | 1008 | 895 | 620 | 7 | 2612 |
| | ToT | 1087 | 802 | 1113 | 8 | 2553 |
| Medium MLLM | ZS | 1007 | 452 | 967 | 202 | 1844 |
| | OS | 985 | 416 | 931 | 280 | 1857 |
| | FS | 1030 | 444 | 1048 | 406 | 1899 |
| | CoT | 1151 | 496 | 1074 | 452 | 2402 |
| | Anl | 1482 | 663 | 1406 | 397 | 3190 |
| | GK | 1339 | 707 | 1193 | 409 | 3343 |
| | ToT | 1645 | 705 | 1573 | 691 | 3845 |
| Large MLLM | ZS | 1417 | 780 | 1313 | 81 | 3111 |
| | OS | 1179 | 630 | 1164 | 86 | 2437 |
| | FS | 1206 | 708 | 1083 | 79 | 2580 |
| | CoT | 1459 | 642 | 1410 | 438 | 2810 |
| | Anl | 1630 | 660 | 1595 | 392 | 2725 |
| | GK | 1435 | 574 | 1439 | 453 | 2457 |
| | ToT | 1768 | 905 | 1602 | 770 | 4432 |

Table 11: Response Time Statistics for EA3 (Code Generation and Execution) Tasks. Descriptive statistics (in seconds: AVG, STD, MEDIAN, MIN, MAX) for response time across prompting techniques for Small, Medium, and Large MLLMs. Abbreviations: ZS = Zero-Shot, OS = One-Shot, FS = Few-Shot, CoT = Chain-of-Thought, Anl = Analogical, GK = Generated Knowledge, ToT = Tree-of-Thought. Refer Section 3.2.2.

| Model Category | Prompt Technique | AVG | STD | MEDIAN | MIN | MAX |
|---|---|---|---|---|---|---|
| Small MLLM | ZS | 5.35 | 3.95 | 4.97 | 0.62 | 15.46 |
| | OS | 3.47 | 2.69 | 2.98 | 0.47 | 14.68 |
| | FS | 4.64 | 3.50 | 3.82 | 0.65 | 14.63 |
| | CoT | 6.28 | 4.67 | 6.05 | 0.64 | 15.03 |
| | Anl | 7.23 | 4.16 | 7.13 | 0.85 | 14.67 |
| | GK | 5.18 | 3.94 | 4.63 | 0.57 | 14.68 |
| | ToT | 8.50 | 5.64 | 8.28 | 0.63 | 20.91 |
| Medium MLLM | ZS | 6.77 | 3.55 | 6.63 | 1.53 | 17.50 |
| | OS | 4.48 | 1.38 | 4.26 | 2.15 | 7.73 |
| | FS | 4.60 | 1.44 | 4.32 | 1.96 | 7.21 |
| | CoT | 7.40 | 4.70 | 6.43 | 1.58 | 24.34 |
| | Anl | 13.08 | 6.02 | 11.37 | 5.65 | 31.50 |
| | GK | 7.46 | 3.29 | 7.61 | 1.98 | 15.04 |
| | ToT | 12.67 | 6.21 | 11.02 | 3.31 | 30.59 |
| Large MLLM | ZS | 16.48 | 15.63 | 11.41 | 1.90 | 68.60 |
| | OS | 12.63 | 11.82 | 11.37 | 0.31 | 50.79 |
| | FS | 12.02 | 10.90 | 10.82 | 0.37 | 50.76 |
| | CoT | 19.70 | 23.71 | 13.89 | 0.53 | 97.30 |
| | Anl | 34.01 | 38.85 | 20.91 | 0.44 | 126.27 |
| | GK | 17.78 | 20.62 | 13.34 | 0.65 | 88.48 |
| | ToT | 29.46 | 33.48 | 20.44 | 0.62 | 124.55 |

Table 12: Response Length (Character) Statistics for EA3 (Code Generation and Execution) Tasks. This table reports the output length (in characters) descriptive statistics for each prompting technique for Small, Medium, and Large MLLMs. Abbreviations: ZS = Zero-Shot, OS = One-Shot, FS = Few-Shot, CoT = Chain-of-Thought, Anl = Analogical, GK = Generated Knowledge, ToT = Tree-of-Thought. Refer Section 3.2.2.

| Model Category | Prompt Technique | AVG | STD | MEDIAN | MIN | MAX |
|---|---|---|---|---|---|---|
| Small MLLM | ZS | 577 | 417 | 562 | 41 | 1621 |
| | OS | 364 | 340 | 256 | 9 | 1690 |
| | FS | 565 | 494 | 392 | 26 | 2054 |
| | CoT | 736 | 619 | 807 | 10 | 2189 |
| | Anl | 1009 | 592 | 1134 | 61 | 2321 |
| | GK | 637 | 555 | 396 | 4 | 1880 |
| | ToT | 1129 | 772 | 1192 | 2 | 2468 |
| Medium MLLM | ZS | 559 | 345 | 467 | 157 | 1426 |
| | OS | 353 | 177 | 316 | 176 | 1039 |
| | FS | 374 | 201 | 341 | 176 | 899 |
| | CoT | 671 | 461 | 513 | 131 | 1571 |
| | Anl | 1304 | 550 | 1176 | 478 | 2946 |
| | GK | 691 | 365 | 655 | 163 | 1383 |
| | ToT | 1292 | 599 | 1311 | 161 | 2827 |
| Large MLLM | ZS | 995 | 591 | 875 | 112 | 2799 |
| | OS | 980 | 813 | 771 | 170 | 3585 |
| | FS | 780 | 489 | 748 | 112 | 2143 |
| | CoT | 1130 | 654 | 1074 | 142 | 2451 |
| | Anl | 1692 | 672 | 1770 | 485 | 3246 |
| | GK | 1016 | 485 | 1080 | 117 | 1911 |
| | ToT | 1527 | 657 | 1520 | 122 | 2807 |

Table 13: Response Time Statistics for EA4 (Knowledge Retrieval and Integration) Tasks. This table provides descriptive statistics (in seconds) for response time across the seven prompting techniques for Small, Medium, and Large MLLMs. Abbreviations: ZS = Zero-Shot, OS = One-Shot, FS = Few-Shot, CoT = Chain-of-Thought, Anl = Analogical, GK = Generated Knowledge, ToT = Tree-of-Thought. Refer Section 3.2.2.

| Model Category | Prompt Technique | AVG | STD | MEDIAN | MIN | MAX |
|---|---|---|---|---|---|---|
| Small MLLM | ZS | 8.08 | 5.11 | 8.56 | 0.74 | 18.01 |
| | OS | 5.54 | 3.17 | 6.02 | 0.49 | 12.51 |
| | FS | 6.24 | 3.84 | 6.21 | 0.60 | 14.19 |
| | CoT | 6.19 | 3.62 | 6.14 | 0.62 | 13.41 |
| | Anl | 9.97 | 5.59 | 11.71 | 0.86 | 18.20 |
| | GK | 5.99 | 4.01 | 6.37 | 0.65 | 14.59 |
| | ToT | 8.17 | 5.64 | 8.01 | 0.82 | 19.09 |
| Medium MLLM | ZS | 9.95 | 3.19 | 9.54 | 4.72 | 17.92 |
| | OS | 9.31 | 4.29 | 8.00 | 4.31 | 21.60 |
| | FS | 8.42 | 3.70 | 7.44 | 2.81 | 20.88 |
| | CoT | 11.48 | 4.68 | 9.97 | 5.85 | 24.76 |
| | Anl | 14.83 | 6.26 | 12.80 | 4.34 | 26.58 |
| | GK | 12.49 | 6.39 | 10.63 | 0.95 | 29.66 |
| | ToT | 13.44 | 5.80 | 11.99 | 0.76 | 30.51 |
| Large MLLM | ZS | 29.54 | 36.69 | 14.72 | 2.22 | 122.41 |
| | OS | 21.42 | 24.86 | 13.45 | 0.62 | 117.56 |
| | FS | 21.69 | 24.82 | 13.33 | 0.75 | 89.18 |
| | CoT | 27.32 | 32.82 | 15.00 | 0.90 | 122.06 |
| | Anl | 34.61 | 39.30 | 19.70 | 0.78 | 123.03 |
| | GK | 26.50 | 32.38 | 14.46 | 0.95 | 123.01 |
| | ToT | 31.71 | 36.70 | 19.34 | 0.95 | 123.46 |

Table 14: Response Length (Character) Statistics for EA4 (Knowledge Retrieval and Integration) Tasks. This table provides output length (in characters) descriptive statistics for Small, Medium, and Large MLLMs across the seven prompting techniques. Abbreviations: ZS = Zero-Shot, OS = One-Shot, FS = Few-Shot, CoT = Chain-of-Thought, Anl = Analogical, GK = Generated Knowledge, ToT = Tree-of-Thought. Refer Section 3.2.2.

| Model Category | Prompt | AVG | STD | MEDIAN | MIN | MAX |
|---|---|---|---|---|---|---|
| Small MLLM | ZS | 1236 | 780 | 1186 | 39 | 2649 |
| | OS | 859 | 500 | 987 | 9 | 1642 |
| | FS | 984 | 664 | 987 | 10 | 2483 |
| | CoT | 956 | 597 | 980 | 9 | 2371 |
| | Anl | 1572 | 892 | 2018 | 62 | 2509 |
| | GK | 939.5 | 710 | 935 | 7 | 2680 |
| | ToT | 1266 | 890 | 1266 | 29 | 2588 |
| Medium MLLM | ZS | 1161 | 325 | 1203 | 449 | 1725 |
| | OS | 1042 | 389 | 1012 | 453 | 2111 |
| | FS | 974 | 430 | 837 | 212 | 2332 |
| | CoT | 1281 | 398 | 1253 | 673 | 2375 |
| | Anl | 1742 | 488 | 1772 | 676 | 2743 |
| | GK | 1415 | 734 | 1314 | 588 | 4174 |
| | ToT | 1520 | 584 | 1404 | 624 | 3439 |
| Large MLLM | ZS | 1655 | 599 | 1726 | 607 | 3879 |
| | OS | 1436 | 552 | 1321 | 612 | 2719 |
| | FS | 1310 | 508 | 1217 | 613 | 2597 |
| | CoT | 1590 | 663 | 1469 | 448 | 3175 |
| | Anl | 2103 | 1263 | 1855 | 542 | 6551 |
| | GK | 1523 | 691 | 1295 | 549 | 3074 |
| | ToT | 1892 | 940 | 1747 | 579 | 4965 |

Table 15: Model Memory Allocation and Categorization. This table lists individual MLLMs along with their category (Small, Medium, or Large), model size (in billions), and allocated GPU memory (in GB). Refer Section 3.2.3.

| Model Category | Model Name | Model Size (B) | Allocated Memory (GB) |
|---|---|---|---|
| Small MLLM | InternVL2-1B | 1 | 0.05 |
| | Qwen2-vl | 2 | 4.16 |
| | MiniMonkey | 2.2 | 4.49 |
| | Paligemma | 3 | 10.96 |
| Medium MLLM | Phi3.5 | 4 | 7.77 |
| | Llava-one-vision | 8 | 15.05 |
| | Ovis1.5-8B | 8 | 17.33 |
| | Glm-4v | 9 | 25.94 |
| Large MLLM | Ovis1.6 | 10.2 | 19.02 |
| | Llama3.2-vision | 11 | 11.54 |
| | Pixtral | 12 | 35.30 |
| | Omchat | 13 | 24.62 |
| | InternVL2-26B | 26 | 25.27 |

through targeted research will be crucial in advancing MLLMs beyond pattern recognition, enabling them to perform more consistent, factually grounded, and contextually aware reasoning in complex decision-making tasks.

Additionally, exploring adaptive prompting strategies and self-correcting mechanisms could provide a pathway toward enhancing MLLMs' generalizability and reliability across diverse domains. This study is a motivation for advancing AI systems from reactive models to proactive, agentic entities (Russell & Norvig, 2020) capable of sustained, goal-oriented reasoning. As Agentic AI continues to evolve, robust evaluation frameworks such as the one presented in our work will be essential for ensuring that MLLMs are not only technically proficient but also trustworthy, interpretable, and capable of autonomous knowledge synthesis in complex real-world scenarios.

## 5 Conclusion

This study systematically evaluated open-source MLLMs across a diverse scale of model sizes using a structured benchmarking framework to assess their performance across four key evaluation aspects spanning 24 tasks. By employing diverse prompting techniques, including Zero-shot, One-shot, Few-shot, Chain-of-Thought, Analogical reasoning, Generated Knowledge, Tree-of-Thought; which together combine example-based guidance with structured reasoning, the evaluation provided insights into how these models process multimodal inputs and generate outputs aligned with expected task solutions. These strategies were empirically tested and quantitatively validated, with full prompt templates and outputs documented in the supplementary materials. Our findings highlight the varying effectiveness of different prompting strategies in enhancing MLLMs' interpretability, consistency, and reasoning depth. While some models demonstrated strong performance in certain multimodal translation and cross-modal reasoning tasks, challenges persist in areas requiring deeper contextual understanding, abstraction, and nuanced interpretation of complex inputs. The evaluation underscores the necessity for improved prompt engineering methodologies and more robust benchmarks tailored for multimodal AI. Future work will focus on refining the evaluation criteria, expanding the dataset scope, and integrating real-world application scenarios to further stress-test MLLMs. This includes expanding upon this framework by integrating additional modalities such as video comprehension, auditory processing, and multi-turn interactions to build a more comprehensive evaluation paradigm. Ultimately, these efforts aim to inform the design of next-generation MLLMs that are not only technically proficient but also adaptable, context-aware, and aligned with practical, human-centric applications.

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

## Appendix

## A Literature Review on MLLM Evaluation and Rationale for Selected Evaluation Aspects

Evaluating MLLMs is a critical task that determines their capabilities, limitations, and applicability across various domains. Various studies (Li et al., 2024a; Wu et al., 2025; Yu et al., 2023) have employed a wide range of evaluation aspects depending on their focus, whether it be perceptual understanding, compositional reasoning, multimodal alignment, or task-specific problem-solving. Unlike traditional unimodal models, MLLMs require evaluation frameworks that consider their ability to process, integrate, and generate outputs from different modalities, such as text, images, and, in some cases, audio and video. A review of existing studies (Yin et al., 2024b) shows that evaluations are primarily categorized into general multimodal understanding, task-specific assessments, and trustworthiness metrics. These evaluations are particularly important for ensuring that MLLMs can transition from general-purpose reasoning to task-specific applications, making them more reliable for real-world deployments. This section discusses commonly used evaluation aspects in MLLM research and motivated us to select the evaluation aspect those have wider impact in various applications in MLLM models.

Various studies (Cai et al., 2024; Chen et al., 2024b; Li et al., 2024a; Wu et al., 2025; Yu et al., 2023) have approached evaluation through distinct lenses, focusing on foundational multimodal capabilities and task-specific applications. One of the primary evaluation aspects in existing research is perception and object recognition, where models are tested on their ability to identify objects, classify images, and interpret visual attributes. Benchmarks such as VQA v2, COCO, and ImageNet have been widely used to assess how well models recognize and describe elements within an image, ensuring that they correctly map visual features to textual outputs (Fu et al., 2024b). Similarly, scene understanding is another crucial dimension that determines whether models can accurately describe complex environments, capturing spatial relationships and multiple interacting objects, as tested in datasets such as GQA and VizWiz (Yin et al., 2024b).

Another key area of evaluation in existing studies is multimodal reasoning and alignment, which assesses how well MLLMs integrate information across different modalities to derive meaningful conclusions (Wang et al., 2024c). The ability to perform multimodal chain-of-thought (CoT) reasoning has gained increasing attention, as models must generate structured responses by logically sequencing multiple inputs from different modalities(Yin et al., 2024b). Studies (Tong et al., 2024; Zhou et al., 2025) have found that current MLLMs, despite their strong textual reasoning abilities, still struggle to connect visual cues with textual prompts in a coherent manner, limiting their reliability in domains requiring step-by-step logical processing. Additionally, multimodal reasoning is closely tied to vision-language alignment, which evaluates how effectively models fuse textual and visual data to generate accurate and context-aware outputs. Research (Fu et al., 2024b) indicates that poor alignment often leads to failures in high-level reasoning tasks, as models may incorrectly interpret visual information when translating it into text-based responses.

Beyond fundamental perception and reasoning, some studies have investigated task-specific and real-world applications of MLLMs (Li et al., 2024b; Wang et al., 2024a). One of the most prominent applications is medical image analysis, where models are tested on their ability to assist in diagnosing conditions from X-rays, MRIs, and other medical scans (Zhang et al., 2024b; Royer et al., 2024; Li et al., 2023). In this domain, specialized benchmarks such as RadBench and PMC-VQA have been used to evaluate the accuracy of MLLM-generated medical insights (Yin et al., 2024b). Another critical application is autonomous driving and environmental scene interpretation (Shi et al., 2023; Sima et al., 2024), where models process real-world driving scenarios by integrating road signs, obstacles, and pedestrian movements into decision-making processes. Datasets such as NuScenes-QA (Qian et al., 2024) and BDD-X (Kim et al., 2018) serve as standard benchmarks in this space, ensuring that models can reliably analyze traffic conditions and provide accurate assessments of dynamic environments (Yin et al., 2024b).

Some studies have explored multi-round QA and instruction following (Fu et al., 2024b), where models are tested on how well they can retain contextual information across conversational turns. Research indicates that existing MLLMs often exhibit context drift, failing to maintain coherence when responding to sequential queries, which is a key challenge in developing effective AI-powered assistants (Yin et al., 2024b). Further-

more, aspects related to bias and fairness have also been examined in some studies (Zhang et al., 2024d; Li et al., 2024c), as concerns grow about MLLMs inheriting and amplifying societal biases from their training data. Studies (Adler et al., 2024; Anthropic, 2024) stand out as the most ethically aligned models, showing high accuracy and a strong ability to refuse ethically questionable prompts. Fairness evaluation often involves testing for gender, racial, and regional biases, using datasets such as Multi-Trust (Zhang et al., 2024d) and VLBiasBench (Wang et al., 2024b) to assess whether models generate discriminatory or skewed responses (Yin et al., 2024b).

Another critical aspect in evaluating MLLMs is their robustness and reliability (Dang et al., 2024), particularly in minimizing hallucinations (Bai et al., 2024), a phenomenon where models generate factually incorrect or fabricated outputs. Studies assessing hallucination rates have shown that MLLMs, particularly in complex reasoning tasks, tend to produce confident but incorrect statements, making them unreliable for high-stakes applications such as medical diagnosis and financial forecasting (Bai et al., 2024). Efforts to measure and mitigate hallucinations have led to the development of evaluation frameworks that focus on trustworthiness and factual consistency, ensuring that models generate responses that are not only contextually relevant but also grounded in accurate information (Yin et al., 2024c).

Overall, the breadth of evaluation aspects considered in MLLM studies reflects the increasing complexity of these models and the growing need for rigorous assessment frameworks. While some studies focus on fundamental perception and reasoning abilities, others prioritize task-specific applications, conversational capabilities, and fairness concerns. However, despite the diverse range of evaluation methodologies, challenges remain in developing standardized benchmarks that comprehensively measure MLLM performance across all modalities and tasks. This highlights the importance of selecting well-defined evaluation aspects that balance theoretical rigor and real-world applicability, which is a key motivation behind the selection of four core evaluation aspects in our study.

Despite these advancements, researchers still face challenges in defining standardized, reproducible evaluation aspects. Many existing methods rely on task-specific datasets or closed-set evaluations, limiting the generalizability of their findings. There is an increasing need for comprehensive, multi-domain evaluation frameworks that can assess models across perception, reasoning, task execution, and ethical considerations, ensuring that MLLMs are robust, interpretable, and trustworthy for real-world applications (Fu et al., 2024b).

While audio-visual evaluations are critical for advancing MLLMs, current models still exhibit substantial limitations in handling high-dimensional temporal data, making standardized evaluations difficult(Fu et al., 2024b). Audio and video-based evaluation are not considered in our study, driven by the need for a controlled evaluation framework that minimizes confounding factors related to temporal dependencies, data preprocessing, and model specialization (Yin et al., 2024b).

Given the wide range of evaluation aspects discussed in existing research, our study selects four core aspects 2.1: Reasoning and Compositionality, Multimodal Understanding and Alignment, Complex Code Generation and Execution, and Knowledge Retrieval and Integration, due to their direct relevance in assessing task-specific performance and a wide range of real-world applicability and use-cases of MLLMs.

# B   Detailed Review of Key MLLMs

This section provides a comprehensive review of recent advancements in Multimodal Large Language Models (MLLMs), covering both proprietary and open-source developments. The discussion highlights their architectural designs, multimodal processing capabilities, and evaluation suitability, which informed our selection of models for this study.

## B.1   Proprietary MLLMs

Proprietary models such as OpenAI's GPT-4o and GPT-4.5 Preview (OpenAI, 2025), Anthropic's Claude 3 (including variants like Claude 3 Opus, Sonnet, and Haiku) (Anthropic, 2024), and Google's Gemini series (DeepMind, 2025) have set high standards in multimodal reasoning. For example, GPT-4 introduced multimodal inputs that enable the processing of both text and images, thereby improving instruction-following

and contextual understanding. Similarly, the Claude 3 series is optimized for text–image reasoning tasks, leveraging large-scale datasets and fine-tuned architectures to achieve superior generalization and task flexibility. However, the closed-source nature of these models limits customization, transparency, and independent research adaptability.

## B.2 Open-Source MLLMs

In response to the limitations of proprietary systems, the open-source ecosystem has rapidly evolved, providing models with enhanced accessibility and community-driven improvements. Several high-performance models have emerged:

**ChatRex (Jiang et al., 2024a)** ChatRex is designed as a decoupled MLLM that bridges perception and understanding in vision-language tasks. By employing a retrieval-based approach with a Universal Proposal Network (UPN), it improves visual grounding for tasks like object detection and region-based question answering. While it excels in object-level perception and spatial awareness, its specialization limits its adaptability for abstract multimodal reasoning tasks.

**VITA (Fu et al., 2024a)** VITA is a multimodal interactive model that handles video, image, text, and audio modalities in real time. It extends Mixtral 8×7B to enhance bilingual capabilities and integrates specialized encoders for vision and audio. Its duplex interaction pipeline enables continuous environmental monitoring and multi-turn conversations, though it is less optimized for structured vision-language reasoning such as code synthesis.

**Long-VITA (Shen et al., 2025)** Long-VITA targets long-context visual-language understanding, capable of processing up to 1 million tokens. It uses a multi-stage training process that includes long-sequence fine-tuning to maintain coherence over extended interactions. While it excels in video comprehension and document-level understanding, its lack of strict data filtering can impact response consistency.

**Meteor (Lee et al., 2025)** Meteor employs a rationale traversal approach to enhance multimodal reasoning without significantly increasing model size. By leveraging multifaceted rationale embeddings, it improves vision-language understanding and step-by-step reasoning. However, its performance is sensitive to the availability of rationale-enhanced pretraining datasets and may struggle with tasks that require high-resolution image processing.

**MoAI (Lee et al., 2024)** MoAI utilizes a Mixture of Experts (MoE) approach by integrating external computer vision models for panoptic segmentation, object detection, and OCR. Its two-stage pipeline, comprising a MoAI-Compressor and MoAI-Mixer allows for detailed object detection and structured text reasoning. While effective in high-precision tasks, the dependency on external modules increases computational overhead.

**ViP-LLaVA (Cai et al., 2024)** ViP-LLaVA enhances visual reasoning by overlaying visual markers (e.g., bounding boxes, arrows) directly onto input images, facilitating region-specific interactions. Built on a Vicuna v1.5 language backbone and CLIP-336px vision encoder, it excels in object localization and context-aware reasoning. Its training process involves multiple stages, including BLIP-captioned image-text pairs and GPT-4V instruction data.

**Falcon2-11B (Malartic et al., 2024)** Falcon2-11B is an efficient foundation model with 11 billion parameters, optimized for text-based tasks and extended to multimodal functions via a CLIP-based vision encoder. Although it exhibits strong long-context reasoning, its vision-language variant may be less competitive in tasks requiring fine-grained visual comprehension.

**Cambrian-1 (Tong et al., 2025)** Cambrian-1 is designed as a vision-centric MLLM using multiple vision encoders (e.g., OpenAI CLIP ViT-L/14@336, SigLIP ViT-SO400M, DINOv2 ViT-L/14) integrated

via a Spatial Vision Aggregator (SVA). Available in various parameter scales (8B, 13B, 34B), it excels in OCR and chart-based tasks while maintaining efficiency through reduced visual token usage.

**MiniCPM (Yao et al., 2024b)** MiniCPM is a lightweight model focused on efficient processing of vision-language tasks. Despite its compact architecture, it performs competitively on fundamental multimodal reasoning tasks, though its capacity for complex, long-form content is limited.

**mPLUG-Owl3 (Ye et al., 2024)** mPLUG-Owl3 specializes in long image-sequence understanding and video-based multimodal processing using Hyper Attention Transformer Blocks (HATB). It is tailored for tasks requiring temporal coherence but is less suited for structured code generation and fine-tuned knowledge retrieval.

**Molmo (Deitke et al., 2024)** Molmo emphasizes transparency by being trained on the open PixMo dataset. It excels in fine-grained vision-language understanding and visual grounding tasks, though it is not optimized for structured tasks such as code execution or prompt engineering.

**Additional Open-Source Models** Other notable models include InternVL2-1B (Chen et al., 2024b), Qwen2-VL-2B-Instruct (Yang et al., 2024a), MiniMonkey (Huang et al., 2024), Paligemma-3B-mix-448 (Beyer et al., 2024), Phi-3.5 VLM (Abdin et al., 2024), LLaVA OneVision-7B (Li et al., 2024a), Ovis 1.5-Llama 3-8B (Lu et al., 2024), GLM-4v-9B (GLM et al., 2024), Ovis1.6 (Lu et al., 2024), Llama3.2-Vision (Meta, 2024b), Pixtral (Agrawal et al., 2024), OmChat-V2 (OmLab, 2024), and InternVL2-26B (Chen et al., 2024a; InternVL, 2024). Each of these models is characterized by distinct combinations of language and vision encoders (e.g., Qwen, Gemma, Llama, GLM; ViT, SigLIP, CLIP, EVA) and varying parameter counts, offering a wide spectrum of capabilities from lightweight, efficient inference to high-precision multimodal reasoning. This comprehensive review highlights the diverse strategies in multimodal model design and evaluation, which underpin the criteria for our model selection.

## C    Detailed Review of Prompt Engineering Methods

This appendix provides an in-depth review of the seven prompting techniques employed in this study. For each method, we discuss the underlying principles, key features, and provide example templates along with usage scenarios in both unimodal and multimodal contexts.

### C.1    Zero-Shot Prompting

Zero-shot prompting involves providing the model with only the task description, relying entirely on its pre-trained knowledge (Radford et al., 2019). This method simplifies prompt design and reduces computational overhead, making it ideal for general-purpose tasks.

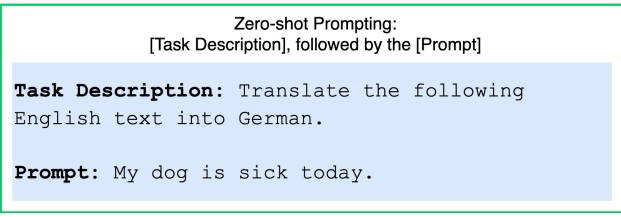

Figure 6: Zero-shot Prompting Syntax

### C.2    One-Shot Prompting

One-shot prompting includes a single example alongside the task description to direct the model toward the desired output (Mann et al., 2020). This method provides minimal contextual guidance, balancing efficiency and accuracy for moderately complex tasks.

Figure 7: One-shot Prompting Syntax

## C.3  Few-Shot Prompting

Few-shot prompting incorporates multiple examples to establish clear input-output patterns, which is particularly useful for tasks that require structured responses (Mann et al., 2020).

Figure 8: Few-shot Prompting Syntax

## C.4  Chain-of-Thought (CoT) Prompting

Chain of Thought prompting encourages models to decompose problems into intermediate reasoning steps, thereby improving logical progression and accuracy in complex tasks (Wei et al., 2022; Kojima et al., 2022; Zhang et al., 2022).

Figure 9: Chain-of-Thought (CoT) Prompting Syntax

## C.5 Analogical Prompting

Analogical prompting utilizes analogous examples that closely align with the task's requirements to foster indirect reasoning and creativity. This method enables models to transfer knowledge based on structural similarities between scenarios (Yasunaga et al., 2023; Lu et al., 2021; Wu et al., 2022; Guo et al., 2024).

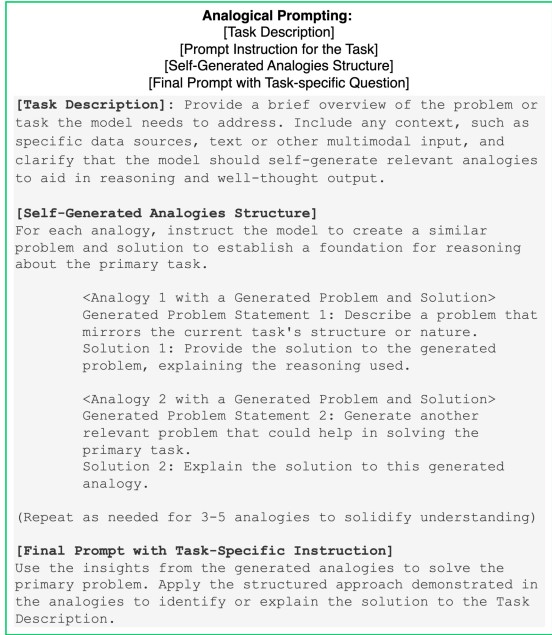

Figure 10: Analogical Prompting Syntax

## C.6 Generated Knowledge Prompting

Generated Knowledge Prompting involves prompting the model to generate additional task-relevant background knowledge, which is then used to improve reasoning and decision-making (Liu et al., 2021; 2023). This technique enriches the input context, leading to improved output accuracy.

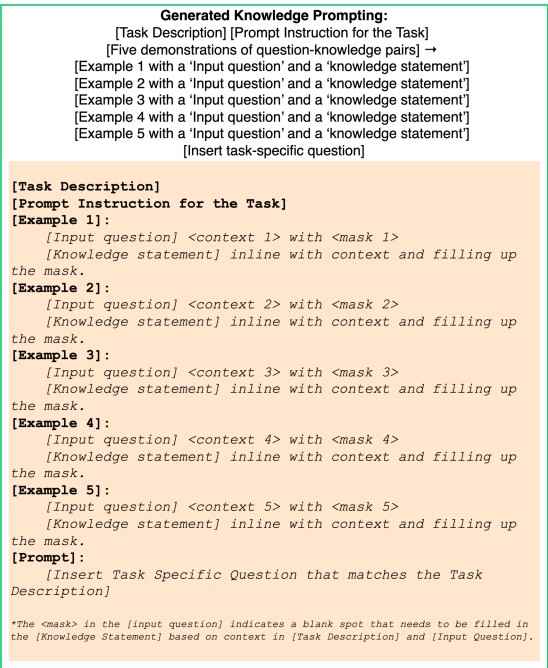

Figure 11: Generated Knowledge Prompting Syntax

## C.7 Tree-of-Thought (ToT) Prompting

Tree of Thought Prompting extends the Chain of Thought framework by organizing reasoning into a decision tree. This structure allows the model to explore multiple reasoning paths before converging on a final solution (Yao et al., 2024a), making it particularly effective for exploratory and decision-making tasks.

```
Tree of Thought (ToT) Prompting:
[Task Description]
[Prompt Instruction for the Task]
[Example 1 with thought steps and evaluation]
[Example 2 with thought steps and evaluation]
[Example n with thought steps and evaluation]
[Final Prompt with task-specific question]

[Task Description]: Describe the problem or task that the
model needs to solve through deliberate reasoning and multiple
thought steps.

[Prompt Instruction for the Task]:

  1. Generate multiple thoughts (ideas, intermediate steps)
     based on the problem.
  2. Evaluate each thought for its contribution to solving the
     task.
  3. Select the most promising thought and proceed to the next
     thought step.

[Example 1 with <Thought steps> and <Evaluation>]:
[Problem Statement]: Provide the problem or task that needs to
be solved.
<Thought 1>: Propose a potential thought or reasoning step.
<Evaluation 1>: Evaluate whether the thought is promising,
ambiguous, or leads to a dead-end.
<Thought 2>: Propose a subsequent thought or reasoning step
based on Thought 1.
<Evaluation 2>: Evaluate this thought step in the context of
the overall problem.
.
.
Repeat for 3-5 Thought steps, as necessary.

[Example 2 with <Thought steps> and <Evaluation>]:
[Problem Statement]: Provide another problem similar to the
one that needs solving.
<Thought 1>:
<Evaluation 1>:
<Thought 2>:
<Evaluation 2>:
.
.
Repeat for 3-5 Thought steps, as necessary.

[Final Prompt with task-specific question]:
Insert a specific question where the model is asked to apply
Tree of Thought prompting to generate and evaluate thoughts,
making sure the final output is a well-reasoned solution.
```

Figure 12: Tree-of-Thought Prompting Syntax

# D  Task Design Methodology for Evaluation Aspects

This appendix section provides a detailed account of the task design methodology for each of the four Evaluation Aspects (EA1-EA4). Each subsection outlines the specific objectives, rationale, and design principles used to create the tasks, with a focus on probing different multimodal capabilities of MLLMs. EA1 focuses on reasoning and compositionality, EA2 on multimodal understanding and alignment, EA3 on code generation from visual inputs, and EA4 on integrating contextual knowledge with multimodal cues. Together, these subsections offer a comprehensive view of how the tasks were systematically designed to simulate real-world multimodal scenarios and evaluate model performance across diverse cognitive and functional dimensions.

## D.1  EA1 Task Design Methodology

The tasks in the study were systematically crafted to evaluate the reasoning and compositionality abilities of MLLMs. Each task was designed with a specific objective and level of complexity, and the accompanying

visuals were tailored to challenge the models' multimodal reasoning capabilities. Below, we outline the thought process behind designing the tasks in Evaluation Aspect 1 (Reasoning and Compositionality) and their corresponding visual aids.

The images for each task were crafted to align with the task objectives while enhancing multimodal reasoning. They provide sufficient context for reasoning without overloading unnecessary details. Every visual element (such as shapes, colors, or character actions) directly contributes to the reasoning challenge. Additionally, the visual style across tasks maintains a professional yet engaging appearance, aiding comprehension and focus. The design of these tasks reflects a deliberate effort to test different dimensions of reasoning and compositionality in MLLMs. By combining well-structured multimodal inputs with incrementally complex challenges, the study ensures that the evaluation methodology is robust, comprehensive, and replicable.

The design of these tasks reflects a deliberate effort to test different dimensions of reasoning and compositionality in MLLMs. By combining well-structured multimodal inputs with incrementally complex challenges, the study ensures that the evaluation methodology is robust, comprehensive, and replicable.

For Task 1 (Pattern Recognition in Visual Sequences), our objective was to assess the model's ability to identify and extrapolate logical patterns across multiple modalities including numbers, shapes, and colors. The task integrates visual and textual elements, with images depicting sequences of numbers, geometric shapes, and colors. The pattern increases in complexity by combining arithmetic progression, geometric reasoning, and ambiguous color sequences. The sequences provide sufficient data points for models to deduce the logical rule while introducing ambiguity in color patterns to test reasoning flexibility. A carefully designed image illustrates the sequence, ensuring clarity and challenge. In Task 2 (Logical Deduction from Text and Simplified Diagram), we aimed to evaluate deductive reasoning by combining textual descriptions and a simplified diagram of a real-world scenario. The task requires the model to synthesize text and visual clues (such as a broken window and a football near the window) to deduce the most likely event. A relatable scenario involving siblings and hobbies was chosen to simulate real-life reasoning challenges. Conflicting clues were introduced to evaluate the model's ability to prioritize relevant evidence. A clear, engaging image illustrates the suspects and the scene, making the task visually intuitive. Task 3 (Mathematical Puzzle with Visual Data) was designed to test numerical reasoning and trend identification by combining tabular and graphical representations of data. Sales figures across four quarters provide a structured data source, while a bar chart complements the table, allowing for cross-referencing between modalities. The task requires models to calculate totals, identify trends, and predict future performance based on observed data. The bar chart and table are designed for easy interpretation while presenting enough complexity to challenge reasoning skills. Finally, Task 4 (Story Synthesis from Text and Image) evaluates narrative synthesis by requiring models to generate coherent continuations from textual and visual inputs. The task integrates a textual scenario about a science fair and an image of a student with a renewable energy project to provide rich input. The visual representation of characters and their environment helps models establish a narrative context. The task encourages the model to infer plausible events and outcomes based on the given details. The image highlights key elements of the narrative, ensuring alignment with the text.

## D.2 EA2 Task Design Methodology

The tasks in EA2 (Multimodal Understanding and Alignment) were developed to assess how well MLLMs integrate and align information from diverse modalities, such as images, text, and charts. Each task was designed to evaluate a distinct dimension of multimodal reasoning, such as matching, inference, translation, and consistency detection. The design approach for EA2 tasks involves real-world multimodal challenges inspired by scenarios where visual and textual data coexist, such as articles with accompanying images and graphs with interpretations. Tasks progress from simpler matching exercises to more abstract reasoning and detailed cross-modal verification, incorporating incremental complexity. Each task requires the model to go beyond basic alignment by explaining its reasoning, testing both understanding and coherence. The images for EA2 tasks were carefully created to enhance multimodal alignment, providing clear and meaningful data and ensuring that every detail contributes to the task's objective. While maintaining visual richness, the images avoid unnecessary complexity, allowing the focus to remain on reasoning and interpretation. The visuals simulate realistic situations, such as kayaking in nature, abstract art, and unemployment graphs, to

make tasks relatable and engaging. Each visual is paired with textual or graphical content in a way that highlights dependencies between modalities.

The objective of Task 1 (Image-Text Matching and Explanation) is to evaluate the model's ability to match visual scenes with corresponding textual descriptions and provide reasoning for each match. The task integrates visual and textual modalities, requiring the model to align content based on key characteristics such as kayaking, manufacturing, and cooking. The model must discern details in each image and accurately match them to abstract textual concepts, testing interpretive and alignment skills. By providing explanations for each match, the task ensures the model demonstrates understanding rather than guessing. Clear and detailed images are used to depict diverse scenes (adventure, industry, and culinary arts), making the task visually intuitive while challenging. Explanation prompts ensure the reasoning process is transparent and logical. Task 2 (Inferring Context from Combined Modalities) aims to assess the model's ability to infer additional context by synthesizing textual and visual information. The task combines a descriptive paragraph (e.g., Maria walking on an empty street at dusk) with an image (a clock tower in a quiet city square) to test the model's ability to infer time, setting, and motivations. The model must analyze textual hints (e.g., streetlights and vendors closing) and visual cues (e.g., clock showing 7:30 PM) to arrive at accurate conclusions. The task introduces subtle contextual clues, requiring the model to resolve ambiguity using logical inference. The image is designed to evoke a specific time and atmosphere, enhancing the model's ability to integrate visual and textual data. The objective of Task 3 (Cross-Modal Translation) is to evaluate the model's ability to interpret abstract visual art and translate it into coherent literary themes or narratives. The abstract painting (depicting turbulent seas under sunlight) challenges the model to interpret artistic elements (colors, patterns, and mood) and map them to poetic themes. The task emphasizes translating visual impressions into descriptive language, requiring creative reasoning. By interpreting the painting's mood (e.g., chaotic yet hopeful), the model's ability to infer emotional tones from visuals is tested. The abstract painting is detailed and expressive, providing ample cues for interpretation while leaving room for subjective reasoning. Task 4 (Aligning Data from Charts and Text) tests the model's capability to detect inconsistencies between data presented in a visual chart and textual descriptions. The task involves analyzing a line graph (e.g., unemployment rates) and cross-referencing it with a textual description to identify mismatches. It evaluates the model's ability to spot errors or inconsistencies (e.g., the text claims a decrease to 2% while the chart shows an increase to 7% in 2020). The task requires detailed attention to both visual and textual details, ensuring the model's output is precise and evidence-backed. A simple, clean line chart is used to focus on key data points while minimizing distractions.

### D.3 EA3 Task Design Methodology:

The tasks in EA3 are designed to evaluate the ability of MLLMs to generate accurate and functional code from multimodal inputs. Each task assesses the model's capacity to interpret visual instructions, generate context-specific code, and execute logical steps. These include interpreting tables, flowcharts, and images that describe problems or provide structured data inputs. The model is expected to produce code that performs well-defined operations such as data visualization, arithmetic computation, or sequence generation, transforming abstract prompts into executable logic. The tasks are carefully crafted to reflect a progression in complexity, ranging from basic operations, such as extracting numerical data from an image to more intricate challenges like generating code from flowchart-based instructions. This incremental design facilitates the evaluation of a model's ability to reason and scale its performance in increasingly demanding coding scenarios.

To support this, the visual components of EA3 tasks were created with clarity, relevance, and realism in mind. Images were designed to be well-organized and unambiguous, directly tied to the task objectives without introducing extraneous elements. They simulate realistic programming scenarios, including the interpretation of structured data or the automation of repetitive tasks. As complexity increases across tasks, so too does the visual intricacy, providing a robust benchmark for assessing the model's capacity for abstraction and stepwise code generation.

Task 1's objective is to test the model's ability to interpret tabular data in an image and generate Python code to visualize it as a bar chart. The task involves extracting structured data (e.g., sales over four quarters) from an image. The model must generate correct visualization code, including libraries (e.g., Matplotlib).

By providing a structured table, the task ensures a clear yet challenging input for code generation. The task simulates a common data science workflow, aligning with practical applications. The key goal of Task 2 is to assess the model's ability to generate Python code (e.g., Turtle graphics) to draw a shape depicted in an image. The task requires interpreting an image of a geometric shape (e.g., a star or hexagon). The model must write accurate code that replicates the shape using a specific library. The task evaluates the model's attention to dimensions and proportions in the visual input. The task mirrors scenarios in educational coding exercises, emphasizing beginner-friendly challenges. Similarly, Task 3 focuses to test the model's ability to extract numerical information from textual input in an image and compute a sum. The task requires extracting numbers from an image (e.g., a shopping receipt). The model must compute a sum based on the parsed input, testing logical reasoning. The task combines OCR (optical character recognition) capabilities with mathematical operations. The scenario mimics practical use cases, such as expense calculation from invoices. Task 4 is designed to evaluate the model's ability to convert chart data into a structured Python dictionary. The task requires extracting information from a bar chart or pie chart. The model must organize extracted data into a Python dictionary with key-value pairs. The task tests the model's ability to ensure the structured output matches the visual input. This mirrors tasks in data engineering or ETL (Extract, Transform, Load) workflows. Task 5 tests the models' ability to parse text in an image (e.g., itemized receipt) and perform basic arithmetic. The task involves identifying item prices in an image and summing them up. Models must handle inconsistencies or unclear inputs in the image (e.g., smudged text). The task simulates expense tracking and financial analysis workflows. Task 6 assess the models' ability to interpret a CSV-like structure in an image and convert it into a Python-compatible format. The task requires parsing tabular data with headers and rows. The model must generate Python code to represent the CSV data as a list of dictionaries or Pandas DataFrame. Combines OCR capabilities with data engineering skills. This task reflects real-world scenarios in data preprocessing. Task7 evaluate the models' ability to interpret step-by-step algorithm instructions in an image and generate functional code. The task requires the model to translate a flowchart or textual steps into Python code. The model must follow a structured logical process (e.g., loops, recursion) to compute the sequence. This task emphasizes algorithm design and educational coding scenarios. Task 8 is designed to test the model's ability to follow decision-making logic in a flowchart and implement it as a Python function. The task requires interpreting decision nodes and paths in the flowchart. The model must translate the flowchart logic into executable code. The task evaluates whether the model can adhere to predefined logical structures. Flowchart-based programming tasks are widely used in both educational and industrial contexts.

### D.4 EA4 Task Design Methodology

The EA4 task set simulates a wide range of real-world challenges that demand the integration of multimodal inputs with contextual knowledge. Designed to rigorously test MLLMs' ability to reason, retrieve, and synthesize across domains, these tasks combine clear visual inputs with meaningful textual prompts to evaluate model performance in complex knowledge-driven scenarios. Each task in EA4 requires the model to retrieve relevant contextual knowledge, either from external sources or embedded within the input and make sense of the given scenario. This involves the ability to combine textual and visual content, such as maps, historical images, scientific charts, or cultural references, and generate informed outputs. The tasks are crafted to reflect authentic applications in domains such as history, science, and fact-checking, often requiring the model to resolve ambiguity by reasoning through incomplete or conflicting information.

To support this, the visuals accompanying EA4 tasks are carefully designed for contextual relevance. Each image aligns closely with the task narrative, whether depicting a historical landmark, cultural artifact, or scientific figure and provides just enough detail to encourage reasoning without introducing unnecessary visual complexity. These visuals are grounded in real-world scenarios, helping simulate tasks such as identifying locations, analyzing diagrams, or verifying facts. Moreover, they are deliberately constructed to complement the associated textual inputs, promoting effective multimodal interaction and integrated reasoning across modalities.

Task 1 evaluates the models' ability to identify a historical monument and explain its significance using external knowledge. The task requires the model to identify a landmark (e.g., a clock tower) from its image; where the model must retrieve relevant historical or cultural information about the landmark. The task

tests the ability to synthesize visual data and external knowledge into a coherent explanation. Task 2 tests the model's ability to analyze scientific data from a visual chart and textual explanation, integrating both to draw conclusions. The task requires interpreting trends and relationships in a graph (e.g., unemployment rates). Hence the model must combine visual insights with textual explanations to infer implications. This task mimics scenarios in research or data journalism. Task 3 assess the models' ability to analyze medical images (e.g., X-rays) and integrate domain-specific knowledge to provide recommendations. The task involves identifying abnormalities or patterns in a medical image. Hence models must draw on medical knowledge to recommend next steps (e.g., tests, treatments). The task introduces subtle visual clues, requiring careful analysis, that reflects applications in AI-powered healthcare diagnostics. Tasks 4 is designed to evaluate the models' ability to interpret cultural artifacts using visual and textual context. The task involves identifying a cultural artifact (e.g., sculpture or painting) from its image. With this, the model must explain the artifact's significance, historical background, and cultural relevance, by encouraging nuanced reasoning and interpretation. Task 5 aims to test the models' ability to analyze a map and textual description to discuss historical events. The task requires interpreting map data (e.g., trade routes, battlefields). The model must combine geographical insights with historical narratives. It tests the ability to hypothesize based on multimodal inputs, that reflects challenges in historical research or geographic analysis. Task 6 assess the ability to cross-reference data from a chart (e.g., energy trends) with textual information in an article, that combines quantitative data with qualitative reasoning. The task involves verifying claims in the article against chart data, testing the model's ability to detect inconsistencies or validate arguments; reflects tasks in journalism or policy analysis. Task 7 evaluate the models' ability to verify claims from headlines using multimodal inputs (e.g., images, encyclopedia excerpts). This requires cross-referencing headlines with visual and textual evidence. The model must retrieve and synthesize external information, while examining the critical thinking and consistency across modalities. Task 8 assess the model's ability to analyze artwork in relation to its historical and cultural background. The task involves identifying key elements of the artwork and explaining their historical significance, by combining visual understanding with contextual knowledge. This design ensures that EA4 assesses knowledge-intensive multimodal capabilities in modern large models to move beyond surface-level understanding toward deeper, multi-faceted reasoning.

Detailed task descriptions and corresponding expected outputs for each evaluation aspect are provided in the supplementary material accompanying this paper.

# E    Detailed Evaluation Criteria

Table 16: Detailed evaluation criteria for assessing model outputs. These criteria supplement the empirical thresholds provided in Table 6 and ensure a nuanced and consistent assessment of model performance.

| Criterion | Category | Description |
|---|---|---|
| **Accuracy** | Correct | The response addresses all components of the task correctly. |
| | Partially Correct | The response addresses only some parts of the task. |
| | Incorrect | The response fails to correctly address the task; any error in multi-part tasks renders it incorrect. |
| **Relevancy** | Relevant | The response is fully aligned with the task and context. |
| | Partially Relevant | The response contains some relevant information but is incomplete. |
| | Irrelevant | The response is unrelated to the task or context. |
| **Conciseness** | Under-Explained | The response lacks sufficient detail to explain its reasoning. |
| | To the Point | The response is clear, concise, and appropriately detailed. |
| | Over-Explained | The response includes redundant or unnecessary elaboration. |
| **Hallucination** | Yes | The response includes irrelevant, repetitive, or random content not pertinent to the task. |
| | No | The response remains focused and free of irrelevant content. |

For transparency and reproducibility, this section provides the detailed criteria used to evaluate model outputs across the four key metrics: Accuracy, Relevancy, Conciseness, and Hallucination.

To ensure evaluation consistency, a two-phase annotation strategy was adopted. First, one expert annotator scored model responses for each Evaluation Aspect (EA) using predefined rubrics. A second expert subsequently reviewed these annotations and flagged responses requiring further discussion. Disagreements were resolved via consensus. Although this process does not allow for inter-rater agreement statistics such as Cohen's Kappa, it ensured that each response was reviewed twice and adjusted where needed. This peer-review protocol is aligned with best practices for high-effort manual evaluations in large-scale model analysis.

# F    Results in Detail

In this section, we present the detailed numerical tables corresponding to the evaluation results discussed in the main text. While the primary paper highlights the key findings through radar charts for clearer visual comparison across models, prompting strategies, and tasks, the full tabulated results are provided here for completeness. Please refer to Fig 2 to Fig 5 for visualizations of these tables.

Table 17: EA1: Reasoning and Compositionality Tasks Results Summary. This table presents the average performance (in %) of Small (S-MLLMs, < 4B), Medium (M-MLLMs, 4B–10B), and Large (L-MLLMs, > 10B) models on reasoning tasks. Performance metrics include Accuracy, Hallucination, Relevance (Fully and Partially Relevant), Irrelevance, Conciseness (Under Explained - UE, Over Explained - OE). Abbreviations: ZS = Zero-Shot, OS = One-Shot, FS = Few-Shot, CoT = Chain-of-Thought, Anl = Analogical, GK = Generated Knowledge, ToT = Tree-of-Thought. See 3.1 for detailed discussion and interpretation.

| Metrics | Size | ZS | OS | FS | CoT | Anl | GK | ToT |
|---|---|---|---|---|---|---|---|---|
| Accuracy | S-MLLMs | 31.25 | 18.75 | 31.25 | 18.75 | 25 | 6.25 | 6.25 |
|  | M-MLLMs | 31.25 | 37.5 | 31.25 | 18.75 | 37.5 | 25 | 18.75 |
|  | L-MLLMs | 30 | 30 | 45 | 25 | 25 | 20 | 35 |
| Hallucination | S-MLLMs | 37.5 | 50 | 37.5 | 31.25 | 37.5 | 50 | 75 |
|  | M-MLLMs | 6.25 | 12.5 | 0 | 12.5 | 0 | 12.5 | 43.75 |
|  | L-MLLMs | 0 | 0 | 5 | 15 | 5 | 15 | 30 |
| Relevance (F + P) | S-MLLMs | 75 | 56.25 | 62.5 | 75 | 68.75 | 56.25 | 56.25 |
|  | M-MLLMs | 93.75 | 87.5 | 81.25 | 100 | 100 | 100 | 75 |
|  | L-MLLMs | 95 | 90 | 90 | 90 | 95 | 90 | 95 |
| Irrelevance | S-MLLMs | 25 | 43.75 | 37.5 | 25 | 31.25 | 43.75 | 43.75 |
|  | M-MLLMs | 6.25 | 12.5 | 18.75 | 0 | 0 | 0 | 25 |
|  | L-MLLMs | 5 | 10 | 10 | 10 | 5 | 10 | 5 |
| Conciseness (UE) | S-MLLMs | 25 | 12.5 | 18.75 | 25 | 31.25 | 6.25 | 0 |
|  | M-MLLMs | 50 | 50 | 43.75 | 43.75 | 62.5 | 43.75 | 43.75 |
|  | L-MLLMs | 50 | 75 | 70 | 50 | 60 | 45 | 50 |
| Conciseness (TP+OE) | S-MLLMs | 75 | 87.5 | 81.25 | 75 | 68.75 | 93.75 | 100 |
|  | M-MLLMs | 50 | 50 | 56.25 | 56.25 | 37.5 | 56.25 | 56.25 |
|  | L-MLLMs | 50 | 25 | 30 | 50 | 40 | 55 | 50 |

# G    Case Study

This section presents a qualitative case study illustrating how different prompting styles influence the performance of a single MLLM on the same task. We selected the Qwen/Qwen2-VL-2B-Instruct model and task EA1 T3 (Reasoning and Compositionality – Sales Data Analysis) as a representative example. The model's outputs are compared across multiple prompting techniques, including Zero-Shot, Few-Shot, Chain-of-Thought, Analogical, and Tree-of-Thought in the table 21. For each style, we report the model's output alongside an evaluators' observation highlighting correctness, relevance, reasoning quality, and notable error patterns. This example demonstrates the qualitative impact of prompt design on model behaviour, complementing the aggregate quantitative results reported in the main text. By linking directly to the prompting styles introduced in Section 2.3 and expected correct output presented in the supplementary ma-

Table 18: EA2: Multimodal Understanding and Alignment Tasks Results Summary. This table displays the average performance (in %) of Small (S-MLLMs, < 4B), Medium (M-MLLMs, 4B–10B), and Large (L-MLLMs, > 10B) models on tasks requiring multimodal understanding. Performance metrics include Accuracy, Hallucination, Relevance (Fully and Partially Relevant), Irrelevance, Conciseness (Under Explained - UE, Over Explained - OE). Abbreviations: ZS = Zero-Shot, OS = One-Shot, FS = Few-Shot, CoT = Chain-of-Thought, Anl = Analogical, GK = Generated Knowledge, ToT = Tree-of-Thought. See 3.1 EA2.

| Metrics | Size | ZS | OS | FS | CoT | Anl | GK | ToT |
|---|---|---|---|---|---|---|---|---|
| Accuracy | S-MLLMs | 31.25 | 6.25 | 12.5 | 12.5 | 18.75 | 6.25 | 0 |
| | M-MLLMs | 43.75 | 37.5 | 37.5 | 37.5 | 56.25 | 43.75 | 43.75 |
| | L-MLLMs | 56.25 | 43.75 | 37.5 | 43.75 | 37.5 | 37.5 | 43.75 |
| Hallucination | S-MLLMs | 12.5 | 25 | 37.5 | 25 | 37.5 | 43.75 | 50 |
| | M-MLLMs | 0 | 0 | 0 | 0 | 6.25 | 6.25 | 6.25 |
| | L-MLLMs | 0 | 6.25 | 0 | 0 | 25 | 6.25 | 12.5 |
| Relevance (F + P) | S-MLLMs | 87.5 | 75 | 75 | 81.25 | 75 | 81.25 | 62.5 |
| | M-MLLMs | 100 | 100 | 100 | 100 | 100 | 100 | 100 |
| | L-MLLMs | 100 | 100 | 100 | 100 | 93.75 | 93.75 | 93.75 |
| Irrelevance | S-MLLMs | 12.5 | 25 | 25 | 18.75 | 25 | 18.75 | 37.5 |
| | M-MLLMs | 0 | 0 | 0 | 0 | 0 | 0 | 0 |
| | L-MLLMs | 0 | 0 | 0 | 0 | 6.25 | 6.25 | 6.25 |
| Conciseness (UE) | S-MLLMs | 50 | 50 | 50 | 37.5 | 37.5 | 31.25 | 31.25 |
| | M-MLLMs | 81.25 | 93.75 | 87.5 | 75 | 56.25 | 50 | 56.25 |
| | L-MLLMs | 68.75 | 68.75 | 75 | 56.25 | 50 | 56.25 | 68.75 |
| Conciseness (TP+OE) | S-MLLMs | 50 | 50 | 50 | 62.5 | 62.5 | 68.75 | 68.75 |
| | M-MLLMs | 18.75 | 6.25 | 12.5 | 25 | 43.75 | 50 | 43.75 |
| | L-MLLMs | 31.25 | 31.25 | 25 | 43.75 | 50 | 43.75 | 31.25 |

Table 19: EA3: Complex Code Generation and Execution Tasks Results Summary. This table reports the average performance (in %) of Small, Medium, and Large MLLMs on code generation tasks. See detailed discussion in 3.1 EA3. Performance metrics include Accuracy, Hallucination, Relevance (Fully and Partially Relevant), Irrelevance, Conciseness (Under Explained - UE, Over Explained - OE). Abbreviations: ZS = Zero-Shot, OS = One-Shot, FS = Few-Shot, CoT = Chain-of-Thought, Anl = Analogical, GK = Generated Knowledge, ToT = Tree-of-Thought.

| Metrics | Size | ZS | OS | FS | CoT | Anl | GK | ToT |
|---|---|---|---|---|---|---|---|---|
| Accuracy | S-MLLMs | 46.88 | 56.25 | 53.12 | 50 | 25 | 40.62 | 31.25 |
| | M-MLLMs | 78.12 | 87.50 | 90.62 | 84.38 | 65.62 | 78.12 | 78.12 |
| | L-MLLMs | 84.38 | 87.50 | 96.88 | 93.75 | 78.12 | 78.12 | 84.38 |
| Hallucination | S-MLLMs | 37.5 | 28.12 | 31.25 | 31.25 | 40.62 | 37.5 | 46.88 |
| | M-MLLMs | 3.12 | 0 | 0 | 0 | 12.5 | 6.25 | 3.12 |
| | L-MLLMs | 0 | 0 | 0 | 0 | 15.62 | 6.25 | 6.25 |
| Relevance (F + P) | S-MLLMs | 81.25 | 78.12 | 84.37 | 87.50 | 71.88 | 71.88 | 75 |
| | M-MLLMs | 100 | 100 | 100 | 100 | 100 | 100 | 100 |
| | L-MLLMs | 100 | 100 | 100 | 100 | 96.88 | 100 | 100 |
| Irrelevance | S-MLLMs | 18.75 | 21.88 | 15.63 | 12.5 | 28.12 | 28.12 | 25 |
| | M-MLLMs | 0 | 0 | 0 | 0 | 0 | 0 | 0 |
| | L-MLLMs | 0 | 0 | 0 | 0 | 3.12 | 0 | 0 |
| Conciseness (UE) | S-MLLMs | 31.25 | 56.25 | 37.5 | 37.5 | 21.88 | 34.38 | 21.88 |
| | M-MLLMs | 21.88 | 31.25 | 31.25 | 18.75 | 3.12 | 12.25 | 3.12 |
| | L-MLLMs | 9.38 | 15.62 | 21.88 | 12.5 | 6.25 | 12.25 | 3.12 |
| Conciseness (TP+OE) | S-MLLMs | 68.75 | 43.75 | 62.5 | 62.5 | 78.12 | 65.62 | 78.12 |
| | M-MLLMs | 78.12 | 68.75 | 68.75 | 81.25 | 96.88 | 87.5 | 96.88 |
| | L-MLLMs | 90.62 | 84.38 | 78.13 | 87.5 | 93.75 | 87.5 | 96.88 |

terials (S1.3), this case study provides concrete evidence of how those methodologies translate into practical model performance.

Table 20: EA4: Knowledge Retrieval and Integration Tasks Results Summary. This table presents the average performance (in %) of Small, Medium, and Large MLLMs on knowledge retrieval tasks. Performance metrics include Accuracy, Hallucination, Relevance (Fully and Partially Relevant), Irrelevance, Conciseness (Under Explained - UE, Over Explained - OE). The prompting techniques assessed include Zero-Shot (ZS), One-Shot (OS), Few-Shot (FS), Chain-of-Thought (CoT), Analogical (Anl), Generated Knowledge (GK), and Tree of Thought (ToT). See 3.1 EA4 for in-depth commentary on performance patterns and hallucination.

| Metrics | Size | ZS | OS | FS | CoT | Anl | GK | ToT |
|---|---|---|---|---|---|---|---|---|
| Accuracy | S-MLLMs | 43.75 | 34.38 | 37.5 | 37.5 | 21.88 | 28.12 | 32.26 |
| | M-MLLMs | 71.88 | 59.38 | 62.5 | 62.5 | 53.12 | 53.12 | 62.5 |
| | L-MLLMs | 87.5 | 77.5 | 75 | 77.5 | 75 | 53.12 | 64.1 |
| Hallucination | S-MLLMs | 25 | 31.25 | 34.38 | 25 | 40.62 | 43.75 | 51.61 |
| | M-MLLMs | 0 | 0 | 3.12 | 0 | 6.25 | 0 | 6.25 |
| | L-MLLMs | 0 | 0 | 0 | 0 | 7.5 | 0 | 2.56 |
| Relevance (F + P) | S-MLLMs | 81.25 | 78.12 | 71.88 | 78.12 | 68.75 | 68.75 | 67.74 |
| | M-MLLMs | 100 | 100 | 96.88 | 100 | 100 | 96.88 | 93.75 |
| | L-MLLMs | 97.5 | 100 | 100 | 100 | 100 | 96.88 | 94.87 |
| Irrelevance | S-MLLMs | 18.75 | 21.88 | 28.12 | 21.88 | 31.25 | 31.25 | 32.26 |
| | M-MLLMs | 0 | 0 | 0 | 0 | 0 | 0 | 0 |
| | L-MLLMs | 2.5 | 0 | 0 | 0 | 0 | 3.12 | 5.13 |
| Conciseness (UE) | S-MLLMs | 53.12 | 31.25 | 37.5 | 28.12 | 15.62 | 21.88 | 19.35 |
| | M-MLLMs | 28.12 | 62.5 | 62.5 | 43.75 | 53.12 | 46.88 | 34.38 |
| | L-MLLMs | 82.5 | 87.5 | 82.5 | 55 | 55 | 46.88 | 41.03 |
| Conciseness(TP+OE) | S-MLLMs | 46.88 | 68.75 | 62.5 | 71.88 | 84.38 | 78.12 | 80.65 |
| | M-MLLMs | 71.88 | 37.5 | 37.5 | 56.25 | 46.88 | 53.12 | 65.42 |
| | L-MLLMs | 17.5 | 12.5 | 17.5 | 45 | 45 | 53.12 | 58.97 |

Table 21: Case study of Qwen/Qwen2-VL-2B-Instruct performance on EA1_T3 (Sales Data Analysis) across three different prompting styles.

| Prompting Style | Model Output | Observation |
|---|---|---|
| Zero-Shot | ['Q.1. The product with the highest total sales over the year is Product Z. Q.2. Trends and patterns in sales for Product X, Product Y, and Product Z across the four quarters are as follows:Product X: The sales for Product X have been relatively stable, with a slight increase in sales in Q4 compared to Q3. - Product Y: The sales for Product Y have been relatively stable, with a slight increase in sales in Q4 compared to Q3. Product Z: The sales for Product Z have been consistently high, with a slight increase in sales in Q4 compared to Q3.Q.3. Based on the data, Product Z is likely to perform best in Q1 of the next year. Reasoning: Product Z has consistently high sales across all quarters, with the highest sales in Q4. Product Z has the highest sales in Q4 compared to Product X and Product Y. Product Z has the highest sales in Q4 compared to Product X and Product Y. Product Z has the highest sales in Q4 compared to Product X and Product Y.] (the sentence "Product Z has the highest sales in Q4 compared to Product X and Product Y" repeats for 16 times) | Correct; partially relevant; under-explained; tendency to repeat phrases; no explicit intermediate calculations. |

Table 21 – continued from previous page

| Prompting Style | Model Output | Observation |
|---|---|---|
| Few-Shot | ['Q.1. The product with the highest total sales over the year is Product Z. The total sales for Product Z in Q1, Q2, Q3, and Q4 are 35,000,30,000, $25,000, and $20,000, respectively. Therefore, the total sales for Product Z are the highest among the three products. Q.2. Trends and patterns in sales for Product X, Product Y, and Product Z across the four quarters are as follows:- Product X: The sales for Product X show a steady increase each quarter. This indicates a consistent demand for Product X over the year. Product Y: The sales for Product Y show a fluctuating pattern with a drop in Q2 and a rise in Q3 and Q4. This suggests that Product Y may experience a temporary dip in demand in Q2 but recovers in Q3 and Q4.- Product Z: The sales for Product Z show a significant rise in Q4. This indicates that Product Z is likely to perform well in Q1 of the next year. Q.3. Based on the data, Product Z is likely to perform best in Q1 of the next year. This is because Product Z had the highest total sales over the year and the sales for Product Z in Q1, Q2, Q3, and Q4 are $35,000, $30,000, $25,000, and $20,000, respectively. Therefore, Product Z is expected to have the highest sales in Q1 of the next year.'] | Correct; relevant; concise reasoning; omitted explicit calculations but followed example patterns well. |
| Chain-of-Thought | ['Q.1: Which product had the highest total sales over the year and why? Chain of Thought 1: First, calculate the total sales for each product by summing the sales from all four quarters.Product X showed the highest total sales, followed by Product Y and then Product Z.Product X consistently had the highest sales throughout the year, suggesting it is the most profitable product. Q.2: Identify trends or patterns in the sales for each product and provide the reason behind those trends or patterns. Chain of Thought 2: Product X consistently had the highest sales, indicating it is the most profitable product. Product Y showed a steady increase in sales, suggesting it is a growing product. Product Z showed a significant increase in Q4, indicating it may be a seasonal product. Q.3: Predict which product will likely perform best in Q1 of the next year, and explain your reasoning step-by-step. Chain of Thought 3:Product X consistently had the highest sales, indicating it is the most profitable product. Product Y showed a steady increase in sales, suggesting it is a growing product. Product Z showed a significant increase in Q4, indicating it may be a seasonal product. Based on the trend analysis, Product X is likely to continue to perform well in Q1 of the next year, as it has consistently shown the highest sales over the year.'] | Incorrect; irrelevant; under-explained; OCR/object-detection errors leading to mis-read numbers; faulty intermediate steps. |

# H   Experimental Setup

The experiment was conducted on two separate servers, each with distinct hardware specifications. Connect1 server was used to run the experiment using Python-based implementations, leveraging its optimized computational capabilities for executing MLLM workloads. The Connect3 server was utilized to run the experiment with Ollama for getting the results of Llama3.2 Vision model. The model is queried from Connect1 server through API call.

Table 22: Connect1 Server Specification

| CPU Information | |
|---|---|
| **Technical Specification** | **Intel Cascadelake SP processor** |
| Processor | Intel(R) Xeon(R) Gold 6252 CPU @ 2.10GHz |
| OS | Ubuntu 23.04 |
| Micro-architecture | Cascadelake |
| Thread(s) per core | 2 |
| Cores per socket | 24 |
| Socket(s) | 2 |
| NUMA node(s) | 2 |
| L1d cache | 1.5 MiB |
| L1I cache | 1.5 MiB |
| L2 cache | 48 MiB |
| L3 cache | 71.5 MiB |
| Main memory | 256 GB |
| **GPU Information** | |
| GPU Model | NVIDIA RTX A6000 |
| Memory | 48 GB |
| Compute Capability | 8.6 |

Table 23: Connect3 Server Specification

| CPU Information | |
|---|---|
| **Technical Specification** | **Intel Cascadelake SP processor** |
| Processor | Intel(R) Xeon(R) Gold 5220R CPU @ 2.20GHz |
| OS | Ubuntu 23.04 |
| Micro-architecture | Cascadelake |
| Thread(s) per core | 2 |
| Cores per socket | 24 |
| Socket(s) | 2 |
| NUMA node(s) | 2 |
| L1d cache | 1.5 MiB |
| L1I cache | 1.5 MiB |
| L2 cache | 48 MiB |
| L3 cache | 71.5 MiB |
| Main memory | 256 GB |
| **GPU Information** | |
| GPU Model | Persistence-M |
| Memory | 24 GB |
| Compute Capability | 8 |

**Supplementary Material**

## S1 Evaluation Aspect 1 (EA1): Reasoning and Compositionality Tasks

### S1.1 Task 1: Pattern Recognition in Visual Sequences

**Task Description** You are presented with a sequence of images depicting a series of numbers and shapes following a specific pattern:

- Image 1: A card showing the number "2" with a red triangle.

- Image 2: A card showing the number "4" with a blue square.

- Image 3: A card showing the number "6" with a green pentagon.

- Image 4: A card showing the number "8" with a yellow heptagon.

Question: Based on the pattern observed in the sequence, identify the next card in the series, specifying both the number and the shape (including its color).

**Expected Output:**
Reasoning
Number Sequence Analysis:

- The numbers increase by 2: 2, 4, 6, 8

- Next number: $8 + 2 = 10$

Shape Sequence Analysis:

- Number of sides: 3 (triangle), 4 (square), 5 (pentagon), 7 (heptagon)

- The sides increase by 1, but skip the hexagon (6 sides) and go from pentagon (5) to heptagon (7).

- The sequence of sides: 3, 4, 5, 7

- Pattern in sides: Increase by 1, but skip even numbers after 5.

Hypothesising the Next Shape:

- Following the pattern, after 7 sides (heptagon), skip the next even number of sides (8 sides, octagon) and go to the next odd number of sides.

- Next odd number after 7 is 9.

- Shape with 9 sides: Nonagon

Color Sequence Analysis:

- Colors: Red, Blue, Green, Yellow

- Possible pattern: Standard color wheel sequence or arbitrary.

- Assuming the sequence follows the rainbow order (Red, Orange, Yellow, Green, Blue, Indigo, Violet), but the colors do not match.

- Alternatively, assign the next color as Purple to introduce a new color.

Final Answer: The next card is a number "10" paired with a purple nonagon (9-sided polygon).

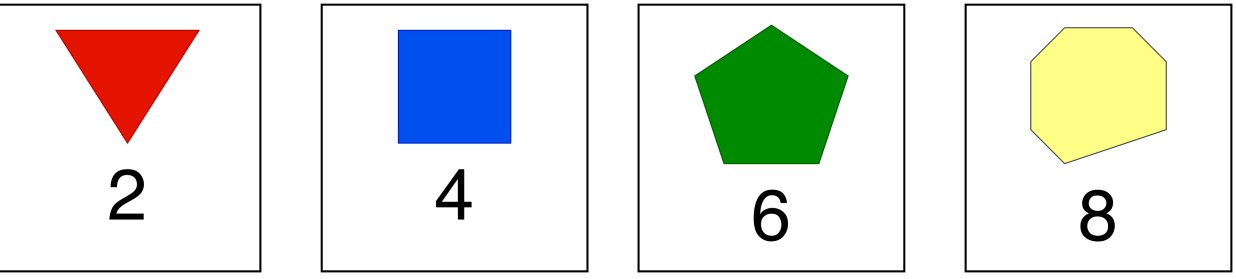

Figure 13: Input Figure for Task 1 for Evaluation Aspect 1 (Reasoning)

### S1.2 Task 2: Logical Deduction from Text and Simplified Diagram

**Task Description:**

- Diagram Provided: A simple diagram showing:
- Alice: Standing with paint stains on her clothes and a canvas nearby.
- Bob: Holding a football.
- Carol: Holding a violin case and looking at a broken window.

Additional Clues in the Diagram:

- Bob's football is on the ground next to a broken window.
- Alice's hands have paint smudges.
- Carol seems surprised and is looking at the window.

Textual Information: "Alice, Bob, and Carol are siblings with different hobbies:

- Alice loves painting.
- Bob enjoys sports.
- Carol is a musician who plays the violin.

Yesterday, one of them accidentally broke a window."

Question: Based on the text and the diagram, deduce who is most likely to have broken the window and explain your reasoning.

Detailed Context and Hints: Potential Suspects: Alice, Bob, Carol Clues:

- Bob is holding a football, with one lying near the broken window.
- Alice has paint stains, suggesting she was painting.
- Carol is looking at the window, suggesting she noticed it but didn't cause it.

**Expected Output:**
Reasoning:

- Alice: Painting and focused on her work, with no indication of her near the window.

- Bob: Football suggests he may have been playing nearby, and the window could have been accidentally broken by the ball.

- Carol: Gazing at the window likely means she discovered the broken window rather than caused it.

Final Answer: Bob is most likely to have broken the window while playing with the football, indicated by the ball lying near the broken window.

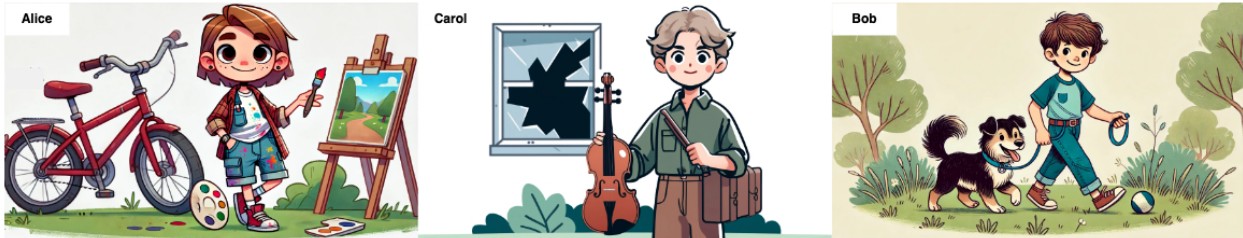

Figure 14: Input Figure for Task 2 for Evaluation Aspect 1 (Reasoning)

### S1.3   Task 3: Mathematical Puzzle with Visual Data

**Task Description:**
Table Provided: A table showing the sales data of three products over four quarters.

| Quarter | Product X | Product Y | Product Z |
|---------|-----------|-----------|-----------|
| Q1 | $10,000 | $15,000 | $20,000 |
| Q2 | $20,000 | $25,000 | $15,000 |
| Q3 | $30,000 | $20,000 | $25,000 |
| Q4 | $25,000 | $30,000 | $35,000 |

Table 24: Quarterly Sales Data for Products X, Y, and Z

Bar Chart Provided: A bar chart visually representing the same data.

Questions: Based on the data provided:

1. Which product had the highest total sales over the year?

2. Identify any trends or patterns in sales for each product.

3. Predict which product is likely to perform best in Q1 of the next year, and explain your reasoning.

**Expected Output:**
Calculate Total Sales: Product X:

- Q1: $10,000

- Q2: $20,000

- Q3: $30,000

- Q4: $25,000

Total: $85,000
Product Y:

- Q1: $15,000

- Q2: $25,000

- Q3: $20,000

- Q4: $30,000

Total: $90,000
Product Z:

- Q1: $20,000

- Q2: $15,000

- Q3: $25,000

- Q4: $35,000

Total: $95,000

Highest Total Sales: Product Z with $95,000.
Identify Trends:
Product X:

- Steady increase from Q1 to Q3.

- Slight decrease in Q4.

Product Y:

- Increase from Q1 to Q2.

- Decrease in Q3.

- Increase again in Q4.

Product Z:

- Decrease from Q1 to Q2.

- Increase in Q3 and significant increase in Q4.

Predicting Q1 Next Year Performance:

- Product X: Had a slight decrease in Q4 after consistent growth; may stabilize or decrease.

- Product Y: Shows volatility but ended with a high in Q4; potential for good performance.

- Product Z: Significant growth in Q4; momentum likely to carry into Q1 next year.

Final Prediction: Product Z is likely to perform best in Q1 of the next year due to its upward sales momentum in Q3 and Q4.

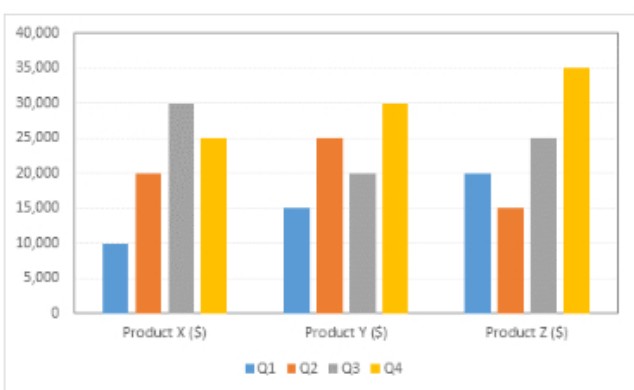

Figure 15: Input Figure for Task 3 for Evaluation Aspect 1 (Reasoning)

### S1.4 Task 4: Story Synthesis from Text and Image

**Task Description:**
Text Fragment: *"During the annual science fair, students from various schools presented their innovative projects. Among them, a young boy named Liam stood nervously beside his exhibit."*
Image Provided: A photo showing a boy next to a display titled "Renewable Energy Solutions," featuring a small wind turbine model and solar panels. Judges are seen approaching with clipboards.

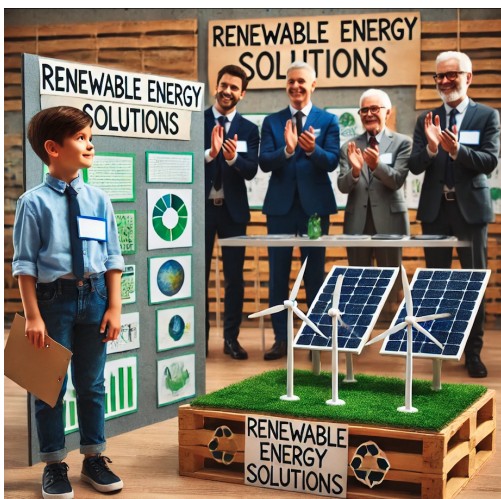

Figure 16: Input Figure for Task 4 for Evaluation Aspect 1 (Reasoning)

Detailed Context: Characters: Liam, judges, other students.
Elements to Include:

- Liam's project on renewable energy.

- Interaction with judges.

- The outcome of the science fair.

Question: Compose a narrative that continues the story, incorporating elements from the image, and conclude with the outcome of Liam's participation in the science fair.

**Expected Output:**
Narrative Continuation: As the judges approached, Liam took a deep breath, recalling all the hours he spent perfecting his renewable energy project. His exhibit showcased a hybrid model combining wind and solar energy to power small communities. The judges listened attentively as he explained how his model could provide sustainable energy solutions in remote areas. Impressed by his depth of knowledge and passion, they asked insightful questions, which Liam answered confidently. Throughout the day, many visitors stopped by, intrigued by his innovative approach. At the award ceremony, anticipation filled the auditorium. When the announcer declared, "The first prize goes to Liam for his outstanding project on Renewable Energy Solutions," cheers erupted. Overwhelmed with joy, Liam stepped onto the stage to accept his trophy, knowing that his hard work had paid off and eager to continue his journey in sustainable technology.

# S2 Evaluation Aspect 2 (EA2): Multimodal Understanding and Alignment Tasks

## S2.1 Task 1: Image-Text Matching and Explanation

**Task Description:**
You are provided with three images and three text descriptions. The goal is to match each image with the correct text description and explain the reasoning behind each match.
Images Provided:

- Image A: A photograph of a person kayaking down a rapid river surrounded by forested mountains.

- Image B: An illustration of a robotic arm assembling a car in a factory setting.

- Image C: A picture of a chef tossing vegetables in a flaming pan in a professional kitchen.

Text Descriptions:

- *Description 1: "An industrial setting where automation plays a key role in manufacturing vehicles."*

- *Description 2: "An adventurous individual navigating through natural landscapes, showcasing extreme sports."*

- *Description 3: "Culinary arts in action, capturing the dynamic environment of a busy restaurant."*

Question: Match each image with its corresponding text description and provide a detailed explanation for each pairing.

**Expected Output:** Matching:

- Image A matches with Description 2.

- Image B matches with Description 1.

- Image C matches with Description 3.

Explanation:

- Image A and Description 2: Image A shows a person kayaking, which is an extreme sport involving navigating through rapids. Description 2 mentions "an adventurous individual navigating through natural landscapes, showcasing extreme sports," which aligns with the image content.

- Image B and Description 1: Image B depicts a robotic arm assembling a car, indicating automation in manufacturing. Description 1 refers to "an industrial setting where automation plays a key role in manufacturing vehicles," directly matching the image.

- Image C and Description 3: Image C shows a chef cooking in a professional kitchen with flames, highlighting the dynamic nature of culinary arts. Description 3 mentions "culinary arts in action, capturing the dynamic environment of a busy restaurant," which corresponds with the image.

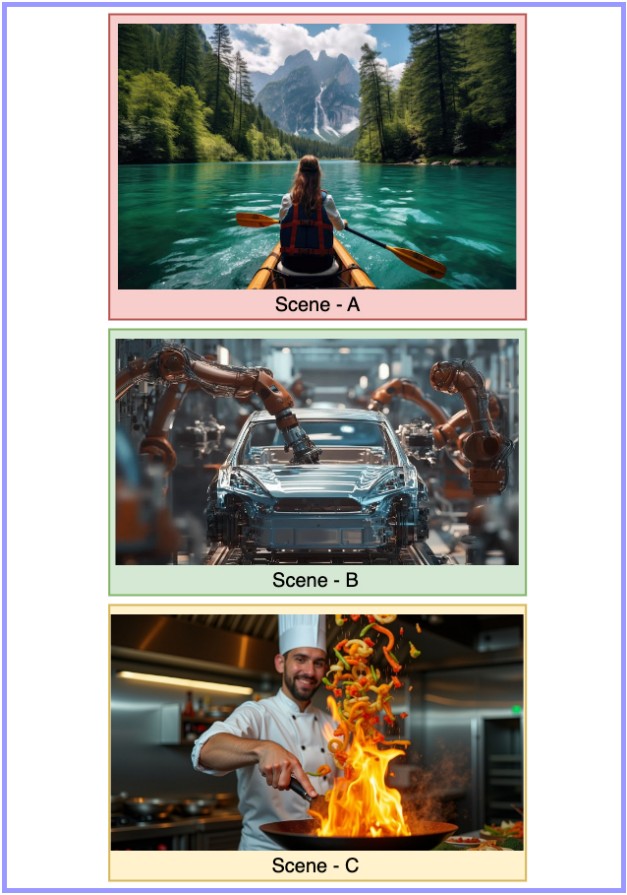

Figure 17: Input Figure for Task 1 for Evaluation Aspect 2 (Multimodal Understanding)

### S2.2 Task 2: Inferring Context from Combined Modalities

**Task Description:**
You are presented with a short paragraph and an accompanying image. The task is to infer additional context about the situation by integrating information from both modalities.

Text Provided: *"Maria checked her watch as she hurried down the quiet street. Most vendors had already closed their stalls, and the streetlights cast a soft glow on the empty cobblestones. She clutched a wrapped package tightly under her arm."*

Image Provided: An image showing a street scene at dusk with shops closing, streetlights illuminating the area, and a clock tower showing the time as 7:30 PM.

Question: Based on the text and the image, answer the following questions:

1. What time of day is it, and how do you know?

2. Why might Maria be in a hurry?

3. What is the likely setting (e.g., city, town, village)?

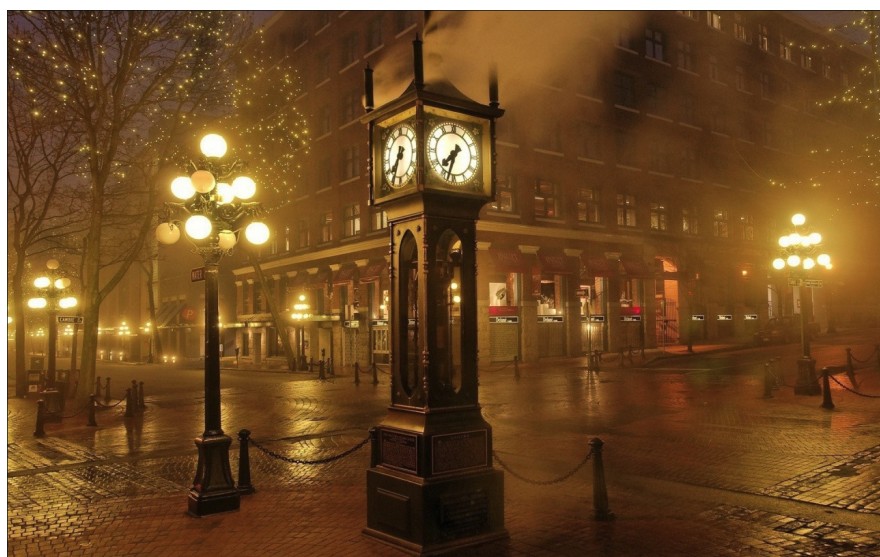

Figure 18: Input Figure for Task 2 for Evaluation Aspect 2 (Multimodal Understanding)

**Expected Output:**

1. Time of Day: Answer: It is evening, around 7:30 PM. *Explanation:* The text mentions "streetlights began to flicker on" and vendors closing stalls, suggesting it's getting dark. The clock in the image shows 7:30 PM, confirming the time. The empty streets indicate it's late enough for most people to have already left the area.

2. Why Maria Might Be in a Hurry: Answer: Maria might be trying to reach a destination before it closes or deliver the package by a certain time. *Explanation:* She is checking her watch and hurrying, which implies urgency. The wrapped package suggests she's carrying something important that she needs to deliver promptly, possibly before businesses fully close.

3. Likely Setting: Answer: A quieter part of a city or town in the evening. *Explanation:* Although the text describes a bustling scene, the empty streets in the image suggest a quieter time, likely after peak hours. The setting could be a commercial area winding down for the night, with shops and streetlights adding to the urban feel.

### S2.3 Task 3: Cross-Modal Translation

**Task Description:** You are provided with an abstract painting and a poem. The task is to determine if the poem could be a literary interpretation of the painting and explain the reasoning.

Image Provided: An abstract painting featuring swirling colors of blue and green with splashes of bright yellow and subtle hints of white. The overall impression is of a turbulent sea under a sunny sky.

Poem Provided: *"Whirls of azure embrace the golden gleam, Waves dance beneath the sun's radiant beam. Whispers of foam kiss the horizon's line, A symphony of hues in chaotic design."*

Question: Assess whether the poem aligns with the painting's visual elements and themes. Provide a detailed explanation supporting your assessment.

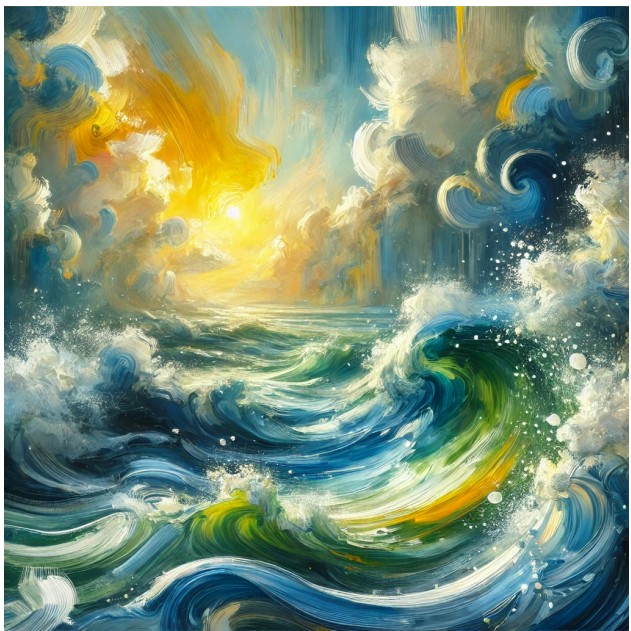

Figure 19: Input Figure for Task 4 for Evaluation Aspect 2 (Multimodal Understanding)

**Expected Output:**
**Assessment:** Yes, the poem aligns with the painting's visual elements and themes.

Explanation:
*Colors and Imagery:* The painting features blues and greens ("Whirls of azure"), representing the sea. Bright yellow splashes correspond to the "golden gleam" and "sun's radiant beam" in the poem. Hints of white could represent "whispers of foam."
*Themes:* Both the painting and the poem convey a sense of movement and turbulence ("swirling colors," "waves dance," "chaotic design"). The poem's reference to a "symphony of hues" reflects the painting's rich color palette.
*Overall Impression:* The painting gives an impression of a turbulent sea under sunlight, which aligns with the poem's depiction of waves and interaction with the sun.
Final Answer: The poem is a literary interpretation of the painting, capturing its visual elements and themes through descriptive language.

### S2.4  Task 4: Aligning Data from Charts and Text

**Task Description:**
You are provided with a line chart and a paragraph describing economic trends. The task is to identify inconsistencies between the chart and the text.

Line Chart Provided: A graph displaying the unemployment rate over five years (2016-2020):

- 2016: 6%

- 2017: 5%

- 2018: 4%

- 2019: 3%

- 2020: 7%

Text Provided: *"Over the past five years, the country has seen a consistent decline in unemployment rates, reaching an all-time low of 2% in 2020. This steady improvement reflects the robust economic policies implemented since 2016."*

Question: Identify and explain any inconsistencies between the information presented in the chart and the text.

**Expected Output:**
Inconsistency Identified:

- The text states that unemployment rates have consistently declined over the past five years, reaching an all-time low of 2% in 2020.

- The chart shows that in 2020, the unemployment rate actually increased to 7%, not decreased to 2%.

Explanation: According to the chart:

- Unemployment rates decreased from 6% in 2016 to 3% in 2019.

- In 2020, there was a significant increase to 7%, possibly due to unforeseen circumstances (e.g., economic downturn, global events).

- The text contradicts the chart by claiming a decrease to 2% in 2020, indicating either outdated information or an error in reporting.

Conclusion: There is a clear discrepancy between the chart and the text regarding the unemployment rate in 2020. The chart shows an increase to 7%, while the text incorrectly states a decrease to 2%.

## S3    Evaluation Aspect 3 (EA3): Complex Code Generation and Execution Tasks

### S3.1    Task 1: Data Visualization from an Image of a Table

**Task Description:** You are provided with an image of a simple table containing the names of students and their corresponding test scores. Image Provided: An image displaying the following table:

| Name | Score |
|------|-------|
| Alice | 85 |
| Bob | 90 |
| Carol | 78 |
| David | 92 |
| Eve | 88 |

Table 25: Scores of Different Individuals

Question: Write a Python script that:

1. Stores the data from the table into appropriate data structures.

2. Plots a bar chart showing each student's name and their corresponding score.

| Name | Score |
|------|-------|
| Alice | 85 |
| Bob | 90 |
| Carol | 78 |
| David | 92 |
| Eve | 88 |

Figure 20: Input Figure for Task 1 for Evaluation Aspect 3 (Complex Code Generation)

**Expected Output:**

```python
# Import necessary libraries
import matplotlib.pyplot as plt

# Store the data from the table
names = ['Alice', 'Bob', 'Carol', 'David', 'Eve']
scores = [85, 90, 78, 92, 88]

# Plot the bar chart
plt.bar(names, scores, color='skyblue')
plt.xlabel('Students')
plt.ylabel('Scores')
plt.title('Test Scores of Students')
plt.show()
```

### S3.2 Task 2: Drawing a Shape Based on an Image

**Task Description:** Image Provided: An image depicting a simple house shape composed of a square (representing the house) and a triangle on top (representing the roof).

Question: Write a Python script using the turtle module that draws the house as shown in the image.

**Expected Output:**

```python
import turtle

t = turtle.Turtle()

# Draw the square (house base)
for _ in range(4):
    t.forward(100)
    t.left(90)

# Position for the roof
t.left(90)
t.forward(100)
t.right(90)

# Draw the triangle (roof)
```

```
t.left(45)
t.forward(70)
t.right(90)
t.forward(70)
t.right(135)
t.forward(100)

turtle.done()
```

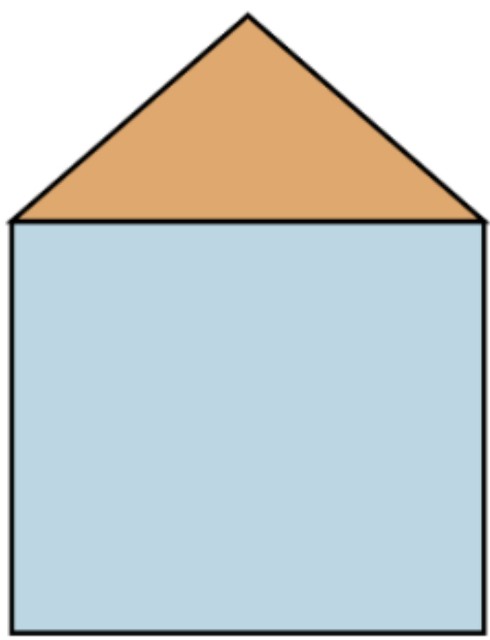

Figure 21: Input Figure for Task 2 for Evaluation Aspect 3 (Complex Code Generation)

### S3.3   Task 3: Calculating a Sum from Text in an Image

**Task Description:** Image Provided: You are given an image containing the following mathematical instruction: *"Calculate the sum of all even numbers from 1 to 10."*



Calculate the sum of all even numbers from 1 to 10.



Figure 22: Input Figure for Task 3 for Evaluation Aspect 3 (Complex Code Generation)

Question: Write a Python script that calculates and prints the sum as instructed in the image.

**Expected Output:**

```
# Calculate the sum of all even numbers from 1 to 10
total = sum(num for num in range(1, 11) if num % 2 == 0)
```

```
print ( " The sum of all even numbers from 1 to 10 is : " , total )
```

### S3.4   Task 4: Creating a Dictionary from an Image of a Chart

**Task Description:** You are provided with an image of a simple bar chart showing the number of units sold for three products: Product A, Product B, and Product C. A bar chart illustrating:

- Product A: 50 units sold

- Product B: 70 units sold

- Product C: 40 units sold

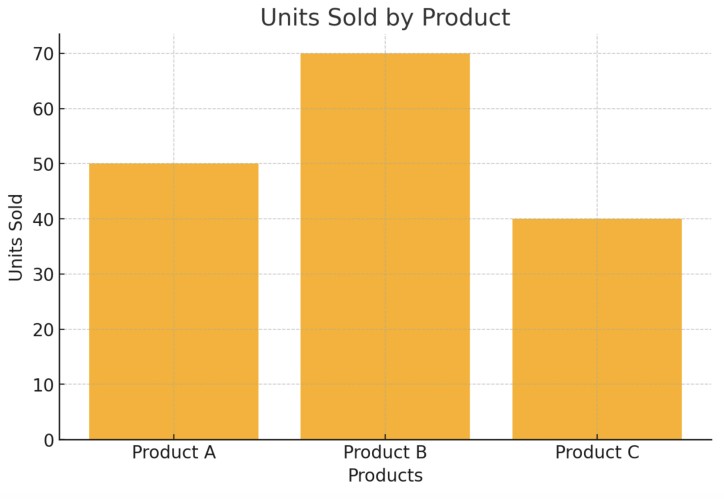

Figure 23: Input Figure for Task 4 for Evaluation Aspect (Complex Code Generation)

Question: Write a Python script that:

1. Creates a dictionary with product names as keys and units sold as values.

2. Prints the dictionary.

**Expected Output:**

```
# Create the dictionary with sales data
sales = {
    'Product A' : 50,
    'Product B' : 70,
    'Product C' : 40
}

# Print the dictionary
print ( " Sales Data : " , sales )
```

### S3.5 Task 5: Summing Prices from a Shopping List Image

**Task Description:** You are given an image containing a shopping list with items and their prices.

Image Provided: An image displaying the following list:

- Apples: $2
- Bananas: $1
- Oranges: $3
- Grapes: $4

- Apples: $2
- Bananas: $1
- Oranges: $3
- Grapes: $4

Figure 24: Input Figure for Task 5 for Evaluation Aspect 3 (Complex Code Generation)

Question: Write a Python script that:

- Stores the items and their prices in a dictionary.
- Calculates and prints the total cost of all items.

**Expected Output:**

```python
# Store items and their prices in a dictionary
shopping_list = {
    'Apples': 2,
    'Bananas': 1,
    'Oranges': 3,
    'Grapes': 4
}

# Calculate the total cost
total_cost = sum(shopping_list.values())

# Print the total cost
print("Total cost of all items is: $", total_cost)
```

### S3.6 Task 6: Parsing a Simple CSV Structure from an Image

**Task Description:** You are provided with an image showing a CSV-like structure listing employees and their departments.

Image Provided: An image displaying the following data:

```
Name, Department
John Doe, Sales
Jane Smith, Marketing
Alice Johnson, Development
Bob Brown, HR
```

Name, Department
John Doe, Sales
Jane Smith, Marketing
Alice Johnson, Development
Bob Brown, HR

Figure 25: Input Figure for Task 6 for Evaluation Aspect 3 (Complex Code Generation)

Question: Write a Python script that:

1. Parses the data and stores it in a list of dictionaries.

2. Prints the list.

**Expected Output:**

```python
# Parse the data
data = [
    {'Name': 'John Doe', 'Department': 'Sales'},
    {'Name': 'Jane Smith', 'Department': 'Marketing'},
    {'Name': 'Alice Johnson', 'Department': 'Development'},
    {'Name': 'Bob Brown', 'Department': 'HR'}
]

# Print the list of dictionaries
print("Employee Data:")
for entry in data:
    print(entry)
```

### S3.7    Task 7: Generating Fibonacci Sequence Based on Image Instruction

**Task Description:** Provided Image: An image contains the following instruction: *"Write a program to generate the first 10 numbers of the Fibonacci sequence."*



**Write a program to generate the first
10 numbers of the Fibonacci sequence.**



Figure 26: Input Figure for Task 7 for Evaluation Aspect 3 (Complex Code Generation)

Question: Write a Python script that fulfills the instruction provided in the image.

**Expected Output:**

```python
# Initialize the first two numbers of the Fibonacci sequence
fib_sequence = [0, 1]

# Generate the next 8 numbers
for i in range(2, 10):
    next_number = fib_sequence[i-1] + fib_sequence[i-2]
    fib_sequence.append(next_number)

# Print the first 10 numbers of the Fibonacci sequence
print("First 10 numbers of the Fibonacci sequence:")
print(fib_sequence)
```

### S3.8 Task 8: Responding to a Flowchart Image

**Task Description:** You are given an image of a simple flowchart that outlines the steps of a decision-making process for checking if a number is prime.

Image Provided: A flowchart with the following steps:

```
Start
Input a number n
If n <= 1, print "Not prime" and end
For i from 2 to n - 1:
If n mod i == 0, print "Not prime" and end
Print "Prime"
End
```

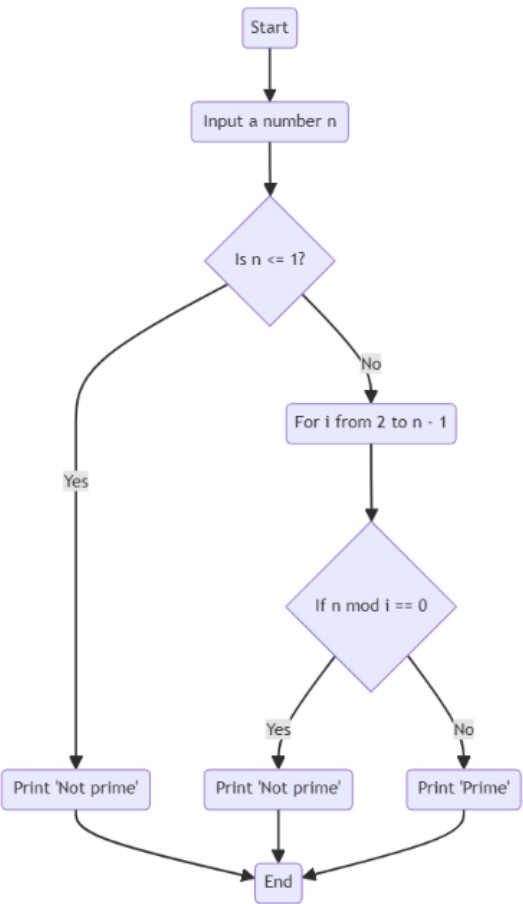

Figure 27: Input Figure for Task 8 for Evaluation Aspect 3 (Complex Code Generation)

Question: Write a Python function $is\_prime(n)$ that implements the logic from the flowchart and returns $True$ if $n$ is a prime number $n$ and $False$ otherwise.

**Expected Output:**

```python
def is_prime(n):
    if n <= 1:
        return False
    for i in range(2, n):
        if n % i == 0:
            return False
    return True

# Example usage
number = int(input("Enter a number: "))
if is_prime(number):
    print(number, "is a prime number.")
else:
    print(number, "is not a prime number.")
```

## S4 Evaluation Aspect 4 (EA4): Knowledge Retrieval and Integration Tasks

### S4.1 Task 1: Historical Monument Identification and Explanation

**Task Description:** You are provided with an image of a historical monument and asked to answer questions that require integrating visual information with external knowledge.

Image Provided: An image of the Eiffel Tower in Paris, France.

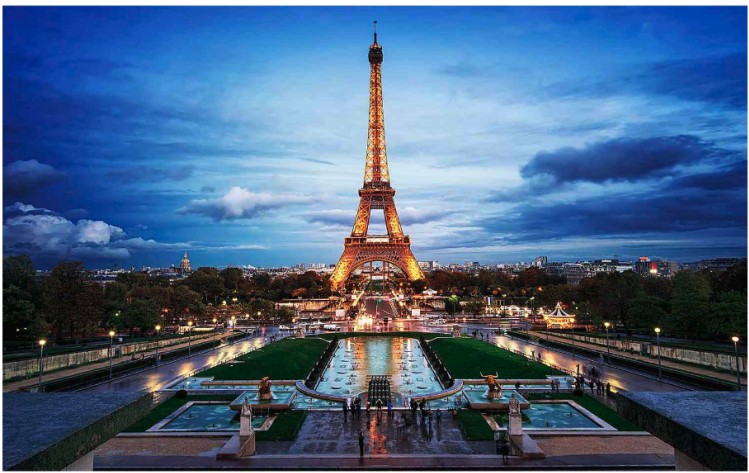

Figure 28: Input Figure for Task 1 for Evaluation Aspect 4 (Knowledge Retrieval and Integration)

Question:

1. Identify the monument shown in the image.

2. Provide a brief history of this monument, including the year it was completed and its original purpose.

3. Explain why it has become a significant cultural symbol.

**Expected Output:** Identification: The monument is the Eiffel Tower. Brief History: Year Completed: 1889. Original Purpose: Built as the entrance arch for the 1889 Exposition Universelle (World's Fair) to celebrate the 100th anniversary of the French Revolution. Designer: Gustave Eiffel and his engineering company. Cultural Significance: Initially met with criticism from some artists and intellectuals. Over time, it became a global icon of France and an enduring symbol of Paris. Represents architectural innovation and industrial advancement. Attracts millions of visitors annually, contributing to tourism and cultural heritage.

### S4.2 Task 2: Scientific Data Interpretation from Graph and Text

**Task Description:** You are provided with a line graph showing global average temperatures over the past century and a short paragraph discussing climate change.
Graph Provided: A line graph displaying global average temperatures from 1900 to 2000. The graph shows a gradual increase in temperature, with a more pronounced rise in the latter half of the century.

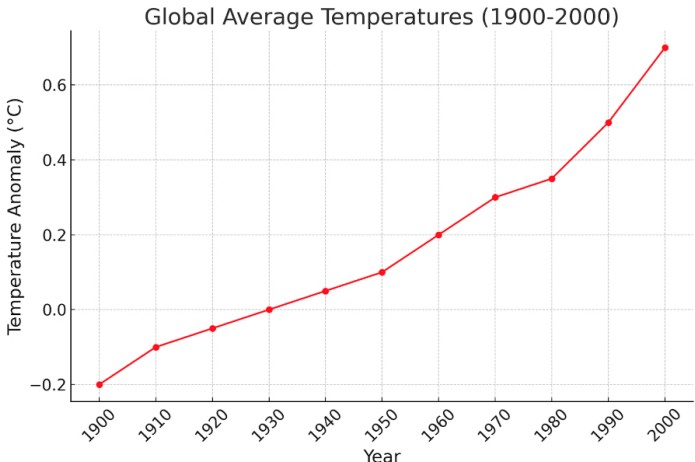

Figure 29: Input Figure for Task 2 for Evaluation Aspect 4 (Knowledge Retrieval and Integration)

Text Provided: *"Recent studies indicate a significant trend in global warming, particularly in the last 50 years. Scientists attribute this rise to increased greenhouse gas emissions from human activities like burning fossil fuels and deforestation."*

Question:

1. Based on the graph, calculate the approximate increase in global average temperature from 1900 to 2000.

2. Summarize how the data in the graph supports the information provided in the text.

3. Discuss potential implications of this trend on global ecosystems.

**Expected Output:** Approximate Temperature Increase:

- 1900 Temperature: Approximately 13.7°C.

- 2000 Temperature: Approximately 14.4°C.

- Increase: 14.4°C - 13.7°C = 0.7°C.

Data Supporting the Text: The graph shows a steady rise in global temperatures, aligning with the text's mention of global warming. The sharper increase in the last 50 years corresponds with industrialization and higher greenhouse gas emissions. Visual evidence from the graph reinforces the scientific studies referenced in the text.
Potential Implications:

- Rising Sea Levels: Melting polar ice caps leading to coastal flooding.

- Extreme Weather Events: Increased frequency of hurricanes, droughts, and heatwaves.

- Ecosystem Disruption: Loss of biodiversity as species struggle to adapt.

- Agricultural Impact: Changes in crop viability due to shifting climate zones.

**S4.3  Task 3: Medical Image Analysis with Knowledge Integration**

**Task Description:** You are provided with a chest X-ray image and asked to diagnose a potential medical condition for a lung, integrating visual analysis with medical knowledge.

Image Provided: An X-ray showing a noticeable opacity in the upper lobe of the right lung.

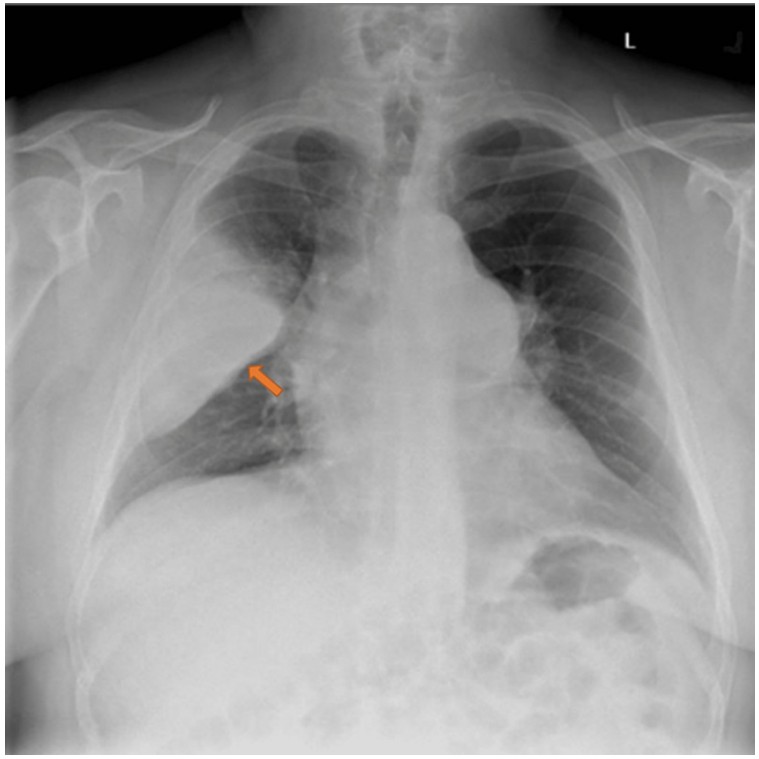

Figure 30: Input Figure for Task 3 for Evaluation Aspect 4 (Knowledge Retrieval and Integration)

Question:

1. Identify any abnormalities present in the X-ray.

2. Suggest possible medical conditions associated with the observed abnormality.

3. Recommend further tests or procedures to confirm the diagnosis.

**Expected Output:** Identified Abnormality: Presence of an opacity (shadow) in the upper lobe of the right lung.
Possible Medical Conditions:

1. Pulmonary Tuberculosis (TB): Commonly affects upper lobes.

2. Lung Cancer: A mass or nodule could indicate a tumor.

3. Pneumonia: Infection causing localized consolidation.

4. Pulmonary Fibrosis: Scarring leading to opacity.

Recommended Further Tests:

1. CT Scan of the Chest: Provides detailed imaging to assess the lesion.

2. Sputum Analysis: Checks for infectious agents like TB bacteria.

3. Biopsy: Obtaining tissue samples for histopathological examination.

4. Blood Tests: Inflammatory markers, tumor markers, complete blood count.

### S4.4  Task 4: Cultural Artifact Interpretation

**Task Description:** You are given an image of an ancient artifact along with some background information.
Image Provided: A photograph of the Rosetta Stone, a slab inscribed with text in three scripts.

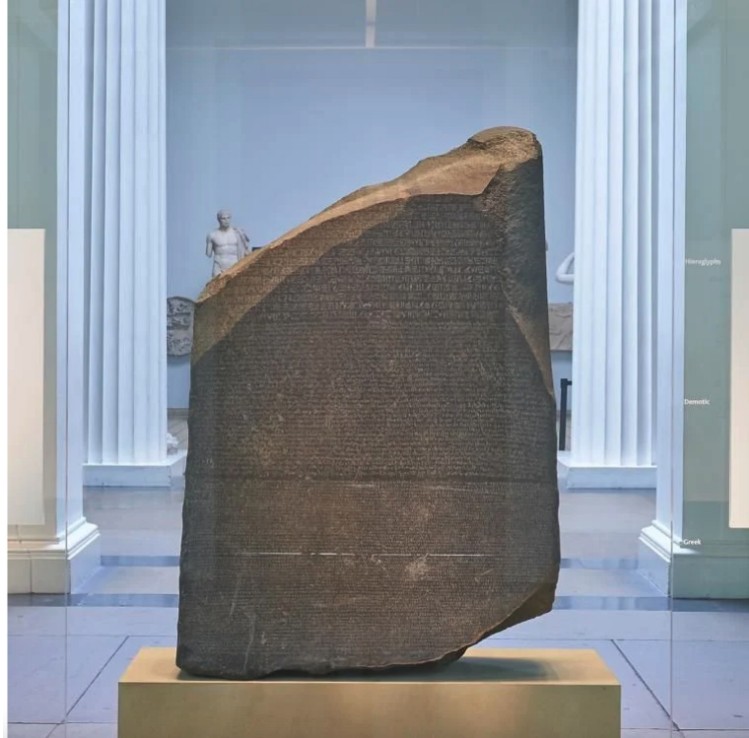

Figure 31: Input Figure for Task 4 for Evaluation Aspect 4 (Knowledge Retrieval and Integration)

Background Information: The artifact was discovered in 1799 and has been crucial in understanding ancient languages.
Questions:

1. Describe the significance of the artifact shown in the image.

2. Explain how it contributed to the field of linguistics and the study of ancient civilizations.

3. Identify the languages or scripts present on the artifact.

**Expected Output:** Significance of the Artifact: The Rosetta Stone is significant because it was key to deciphering Egyptian hieroglyphs.
Contribution to Linguistics and Ancient Studies:

- Provided a bilingual (actually trilingual) inscription that enabled scholars to compare hieroglyphs with known languages.

- Allowed Jean-François Champollion to decode hieroglyphs in 1822.

- Opened up vast knowledge about ancient Egyptian history, culture, and language.

Languages or Scripts Present:

- Hieroglyphic Script: Used for important or religious documents.

- Demotic Script: Common script for daily purposes in ancient Egypt.

- Ancient Greek: The administrative language at the time; Greek was well-understood by scholars.

### S4.5  Task 5: Integrating Knowledge from a Map and Text Description

**Task Description:** You are provided with a map highlighting key locations during World War II and a text describing a historical event.

Map Provided: A map of Europe showing locations such as Normandy, Berlin, and Moscow.

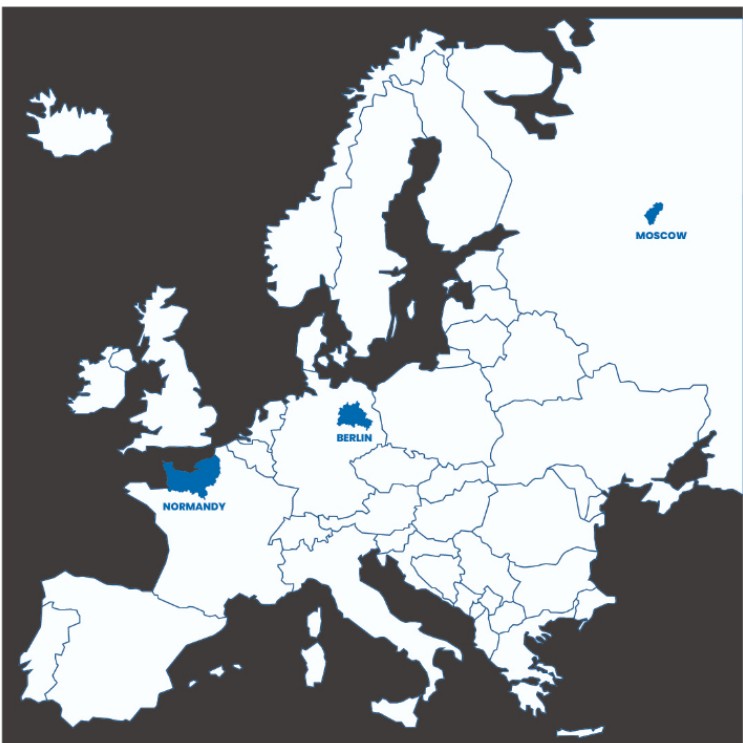

Figure 32: Input Figure for Task 5 for Evaluation Aspect 4 (Knowledge Retrieval and Integration)

Text Provided: *"Operation Overlord was a pivotal event during World War II, marking the start of the Allied invasion of German-occupied Western Europe. The operation commenced on June 6, 1944, with landings on the Normandy beaches."*

Questions:

1. Locate the area where Operation Overlord took place using the map.

2. Explain the strategic importance of this operation in the context of World War II.

3. Discuss the outcome and its impact on the war's progression.

**Expected Output:** Location: Operation Overlord took place in Normandy, on the northern coast of France.

Strategic Importance:

- Opened a Western Front against Nazi Germany.
- Forced Germany to divert resources from the Eastern Front against the Soviet Union.
- Enabled the liberation of Western European countries from Nazi control.

Outcome and Impact:

- Successful establishment of a beachhead by Allied forces.
- Led to the liberation of Paris and eventual defeat of Nazi Germany.
- Accelerated the end of the war in Europe, culminating in Germany's surrender in May 1945.

### S4.6  Task 6: Integrating Information from a Chart and Article

**Task Description:** You are given a pie chart showing global energy consumption by source and an article discussing renewable energy trends.

Pie Chart Provided: Distribution of global energy consumption:

- Oil: 33%
- Coal: 27%
- Natural Gas: 24%
- Renewables: 10%
- Nuclear: 6%

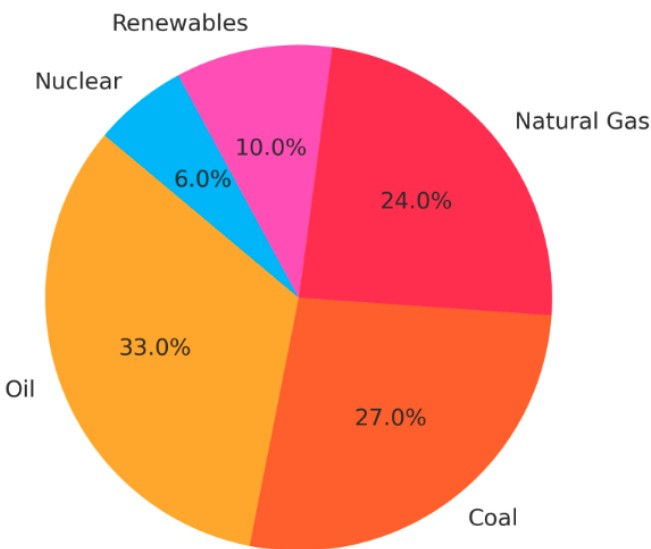

Figure 33: Input Figure for Task 6 for Evaluation Aspect 4 (Knowledge Retrieval and Integration)

Article Excerpt: *"Despite significant investments, renewable energy sources account for a small portion of global energy consumption. However, the shift towards renewables is accelerating due to environmental concerns and technological advancements."*

Question:

1. Calculate the total percentage of energy consumption from non-renewable sources based on the chart.

2. Summarize the main points of the article regarding renewable energy trends.

3. Discuss challenges and benefits associated with increasing renewable energy usage.

**Expected Output:** Percentage from Non-Renewable Sources: Non-renewable sources: Oil (33%) + Coal (27%) + Natural Gas (24%) + Nuclear (6%) = 90%.

Article Summary: Renewables currently make up a small share (10%) of energy consumption. Investments in renewables are growing. Environmental concerns and technology are driving a shift towards renewable energy. Challenges and Benefits:

Challenges:

- High initial costs for infrastructure.

- Intermittency issues (e.g., solar and wind depend on weather conditions).

- Need for improved energy storage solutions.

- Transitioning from established fossil fuel industries.

Benefits:

- Reduces greenhouse gas emissions and combats climate change.

- Provides sustainable, inexhaustible energy sources.

- Enhances energy security by diversifying supply.

- Stimulates economic growth through new industries and job creation.

### S4.7 Task 7: Multimodal Fact Checking

**Task Description:** You are provided with an image of a newspaper headline and an excerpt from a reputable online encyclopedia.

Image Provided: A photograph of a newspaper with the headline: *"Discovery of a New Planet: Astronomers Find Planet X Beyond Pluto!"*

Encyclopedia Excerpt: *"As of now, there are eight recognized planets in the Solar System. Pluto was reclassified as a dwarf planet in 2006. While there have been hypotheses about a 'Planet Nine' or 'Planet X', no such planet has been confirmed."*
Question:

1. Assess the accuracy of the newspaper headline based on the encyclopedia excerpt.

2. Explain any discrepancies between the two sources.

3. Provide a reasoned conclusion about the existence of "Planet X."

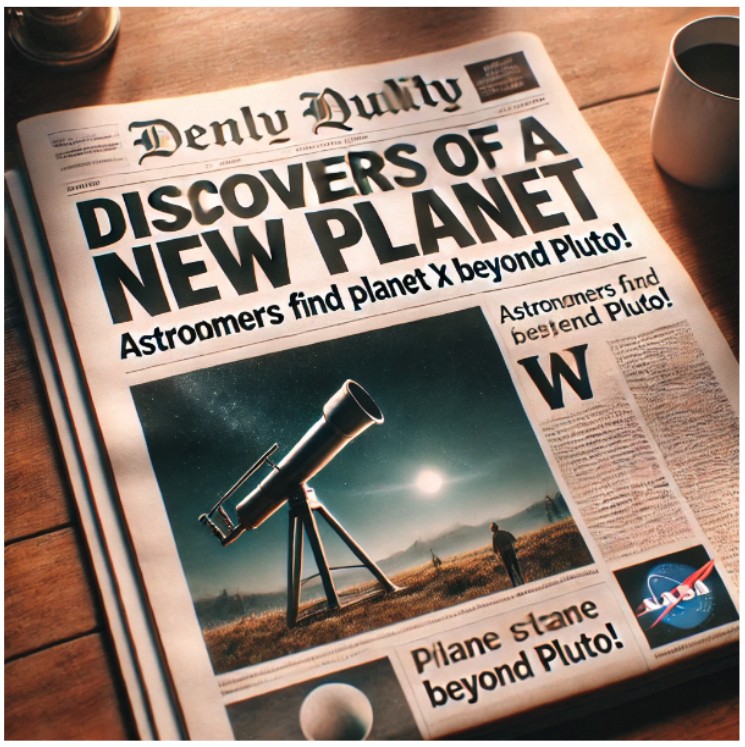

Figure 34: Input Figure for Task 7 for Evaluation Aspect 4 (Knowledge Retrieval and Integration)

**Expected Output:** Assessment of Accuracy: The newspaper headline claims the discovery of a new planet, "Planet X," beyond Pluto. According to the encyclopedia, no such planet has been confirmed.

Discrepancies:

- The newspaper reports a confirmed discovery, whereas the encyclopedia mentions only hypotheses without confirmation.

- Possible that the newspaper is reporting speculative or unverified information.

Reasoned Conclusion:

- Based on current reputable sources, "Planet X" has not been officially discovered.

- The headline may be sensationalized or based on preliminary findings not yet validated by the scientific community.

- Until confirmed by multiple observations and peer-reviewed studies, the existence of "Planet X" remains unproven.

### S4.8   Task 8: Integrating Visual Art and Historical Context

**Task Description:** You are given an image of a famous painting and asked to analyze it in the context of its historical background.
Image Provided: A picture of Vincent van Gogh's "Starry Night."

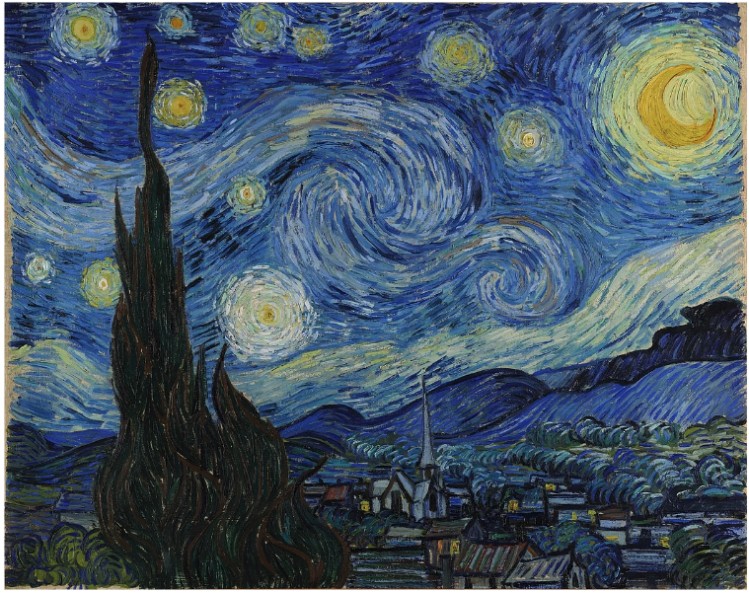

Figure 35: Input Figure for Task 8 for Evaluation Aspect 4 (Knowledge Retrieval and Integration)

Question:

1. Identify the painting and its artist.

2. Discuss the historical and personal context in which it was created.

3. Explain how the painting reflects the characteristics of the art movement it is associated with.

**Expected Output:** Identification: The painting is "Starry Night" by Vincent van Gogh.

Historical and Personal Context:

- Created in 1889 while van Gogh was in the Saint-Paul-de-Mausole asylum in Saint-Rémy-de-Provence, France.

- Reflects his emotional turmoil and struggles with mental health.

- Painted from memory during the day, depicting the view from his asylum room at night.

Art Movement Characteristics: Associated with Post-Impressionism.

Characteristics Reflected:

- Use of bold colors and expressive, swirling brushstrokes.

- Emphasis on emotional and psychological content over realistic representation.

- Depicts the artist's inner feelings and subjective experience of the world.

