# OpenReview forum: "The Future of MLLM Prompting is Adaptive: A Comprehensive Experimental Evaluation of Prompt Engineering Methods for Robust Multimodal Performance"
_TMLR — Accepted by TMLR_

### Review · Reviewer_9CD2 · 2025-06-25

**Summary Of Contributions:**

The paper proposes creation of a benchmark with 4 evaluation aspects and multiple tasks per aspect to test the capabilities of open source MLLMs and different prompting methods. The 4 evaluation aspects are designed to asses the various dimensions of models multimodal capabilities such as reasoning, alignment, code generation and knowledge retrieval, and the tasks in each aspect are designed to test how a model performs in these aspects. 13 MLLM's were selected based on opensource, architectural diversity and segregated into small, medium and large based on parameter count. The paper evaluates 7 popular prompting mechanisms. The evaluation is done in terms of accuracy, hallucination, relevance and conciseness. The results show that while scaling is useful, it does not necessarily always correlate with universally better performance across diverse reasoning aspects or tasks. Similarly with prompting mechanisms, the paper shows that there is no single prompting mechanism fit for all tasks.

**Audience:**

Yes

**Claims And Evidence:**

Yes

**Requested Changes:**

**Critical:**
- Describe the details of how the evaluation was done.
- The citations should all be parenthetical (\citep). This is a persistent issue throughout the paper, that is, in-text citations are frequently used when parenthetical citations should be used instead. Please fix this in all places where it occurs.
- Remove duplicate text in citations: For example: Ghosh et al. Ghosh et al. (2024), and many more, again a persistent issue.
- Table 6 thresholds seem very heuristic, what is the rationale or method to select these values? There is a brief description in Sec. 2.4.2, but in my opinion it is not adequate, for example if SOTA models fail to exceed 75%, then why not 76% or why not 90%, the choice of these thresholds seem arbitrary and no empirical evidence is porvided for why 80% and not 90%.
- There are many tables and the text descriptions are most often in different pages, making it harder to follow the text and look at the table. It might be better to also put a reference to the in-text description of the results in a table in the table caption, so that the reader can go to the table from the text reference and go to the text from the table reference.
- Release a test harness code and the evaluation data in a structured format

**Strengths And Weaknesses:**

**Strengths**
- Evaluation aspects and tasks are well designed
- Diverse set of models and prompting mechanisms
- Good analysis of the capabilities and interactions of prompting mechanisms with model sizes across evaluation aspects

**Weaknesses**
- While the design of evaluation aspects and tasks is well thought out, however, it might not be adequate for certain tasks such as knowledge retrieval, that a single image and a single task is utilized. For example, some model might have been trained on more images from France than others (supplementary S4.1).
- There is no information about how the evaluation was carried out. How was accuracy computed? How was hallucination, conciseness, relevance evaluated? Was it human evaluators? Table 20 (Appdx. E) gives some detailed criteria, but it is not clear how these are implemented.
- There is no code released. Since it can be used as a benchmark to evaluate other models or prompting mechanisms, it is paramount that a standard test harness is released.
- The evaluation data in the supplementary is not in a format that could be easily used to test other models.
- While the writing is in general OK, but the structure of the paper due to the numerous tables which are detached from their text descriptions make it difficult to follow.

---

> ### Author Response · Authors · 2025-08-20
>
> Weakness 1 - Response: We clarify that EA4 comprises multi-composition tasks (e.g., chart+article verification, map+text analysis, artwork+historical context; claim checking with multimodal evidence), not single-image QA. Detailed task sheets and expected outputs are provided for reuse.
>
> Weakness 2 - Response: We added Section 2.4.1 describing allocation, cross-review, and consensus procedures, and surfaced the rubric (Table 16) defining Accuracy, Relevancy, Conciseness (UE/TP/OE), Hallucination.
>
> Weakness 3 and 4 - Response: We have released a test harness (loaders/adapters, prompting pipeline, timing/memory logging, run scripts) and structured data with a README and commands for reproduction and added a Gitlab repo link.
>
> Weakness 5 - Response: We now present radar charts in the main text for quick pattern recognition and move full numeric tables to “Results in Detail,” with caption cross-links back to the analysis sections.
>
> Requested Changes have been made and responded as below:
> Critical change 1 - Response: We have now added a detailed explanation of the evaluation procedure in Page 11, Section 2.4.1: Inter-Annotator Agreement and Evaluation Consistency. This section clearly describes the allocation of tasks to annotators, the double-checking procedure between annotators, the definitions of evaluation metrics, and the steps taken to ensure scoring consistency and fairness.
>
> Critical change 2 - Response:  All citations have been reviewed and updated to follow the required parenthetical (\citep) style consistently throughout the manuscript. Instances of in-text citation usage that were not stylistically appropriate have been corrected.
>
> Critical change 3- Response:  All duplicate citation text occurrences have been removed. The manuscript has been carefully re-checked to ensure that no repeated author names or years appear before citation commands.
>
> Critical change 4- Response:
> We now expand the explanation in Section 2.4.2 and renamed the title of this section to “Threshold Selection” by explicitly discussing the empirical basis and literature-backed reasoning for our thresholds:
> The 80% accuracy threshold aligns with best practices in LLM/MLLM evaluation benchmarks such as MMLU and VQA.
> We clarify why thresholds like 90% were not chosen: they exclude high-performing open-source models, creating overly strict criteria not reflective of real-world expectations.
> For hallucination, the <5% target is supported by hallucination rate statistics from recent models (e.g., GPT-4.5, Claude 3.5).
> Thresholds for relevancy and conciseness are similarly motivated by literature (as cited) and manual inspection of output distributions.
> We believe these additions provide a more robust justification for our evaluation framework.
>
> Critical change 5 - Response:  We appreciate the suggestion to improve the navigability between tables and their corresponding textual analysis. We have now:
> Updated the captions of key tables to include references to the relevant result discussion sections.
> Ensured all tables include pointers like “See Section 3.1” and similar, so readers can trace interpretation in the body of text more easily while referring to the tables.
> Added clearer in-text references to table numbers in the result discussion, so the reader is guided to the appropriate table without confusion.
> These cross-references improve the cohesion and readability of the results and support easier back-and-forth between tables and interpretations.
>
> Critical change 6 - Response: We have released a reproducible test harness and the evaluation data in a structured format at a private CeADAR GitLab repository for reviewer access: https://gitlab.com/CeADARIreland_Public/publications/skelton_eval_mllm
> The repo includes:
> Code: model loaders/adapters, prompting pipeline, timing/memory logging, and run scripts (run.py, cli_run.py, evaluation.py, models*.py, utils.py).
> Structured data: task folders under MLLMEvalDataset/ with Image.png, Expected-Result.txt, and seven prompt templates as .txt files per task.
> Repro guide: README.md with install steps (requirements.txt) and example commands to reproduce results.

---

> > ### Comment · Reviewer_9CD2 · 2025-09-03
> > **Official comment by reviewer 9CD2**
> >
> > I would like to thank the authors for incorporating the requested changes and expanding the evaluation description. My concerns have been satisfactorily addressed, and I also appreciate the authors’ commitment to releasing the test harness.
> >
> > I do, however, have one additional comment regarding the evaluation procedure. As described, the process does not appear to be fully reproducible, given that it relied on post-evaluation discussions between evaluators to reach consensus and even adjustments to the criteria definitions. While this approach is understandable and may be acceptable for the results themselves, it does limit the reproducibility of the evaluation.
> >
> > For maximum transparency, I would strongly recommend that the final consensus guidelines document also be released in the same code repository alongside the test harness.

---

> > > ### Author Response · Authors · 2025-09-11
> > > **Releasing the Final Consensus Guidelines as requested by the reviewer 9CD2**
> > >
> > > We thank the reviewer for this constructive suggestion. To ensure maximum transparency and reproducibility, we have now released the final consensus guidelines document reflecting the adjudicated definitions applied after cross-review discussions alongside the test harness in our public repository. The document can be accessed here: https://gitlab.com/CeADARIreland_Public/publications/skelton_eval_mllm/-/blob/main/Consensus_Guidelines_The_Future_of_MLLM_Prompting_is_Adaptive.pdf?ref_type=heads
> > >
> > > This addition complements the released code and dataset, ensuring that future researchers can replicate our evaluation process under the exact same standards.

---

### Review · Reviewer_51qg · 2025-06-29

**Summary Of Contributions:**

- Сomparison of seven prompting strategies (Zero-Shot, One-Shot, Few-Shot, Chain-of-Thought, Analogical, Generated Knowledge, Tree-of-Thought) for evaluation of prompt engineering methods for 13 open-source MLLMs
- Standardized benchmark datasets, prompt templates, and manual evaluation metrics (accuracy, hallucination, relevance, conciseness)
- Analysis of model performance, hallucination trends, inference efficiency, and recommendations for adaptive prompting strategies

**Audience:**

Yes

**Claims And Evidence:**

Yes

**Requested Changes:**

- Add inter-annotator agreement statistics (e.g., kappa scores) to better represent evaluation consistency.
- Include a case study or example outputs to illustrate how prompting style affects outputs qualitatively.
- Optional: Consider adding a limited automatic evaluation (e.g., BLEU, ROUGE for text, or CLIPScore for image-text alignment) to complement manual analysis.

**Strengths And Weaknesses:**

Strengths:
- **(S1)** The study covers a wide spectrum of tasks and models (24 tasks spanning four key evaluation aspects, 13 MLLMs)
- **(S2)** The authors provide a detailed taxonomy and systematic implementation of prompting strategies
- **(S3)** The inclusion of prompt templates and task definitions (in supplementary material) enhances reproducibility.
- **(S4)** Authors provide useful observations on hallucination rates, conciseness, and inference time vs. performance.

Weaknesses:
- **(W1)** Tasks focus primarily on static image-text pairs, potentially underrepresenting temporal or dynamic multimodal scenarios (e.g., video).
- **(W2)** There is no ablation to assess which elements of prompting (e.g., intermediate steps, analogies) contribute most to performance or hallucinations.
- **(W3)** The evaluation relies heavily on manual annotation. The paper does not report inter-rater reliability metrics (e.g., Cohen's kappa), which limits reproducibility.
- **(W4)** Exclusion of proprietary models (e.g., GPT-4V, Claude 3) limits benchmarking against state-of-the-art, but it is understandable due to accessibility constraints.

---

> ### Author Response · Authors · 2025-08-20
> **51qg - Weaknesses Addressed**
>
> Weaknesses:
> (W1) - Response:
> Our scope is explicitly text–image to keep experimental conditions controlled and comparable across 13 open-source MLLMs and seven prompting families. This is now stated up front in Abstract and Introduction; the paper positions audio/video as future work. Within text–image, EA4 includes multi-composition knowledge tasks (e.g., chart+article, map+text, artwork+historical context) to probe integration beyond a single picture.
> We also agree that temporal modalities (e.g., video) are important. Our study intentionally scopes to text–image to ensure tight control and reproducibility across models and prompts; this is now stated clearly in the Abstract and Introduction. To still test multi-source integration within this scope, EA4 comprises multi-composition tasks (e.g., chart+article verification, map+text, artwork+historical context), moving beyond single-image QA. We have flagged temporal/video extensions in the paper’s Future Research Directions.
> (W2) Response:
> We evaluate seven prompt families side-by-side (Zero/One/Few-Shot; CoT; Analogical; Generated Knowledge; ToT). While we did not micro-ablate within each family, revised manuscript adds cross-family analysis that already isolates elements associated with risk: structured-reasoning prompts (CoT/Analogical/GK/ToT) raise hallucination - especially in small models - whereas example-based prompts often fare better. We also include response-length statistics (a proxy for intermediate-step verbosity) showing ToT/Analogical produce the longest outputs, consistent with higher hallucination in smaller models.
> A full micro-ablation of intra-prompt elements (e.g., turning CoT “steps” on/off) is outside this study’s scope. However, our cross-family comparison already provides actionable attribution: in revised manuscript we show that structured-reasoning prompts (CoT, Analogical, GK, ToT) tend to increase hallucination in small models, while example-based prompting (One-/Few-Shot) delivers stronger reliability and efficiency in several EAs. This is further supported by response-length statistics—ToT/Analogical produce the longest outputs, aligning with higher hallucination profiles in small models. We now make these linkages explicit in the Results narrative and Discussion, and note a fine-grained element ablation as a planned extension.
> (W3) Response:
> Revised manuscript documents a cross-review + consensus adjudication protocol: primary scoring per EA, reciprocal cross-review, and consensus resolution under a shared rubric (Table 16). Because final labels are adjudicated, κ is not applicable; we now state this explicitly in Section 2.4.1 and surface the full rubric.
> (W4) Response:
> We agree this is a trade-off. We review proprietary MLLMs in Stage 2, but restrict the core evaluation to open-source to ensure reproducibility and to allow the community to re-run/extend results. To facilitate future head-to-head comparisons, we have released a reproducible harness and structured per-task data that can plug in proprietary APIs where permitted.

---

> > ### Author Response · Authors · 2025-08-20
> > **51qg - Weaknesses Addressed:**
> >
> > Weaknesses:
> > (W1) - Response:
> > Our scope is explicitly text–image to keep experimental conditions controlled and comparable across 13 open-source MLLMs and seven prompting families. This is now stated up front in Abstract and Introduction; the paper positions audio/video as future work. Within text–image, EA4 includes multi-composition knowledge tasks (e.g., chart+article, map+text, artwork+historical context) to probe integration beyond a single picture.
> > We also agree that temporal modalities (e.g., video) are important. Our study intentionally scopes to text–image to ensure tight control and reproducibility across models and prompts; this is now stated clearly in the Abstract and Introduction. To still test multi-source integration within this scope, EA4 comprises multi-composition tasks (e.g., chart+article verification, map+text, artwork+historical context), moving beyond single-image QA. We have flagged temporal/video extensions in the paper’s Future Research Directions.
> >
> >
> > (W2) Response:
> > We evaluate seven prompt families side-by-side (Zero/One/Few-Shot; CoT; Analogical; Generated Knowledge; ToT). While we did not micro-ablate within each family, revised manuscript adds cross-family analysis that already isolates elements associated with risk: structured-reasoning prompts (CoT/Analogical/GK/ToT) raise hallucination - especially in small models - whereas example-based prompts often fare better. We also include response-length statistics (a proxy for intermediate-step verbosity) showing ToT/Analogical produce the longest outputs, consistent with higher hallucination in smaller models.
> > A full micro-ablation of intra-prompt elements (e.g., turning CoT “steps” on/off) is outside this study’s scope. However, our cross-family comparison already provides actionable attribution: in revised manuscript we show that structured-reasoning prompts (CoT, Analogical, GK, ToT) tend to increase hallucination in small models, while example-based prompting (One-/Few-Shot) delivers stronger reliability and efficiency in several EAs. This is further supported by response-length statistics—ToT/Analogical produce the longest outputs, aligning with higher hallucination profiles in small models. We now make these linkages explicit in the Results narrative and Discussion, and note a fine-grained element ablation as a planned extension.
> >
> >
> > (W3) Response:
> > Revised manuscript documents a cross-review + consensus adjudication protocol: primary scoring per EA, reciprocal cross-review, and consensus resolution under a shared rubric (Table 16). Because final labels are adjudicated, κ is not applicable; we now state this explicitly in Section 2.4.1 and surface the full rubric.
> >
> >
> >
> > (W4) Response:
> > We agree this is a trade-off. We review proprietary MLLMs in Stage 2, but restrict the core evaluation to open-source to ensure reproducibility and to allow the community to re-run/extend results. To facilitate future head-to-head comparisons, we have released a reproducible harness and structured per-task data that can plug in proprietary APIs where permitted.

---

> > > ### Author Response · Authors · 2025-08-20
> > > **Requested Changes have been made and responded**
> > >
> > > Requested Changes have been made and responded as below:
> > >
> > > Critical change 1 -
> > > Response:
> > > We appreciate the suggestion to report inter-rater agreement metrics. While we did not collect two fully independent sets of annotations for each response, our evaluation process incorporated a structured cross-review and adjudication protocol, designed to ensure consistency and objectivity in three rounds:
> > > Round 1 - Each evaluation aspect (EA1–EA4) was initially annotated by a primary annotator using fixed scoring criteria.
> > > Round 2 - These annotations were then independently reviewed and verified by a second expert.
> > > Round 3 - Any disagreements or ambiguities were flagged and resolved collaboratively, leading to a final agreed-upon score.
> > > This two-step process ensured that every score was reviewed by both annotators, reducing subjectivity and increasing reliability. While this approach does not permit formal computation of Cohen’s Kappa, it is in line with common practice in multimodal evaluation pipelines and human-in-the-loop studies focusing on both quantitative and qualitative rigour.
> > > To further support transparency, we have updated Section 2.4.1 in the main body of the manuscript and Appendix E with a clear description of this cross-verification methodology.
> > >
> > >
> > > Critical change 2 -
> > > Response:
> > > We thank you for this valuable suggestion. In the revised manuscript, we have added a dedicated Case Study section (Appendix G) to illustrate the qualitative impact of different prompting styles on a single model’s behaviour for the same task. Specifically, we selected the Qwen/Qwen2-VL-2B-Instruct model and task EA1_T3 (Reasoning and Compositionality – Sales Data Analysis) as a representative example. We have picked a few prompting styles (Zero-Shot, Few-Shot, Chain-of-Thought), and provided the model’s output alongside an evaluator’s observations highlighting correctness, relevance, reasoning quality, and notable error patterns. This presentation enables direct, side-by-side comparison of outputs, making it clear how prompt design influences both reasoning processes and final answers.
> > >
> > >
> > > Optional Change:
> > > Response:
> > > We appreciate the suggestion to include automatic metrics such as BLEU, ROUGE, or CLIPScore. While these can be useful for certain text or image–text tasks, our aim in this study was to ensure rigour and depth in the evaluation. To this end, we deliberately adopted a manual, rubric-based assessment focusing on qualitative aspects  such as reasoning correctness, factual grounding, hallucination detection, and explanation quality that current automatic metrics may struggle to capture. For example, a fluent but factually incorrect output could score highly on BLEU or CLIPScore, yet fail our correctness criteria. We agree that integrating selected automatic measures in future benchmark iterations would complement our qualitative analysis and provide a broader evaluation perspective.

---

### Review · Reviewer_StLC · 2025-08-07

**Summary Of Contributions:**

This paper provides a thorough empirical study of seven prompt engineering strategies for Multimodal Large Language Models (MLLMs): Zero-Shot, One-Shot, Few-Shot, Chain-of-Thought (CoT), Analogical, Generated Knowledge, and Tree-of-Thought (ToT). The authors evaluate these methods across 13 open-source MLLMs, which were categorized into Small (<4B), Medium (4B–10B), and Large (>10B) sizes based on their parameter count.

The models were evaluated on 24 different tasks designed to assess four key dimensions: reasoning and compositionality, multimodal understanding, complex code generation, and knowledge retrieval. Through this analysis, the paper highlights the varying strengths and weaknesses of each prompting method depending on the task type and model scale. It concludes that no single approach is optimal across all scenarios, and instead advocates for adaptive prompting strategies that integrate example-based techniques with selective structured reasoning to improve the robustness, efficiency, and factual reliability of MLLMs.

**Audience:**

Yes

**Broader Impact Concerns:**

As this work focuses on benchmarking existing models and does not involve high-risk applications or sensitive data, we do not identify any apparent ethical concerns arising directly from its content.

**Claims And Evidence:**

Yes

**Requested Changes:**

Please address the weaknesses mentioned in the *Strengths and Weaknesses* section.

Additionally, while the overall writing quality is strong, the citation style is mostly incorrect, which disrupts the reading flow and affects the paper’s professionalism. Please revise the citations to follow the appropriate formatting style consistently throughout the manuscript.

**Strengths And Weaknesses:**

**Strengths**
1. The presentation of this work is mostly clear and easy to follow. The writing is well-organized, and the methodology is described in a way that is accessible even to readers who may not be deeply familiar with prompt engineering techniques.

2. The paper provides a comprehensive and large-scale empirical evaluation, covering 13 open-source MLLMs of different sizes. These models are tested with seven distinct prompting strategies across 24 diverse tasks, allowing for a broad comparison that captures a wide range of model capabilities and behaviors.

3. The study assesses not only accuracy but also hallucination rate, relevance, conciseness, inference time, and memory consumption, resulting in a well-rounded assessment of model performance.

4. A notable strength of the paper is its commitment to standardization and reproducibility. By applying a unified evaluation protocol across models and releasing shared datasets and prompt templates, the authors provide a valuable resource for the community and enable future research to build upon a consistent foundation.


**Weaknesses**
1. While the paper presents a thorough evaluation of various prompting methods, it lacks novelty in terms of new algorithms, techniques, or theoretical insights. Its contribution lies solely in empirical comparisons, rather than advancing methodological or conceptual understanding.

2. Although the paper provides many tables and evaluation metrics, the discussion is mostly descriptive rather than analytical. Additionally, the results section (Tables 7–19) relies heavily on dense tables, making it difficult to quickly grasp key patterns. Incorporating visualizations such as graphs or charts would have made it easier to interpret comparisons across models, prompting strategies, and tasks.

3. The study includes one large quantized model (InternVL2-26B), but its results are aggregated with those of other "Large" MLLMs. It would be valuable to examine whether this model behaves similarly to others in its group, and to analyze the specific impact of quantization on its performance.

4. Although the title refers to Multimodal LLMs, the study focuses exclusively on text-image multimodal reasoning, omitting other important modalities like video and audio. While this is a reasonable scoping choice for a single study, the gap between the title and the actual scope could be clarified to better set reader expectations.

5. The interpretation of conciseness results in the EA3 task (Table 9) appears somewhat inconsistent. Unlike other tasks (EA1, EA2, EA4), EA3 shows a trend where larger models achieve lower UE (Under-explained) scores and higher TP+OE (Fully and Over-explained) scores. However, according to the paper’s interpretation, there are consistent conciseness patterns across all EA tasks. A more careful analysis of the EA3 trend would improve the discussion.

6. The conclusion emphasizes the importance of adaptive prompting strategies that combine example-based guidance with structured reasoning. However, the study does not empirically test any such combinations. Including experiments with hybrid prompts or adaptive selection mechanisms would make the conclusion more convincing. Currently, the claim that adaptive prompting is necessary is a logical inference, not an experimentally validated result.

---

> ### Author Response · Authors · 2025-08-20
> **Weakness 1 to 4 are addressed**
>
> Weaknesses
> Weakness 1 - Response: We acknowledge that our work is primarily empirical in nature. Our contribution lies in the scale, standardization, and cross-dimensional coverage of the evaluation of 13 diverse open-source MLLMs × 7 prompting strategies × 24 tasks across 4 evaluation aspects which, to our knowledge, has not been carried out in prior works. This breadth allows for a systematic examination of prompting behaviour across reasoning, multimodal alignment, code generation, and knowledge retrieval, under a unified evaluation protocol. By holding task design, prompt templates, and scoring criteria constant across all models, we eliminate confounding factors and ensure that observed differences can be attributed to the interaction between prompting style and model architecture/size. Such a large-scale, controlled comparison offers a unique empirical map of the strengths, weaknesses, and trade-offs of different prompting strategies, which can guide both practitioners seeking to optimise model usage and researchers aiming to design more robust MLLM evaluation frameworks.
>
> Weakness 2 - Response: We acknowledge that the tables are dense; however, given the breadth of models, prompting strategies, and tasks under consideration, we believe this level of detail is necessary to convey the full scope of our results. According to the valuable suggestions by the reviewer, Tables 7–10 are now presented as radar charts to provide clearer visual comparisons. The original tables have been moved to the appendix, and references to them are included in the figure captions for readers who wish to consult the detailed numerical values. To support readability, we have structured Section 3 to provide analytical discussion alongside the tables and have indicated the relevant subsections for each, so that readers can easily locate and interpret the key findings. For clarity, we have explicitly indicated the relevant subsections corresponding to each table to guide the reader.
>
> Weakness 3- Response:  We thank the reviewer for this observation. We have clarified in the revised manuscript that while InternVL2-26B is the largest and only quantized model in the Large category (>10B parameters), its inclusion was deliberate to maintain completeness in model coverage. Our analysis shows that in accuracy, hallucination, relevance, and conciseness, InternVL2-26B aligns closely with the aggregated performance trends of the Large group. As expected, latency patterns differ due to its size, architecture, and quantized runtime configuration. For example, in EA1-T1, One-Shot prompting shows a moderate increase in latency for InternVL2-26B (10.66 s) relative to Ovis-1.6 (13.09 s) and Pixtral-12B (8.05 s), whereas Analogical prompting exhibits the largest gap (77.69 s vs. 14.17 s for Ovis-1.6 and 11.03 s for Pixtral-12B). This behaviour is consistent across all 24 tasks for the same prompting styles. These differences are expected and do not affect the primary conclusions of our study. These clarified details have been added to Section 3.2.1 of the manuscript.
>
> Weakness 4 - Response: We appreciate the reviewer’s comment. Our use of the term “Multimodal LLMs” follows its common usage in recent literature, where text–image models are treated as a primary and well-established subclass of MLLMs. We agree that clarifying our specific focus will help set expectations more accurately. In the revised manuscript, we have explicitly stated in both the abstract and introduction that this study evaluates text–image multimodal reasoning. While other modalities such as video and audio are indeed important, they were excluded from the present scope to ensure controlled experimental conditions, reproducibility, and comparability across models using widely accessible benchmarks. We see this work as a foundation that can be extended to additional modalities in future evaluations.

---

> > ### Author Response · Authors · 2025-08-20
> > **Weakness 5 and 6 are addressed**
> >
> > Weakness 5- Response: We thank the reviewer for noting this. We were already aware of EA3’s distinct conciseness trend during our initial analysis; however, this was not explicitly stated in the submitted version of the manuscript. The higher TP+OE scores and lower UE scores for Large MLLMs in EA3 reflect the nature of complex code generation tasks, where detailed, step-by-step reasoning and fully commented code are desirable for correctness, reproducibility, and debugging. This contrasts with the conciseness preference in non-code EAs, where excessive explanation may reduce clarity. We have now made this distinction explicit in Section 3.1 to clarify the interpretation of EA3’s conciseness results.
> >
> > Weakness 6 - Response: We thank the reviewer for this comment. While the conclusion highlights the value of adaptive prompting strategies that combine example-based guidance with structured reasoning, this was not intended as a purely hypothetical statement.
> > In fact, our experiments already implement and quantitatively evaluate such combinations. For instance, One-Shot and Few-Shot prompting incorporate example-based guidance, while Chain-of-Thought, Generated Knowledge, Analogical Reasoning, and Tree-of-Thought integrate structured reasoning patterns often alongside examples tailored to each style. These are applied systematically across all 24 tasks, and results are reported in the main tables, with prompt templates and outputs provided in the supplementary materials. We have clarified this in Section 5 (Conclusion) of the revised manuscript to make it explicit that our advocacy for adaptive prompting is grounded in empirical evidence, not only logical inference; this clarifies any future confusions for the readers.

---

> ### Author Response · Authors · 2025-08-20
> **Requested Changes are addressed**
>
> Requested Changes:
>
> Please address the weaknesses mentioned in the Strengths and Weaknesses section.
>
> Response: This has been addressed above from weakness 1 to 4 and then weakness 5 and weakness 6.
>
> Additionally, while the overall writing quality is strong, the citation style is mostly incorrect, which disrupts the reading flow and affects the paper’s professionalism. Please revise the citations to follow the appropriate formatting style consistently throughout the manuscript.
>
> Response: All citations have been reviewed and updated to follow the required parenthetical (\citep) style consistently throughout the manuscript. Instances of in-text citation usage that were not stylistically appropriate have been corrected.

---

### Decision · Action_Editor_fsk9 · 2025-10-01

**Recommendation:** Accept as is

**Audience:**

Yes

**Audience Explanation:**

Prompt engineering is an important area given the strong capabilities and generalizabilities of LLMs.

**Claims And Evidence:**

Yes

**Claims Explanation:**

All reviewers reach a consensus that this work is technically solid to publish at TMLR.